# Spike deep mutational scanning helps predict success of SARS-CoV-2 clades

Bernadeta Dadonaite[1], Jack Brown[2], Teagan E. McMahon[1], Ariana G. Farrell[1], Marlin D. Figgins[3,4], Daniel Asarnow[2], Cameron Stewart[2], Jimin Lee[2], Jenni Logue[5], Trevor Bedford[3,6,7], Ben Murrell[8], Helen Y. Chu[5], David Veesler[2,7] & Jesse D. Bloom[1,7 ✉]

SARS-CoV-2 variants acquire mutations in the spike protein that promote immune evasion[1] and affect other properties that contribute to viral fitness, such as ACE2 receptor binding and cell entry[2,3]. Knowledge of how mutations affect these spike phenotypes can provide insight into the current and potential future evolution of the virus. Here we use pseudovirus deep mutational scanning[4] to measure how more than 9,000 mutations across the full XBB.1.5 and BA.2 spikes affect ACE2 binding, cell entry or escape from human sera. We find that mutations outside the receptor-binding domain (RBD) have meaningfully affected ACE2 binding during SARS-CoV-2 evolution. We also measure how mutations to the XBB.1.5 spike affect neutralization by serum from individuals who recently had SARS-CoV-2 infections. The strongest serum escape mutations are in the RBD at sites 357, 420, 440, 456 and 473; however, the antigenic effects of these mutations vary across individuals. We also identify strong escape mutations outside the RBD; however, many of them decrease ACE2 binding, suggesting they act by modulating RBD conformation. Notably, the growth rates of human SARS-CoV-2 clades can be explained in substantial part by the measured effects of mutations on spike phenotypes, suggesting our data could enable better prediction of viral evolution.

Over the past 4 years of SARS-CoV-2 evolution, the virus has accumulated mutations throughout its genome. The most rapid evolution has occurred in the viral spike, for instance, the XBB-descended variants that dominated in 2023 have 45–48 spike protein mutations relative to the earliest known strains from Wuhan in late 2019. The reason for this rapid evolution is that spike mutations can strongly affect both the virus's inherent transmissibility and ability to escape pre-existing immunity[1,3]. A crucial aspect of interpreting and forecasting SARS-CoV-2 evolution is therefore understanding the impact of current and potential future mutations on the spike.

Here we measure how thousands of mutations to the spike glycoprotein of the XBB.1.5 and BA.2 SARS-CoV-2 strains affect three molecular phenotypes critical to viral evolution: cell entry, ACE2 binding and neutralization by human polyclonal serum (Fig. 1a). To do this, we extend a recently described pseudotyped lentivirus deep mutational scanning system[4] that enables safe experimental characterization of mutations throughout the spike[5]. We demonstrate that mutations outside the RBD can substantially affect spike binding to ACE2. We also define the mutations that escape neutralization by sera from humans who have been multiply vaccinated and also recently infected by XBB or one of its descendant lineages (XBB*), and show there is appreciable heterogeneity in the antigenic impact of mutations across individuals. Finally, we show that the spike phenotypes we measure explain much of the changes in viral growth rate among

different SARS-CoV-2 clades that have emerged in humans over the past few years.

## Design of spike mutant libraries

We created mutant libraries of the spikes from the XBB.1.5 and BA.2 strains. We chose these strains because nearly all human SARS-CoV-2 circulating at present descends from either BA.2 or XBB.1.5's parent lineage XBB[6], and because XBB.1.5 is the sole component of the COVID-19 booster vaccine recommended by the WHO in 2023 (ref. 7). We wanted the libraries to contain all evolutionary accessible amino-acid mutations tolerable for spike function. We therefore included all mutations observed an appreciable number of times among the millions of SARS-CoV-2 sequences in Global Initiative on Sharing All Influenza Data (GISAID). In addition, we included all possible mutations at sites that change often during SARS-CoV-2 evolution or are antigenically important[1,8], and deletions at key N-terminal domain (NTD) and RBD sites. These criteria led us to target roughly 7,000 amino-acid mutations in each of the XBB.1.5 and BA.2 libraries (Extended Data Fig. 1a). We created two independent libraries for each spike so we could perform all deep mutational scanning in full biological duplicate. The actual libraries contained between 69,000 and 102,000 barcoded spike variants with an average of two mutations per variant, and covered 99% of the targeted mutations, as well as some extra mutations (Extended

[1]Basic Sciences Division and Computational Biology Program, Fred Hutchinson Cancer Center, Seattle, WA, USA. [2]Department of Biochemistry, University of Washington, Seattle, WA, USA. [3]Vaccine and Infectious Disease Division, Fred Hutchinson Cancer Center, Seattle, WA, USA. [4]Department of Applied Mathematics, University of Washington, Seattle, WA, USA. [5]University of Washington, Department of Medicine, Division of Allergy and Infectious Diseases, Seattle, WA, USA. [6]Department of Epidemiology, University of Washington, Seattle, WA, USA. [7]Howard Hughes Medical Institute, Seattle, WA, USA. [8]Department of Microbiology, Tumor and Cell Biology, Karolinska Institutet, Stockholm, Sweden. ✉e-mail: jbloom@fredhutch.org

Data Fig. 1a). To retrospectively validate that this library design covered most evolutionarily important mutations, we confirmed that our XBB.1.5 libraries provided adequate coverage for high-confidence experimental measurements of nearly all spike mutations now present in XBB, BA.2 and BA.2.86-descended Pango clades—despite the fact that BA.2.86 had not even emerged yet at the time we designed the library (Extended Data Fig. 1b). So although our libraries do not contain all spike mutations, they cover nearly all mutations that are relevant in the near- to mid-term evolution of SARS-CoV-2. Because the RBD is an especially important determinant of ACE2 binding and serum antibody escape[9], we also made duplicate XBB.1.5 libraries that saturated all amino-acid mutations in only the RBD (Extended Data Fig. 1a).

## Effects of spike mutations on cell entry

We measured the effects of all library mutations on spike-mediated cell entry in 293T-ACE2 cells (Extended Data Fig. 1c,d and interactive heat maps at https://dms-vep.github.io/SARS-CoV-2_XBB.1.5_spike_DMS/htmls/293T_high_ACE2_entry_func_effects.html and https://dms-vep.github.io/SARS-CoV-2_Omicron_BA.2_spike_ACE2_binding/htmls/293T_high_ACE2_entry_func_effects.html). These measurements were highly correlated between the replicate libraries for each spike, indicating the experiments have good repeatability (Extended Data Fig. 1e). The effects of mutations were also well correlated between the XBB.1.5 and BA.2 spikes (Extended Data Fig. 1f), consistent with previous reports that most but not all mutations have similar effects on the spikes of different SARS-CoV-2 variants[10,11]. As expected, stop codons were highly deleterious for cell entry (Fig. 1b). Because our full-spike library design strategy favours functionally tolerated mutations in spike, most amino-acid mutations in our libraries just slightly impaired cell entry and some but not all single-residue deletions were also well tolerated (Fig. 1b). SARS-CoV-2 has acquired numerous deletions in the NTD's flexible loops during its evolution[12,13], and consistent with that evolution we find that the flexible loops but not the core β sheets of the NTD are relatively tolerant of deletions (Extended Data Fig. 1g). Overall, the effects of mutations on cell entry were fairly well correlated with the effects of amino-acid mutations on viral fitness estimated from millions of natural human SARS-CoV-2 sequences[14] (Extended Data Fig. 1h).

No individual mutation in either the XBB.1.5 or BA.2 spikes notably increased pseudovirus cell entry, although some mutations did marginally improve entry (Fig. 1b and interactive heat maps linked in figure legend). One mutation that slightly improves pseudovirus entry in both XBB.1.5 and BA.2 is P1143L (Fig. 1c), which is found in the recently emerged BA.2.86 lineage[15]. We previously reported that mutations to P1143 also improve cell entry for BA.1 and Delta pseudoviruses[4]. The deletion mutations in our libraries are usually more deleterious for cell entry than substitutions (Fig. 1b); however, deletion of V483 in the RBD is well tolerated for cell entry, consistent with emergence of this mutation in the BA.2.86 variant[15]. The F456L mutation, which has emerged repeatedly in XBB clades after being rare in earlier BA.2-derived clades, is well tolerated for cell entry in XBB.1.5 but substantially deleterious in BA.2 (Fig. 1c).

## Non-RBD mutations affect ACE2 binding

To measure how mutations in spike affect receptor binding, we leveraged the fact that the soluble ACE2 ectodomain neutralizes spike-mediated infection with a potency proportional to the strength of spike binding to ACE2 (refs. 1,16). To validate this fact, we made pseudoviruses with six different spike variants and quantified their neutralization by monomeric ACE2 (Fig. 2a). Compared to the BA.2 spike, the Wuhan-Hu-1+D614G spike is neutralized less potently by soluble ACE2 consistent with its weaker ACE2 binding[17,18], whereas four

mutants of BA.2 known to have higher ACE2 binding[2] (N417K, N417F, R493Q and Y453F) were all neutralized more potently by soluble ACE2 (Fig. 2a). The quantitative neutralization by soluble ACE2 was highly correlated with previously measured monomeric RBD-ACE2 affinities[2,18,19] (Fig. 2b).

Using this approach, we measured how mutations across both the XBB.1.5 and BA.2 spikes affect apparent ACE2 binding (Extended Data Fig. 2 and interactive heat maps of all mutation effects at https://dms-vep.github.io/SARS-CoV-2_XBB.1.5_spike_DMS/htmls/monomeric_ACE2_mut_effect.html and https://dms-vep.github.io/SARS-CoV-2_Omicron_BA.2_spike_ACE2_binding/htmls/monomeric_ACE2_mut_effect.html). Because our assay measures ACE2 neutralization rather than 1:1 ACE2-RBD affinity there are several distinct mechanisms that could affect what we refer to as ACE2 binding: direct changes in 1:1 RBD-ACE2 binding affinity[2,20], changes in spike that modulate the conformation of the RBDs (such as up and down movements)[21,22] and ACE2-induced shedding of the $S_1$ subunit[23,24].

The effects of RBD mutations on ACE2 binding to the spike measured using pseudovirus deep mutational scanning correlate well with previously reported measurements from RBD yeast display for both XBB.1.5 and BA.2 (ref. 20) (Fig. 2c). We also measured ACE2 binding for the XBB.1.5 pseudovirus libraries with saturating RBD mutations using both monomeric and dimeric soluble ACE2. The RBD-only pseudovirus measurements were highly correlated with the full-spike library measurements (Extended Data Fig. 3a), and the measured values were highly similar for monomeric versus dimeric soluble ACE2 (Extended Data Fig. 3b). ACE2 binding and pseudovirus cell entry are distinct properties, with no strong correlation between these properties among tolerated mutations (Extended Data Fig. 3c), probably reflecting the fact that cell entry can be limited by factors unrelated to receptor binding, especially in target cells expressing moderate to high levels of ACE2, such as those used in our experiments.

A striking observation from the deep mutational scanning is that some mutations outside the RBD appreciably affect binding to ACE2 (Fig. 2d and Extended Data Figs. 2 and 3). To validate these findings, we used mass photometry to measure binding of the soluble native ACE2 dimer to the spike ectodomain trimer (Fig. 3a). Mass photometry measures protein-protein interactions in solution by detecting changes in light scattering that are proportional to protein molecular mass[25], which allows us to detect binding of one or more ACE2 molecules to the spike (Fig. 3a). We produced prefusion-stabilized HexaPro[26] BA.2 and XBB.1.5 spikes, along with mutants that our deep mutational scanning experiments showed to modulate ACE2 binding, and performed mass photometry in the presence of a series of ACE2 concentrations (Fig. 3a,b, Extended Data Fig. 4 and Supplementary Figs. 1–3). As expected, we observed better and worse ACE2 binding for RBD mutations that have been previously identified to either increase (R493Q) or abrogate (R498V) ACE2 engagement, respectively[2] (Fig. 3b, left panels). Furthermore, we detected increased ACE2 binding for all but one of the BA.2 and XBB.1.5 spike trimers harbouring $S_1$ subunit mutations (in NTD, RBD and SD1 domains) that our deep mutational scanning indicated had better binding (Fig. 3b middle panel, Extended Data Fig. 4 and Supplementary Figs. 2 and 3), as well as decreased ACE2 binding for $S_1$ mutations that our deep mutational scanning indicated had worse binding (Fig. 3b). However, mutations to the BA.2 and XBB1.5 $S_2$ subunit found to increase binding to ACE2 in our deep mutational scanning did not lead to increased ACE2 binding detectable by mass photometry (Fig. 3b right panel, Extended Data Fig. 4b,c and Supplementary Figs. 2 and 3). Notably, some of these $S_2$ mutations were previously reported to affect spike fusion[27–29] suggesting that they may indeed affect $S_1$ shedding and in turn affect ACE2 binding consistent with our deep mutational scanning. However, unlike the spikes in deep mutational scanning experiments, the spikes used in mass photometry experiments are prefusion stabilized by introduction of the HexaPro mutations in the fusion machinery[26].

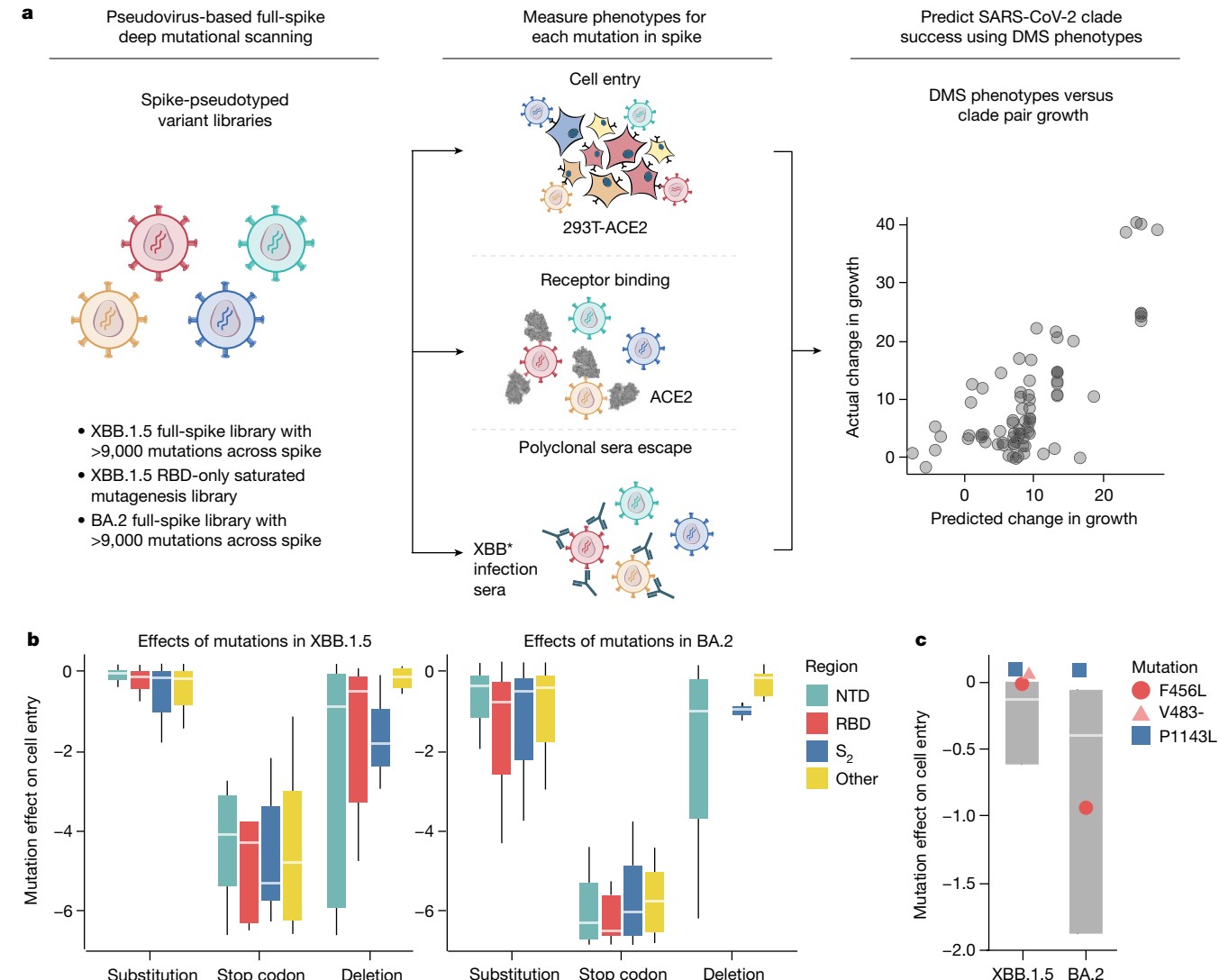

**Fig. 1 | Deep mutational scanning to measure phenotypes of the XBB.1.5 and BA.2 spikes. a**, We measure the effects of mutations in spike on cell entry, receptor binding and serum escape using deep mutational scanning (DMS). We then use these measurements to predict the evolutionary success of human SARS-CoV-2 clades. **b**, Distribution of effects of mutations in XBB.1.5 and BA.2 spikes on entry into 293T-ACE2 cells for all mutations in the deep mutational scanning libraries, stratified by the type of mutation and the domain in spike. Negative values indicate worse cell entry than the unmutated parental spike. Note that the library design favoured introduction of substitutions and deletions that are well tolerated by spike, explaining why many mutations of these types have neutral to only modestly deleterious effects on cell entry. **c**, Cell entry effects of mutations F456L, P1143L and deletion of V483 relative to

the distribution of effects of all substitution and deletion mutations in the libraries. Interactive heat maps with effects of individual mutations across the whole spike on cell entry are at https://dms-vep.github.io/SARS-CoV-2_XBB.1.5_spike_DMS/htmls/293T_high_ACE2_entry_func_effects.html and https://dms-vep.org/SARS-CoV-2_Omicron_BA.2_spike_ACE2_binding/htmls/293T_high_ACE2_entry_func_effects.html. The boxes in **b** and **c** span the interquartile range, with the horizontal white line indicating the median. Whiskers in **b** indicate 0.75 of the interquartile range plotted from the smallest value of the first and highest value of the third quartile. For **c**, the effect of deleting V483 was not measured in the BA.2 spike. The effects of mutations are the mean of two biological replicate measurements made with different deep mutational scanning libraries.

These modifications to spike may limit the propagation of long-range allosteric changes induced by $S_2$ subunit mutations, possibly explaining the discrepancy between deep mutational scanning and mass photometry. Concurring with this hypothesis, we previously showed that ACE2-induced allosteric conformational changes that drive fusion peptide exposure were inhibited by the prefusion-stabilizing 2P mutations[30].

Non-RBD mutations that enhance ACE2 binding have played an important role in SARS-CoV-2 evolution. The following non-RBD mutations that enhance ACE2 binding occurred in the main pre-Omicron variants of concern: A570D (Alpha), A222V (several moderate-frequency Delta sublineages), T1027I (Gamma) and D950N (Delta) (Extended Data Fig. 2d). In addition, the following non-RBD mutations that occurred

in Omicron variants, all of which represent reversions to pre-Omicron residue identities, increase ACE2 binding: K969N, K764N and Y655H. Consistent with previous studies showing that the original D614G mutation increased the proportion of RBDs in the up conformation[21], we find that G614D decreases full-spike ACE2 binding (Fig. 3b and Extended Data Fig. 2d).

To systematically examine the recent evolutionary role of non-RBD-ACE2 binding-enhancing mutations, we tabulated non-RBD mutations that enhance binding and are new mutations in at least four XBB-descended Pango clades (Fig. 3c). Some of these mutations may explain why certain clades had a growth advantage. For example, the NTD mutation Q52H provided the EG.5.1 lineage with a clear growth advantage over EG.5 (ref. 6), despite not measurably

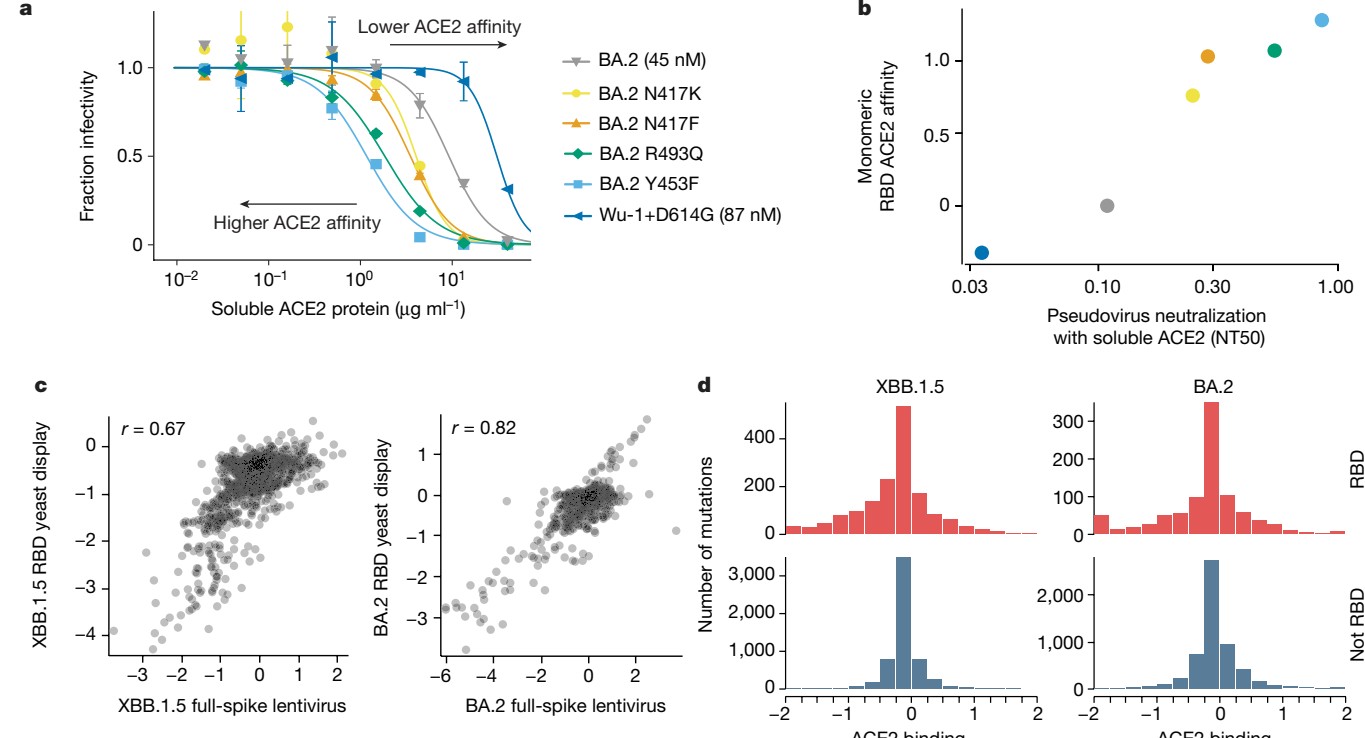

**Fig. 2 | Effects of mutations on full-spike ACE2 binding measured using pseudovirus deep mutational scanning. a**, Neutralization of pseudoviruses with the indicated spikes by soluble monomeric ACE2. Viruses with spikes that have stronger binding to ACE2 are neutralized more efficiently by soluble ACE2 (lower half-maximal neutralizing titers (NT50)), whereas viruses with spikes with worse binding are neutralized more weakly. Error bars indicate standard error between two replicates. ACE2 affinity values measured by surface plasmon resonance for BA.2 and Wu-1+D614G are shown in brackets[18]. **b**, Correlation between neutralization NT50 by soluble ACE2 versus the RBD affinity for ACE2 as measured by titrations using yeast-displayed RBD[2]. **c**, Correlations between the effects of RBD mutations on ACE2 binding measured using the pseudovirus-based approach (this study) and yeast-based RBD display[2,20]. **d**, Distribution of effects of individual mutations on full-spike ACE2 binding for all functionally tolerated mutations in our libraries, stratified by RBD versus non-RBD mutations. Note that effects of magnitude greater than two are clamped to the limits of the plots' $x$ axes. The effects of individual XBB.1.5 spike mutations on ACE2 binding are shown in Extended Data Fig. 3.

affecting serum neutralization[31]. Our deep mutational scanning provides an explanation for the success of EG.5.1 by showing that Q52H enhances ACE2 binding. Similarly, T572I is now appearing convergently in JN.1-descended lineages[6], and our results show that mutation enhances ACE2 binding.

## Heterogeneous sera escape

We next mapped how mutations in spike affect neutralization by the polyclonal antibodies in sera from ten vaccinated individuals who either had a confirmed XBB* infection or whose last infection was during a period when XBB lineages were the dominant circulating variants (Supplementary Table 1). We performed these measurements with the full-spike XBB.1.5 libraries using 293T cells expressing moderate levels of ACE2 that better capture the activities of non-RBD antibodies[32,33], although the key sites of escape were mostly similar if we used 293T cells expressing high levels of ACE2 or the RBD-only libraries (Extended Data Fig. 5). The sites of greatest serum escape were mainly in the RBD (Fig. 4a–c and interactive plot at https://dms-vep.github.io/SARS-CoV-2_XBB.1.5_spike_DMS/htmls/summary_overlaid.html). These sites include 357, 371, 420, the 440–447 loop, 455–456 and 473, as well as a few sites in the NTD, such as positions 200 and 234. At some sites, the escape mutations are strongly deleterious to ACE2 binding (Fig. 4c). For instance, mutations at Y473 cause strong neutralization escape but greatly reduce ACE2 binding, probably explaining their low frequency among circulating SARS-CoV-2 variants. In addition, only some of the antibody escape mutations mapped in our experiments are accessible by single-nucleotide mutations to XBB.1.5 (Fig. 4c). Several escape mutations that are single-nucleotide accessible and do not strongly impair ACE2 binding are found in recent variants, including mutations at site 456 in EG.5.1 and many other XBB variants, mutations at 455 in HK.3.1 and JN.1, mutations at 420 in GL.1 and mutations at 200 in XBB.1.22 (ref. 6).

Whereas the same mutations often escape many sera, there is also heterogeneity such that the sera-average is not fully reflective of the effects of mutations on any individual serum (Fig. 4b,d and Extended Data Fig. 6). For example, whereas mutations to site Y473 strongly escape neutralization by most sera, two sera we analysed (493C and 501C) are largely unaffected by mutations at that site. Other key sites of escape, including 420 and 456, show similar heterogeneity across sera. To validate that escape mutations can have very different effects across sera, we performed standard pseudovirus neutralization assays[5] against a panel of point mutants to the XBB.1.5 spike (Fig. 4d). The changes in neutralization in these validation assays were highly correlated with the escape measured by deep mutational scanning, and confirmed the serum-to-serum heterogeneity. For example, Y473S strongly reduces neutralization by sera 287C and 500C, but actually slightly increases neutralization by serum 501C. Similarly, F456L substantially reduces neutralization by only some sera (Fig. 4d).

The deep mutational scanning identifies mutations that increase, as well as escape, neutralization (Extended Data Fig. 7). Sensitizing mutations often occur at sites that are mutated in XBB.1.5 relative to earlier variants, such as sites 373, 405, 417, 460, 486 and 505 (Extended Data Fig. 7). Presumably in many cases, reverting mutations at these

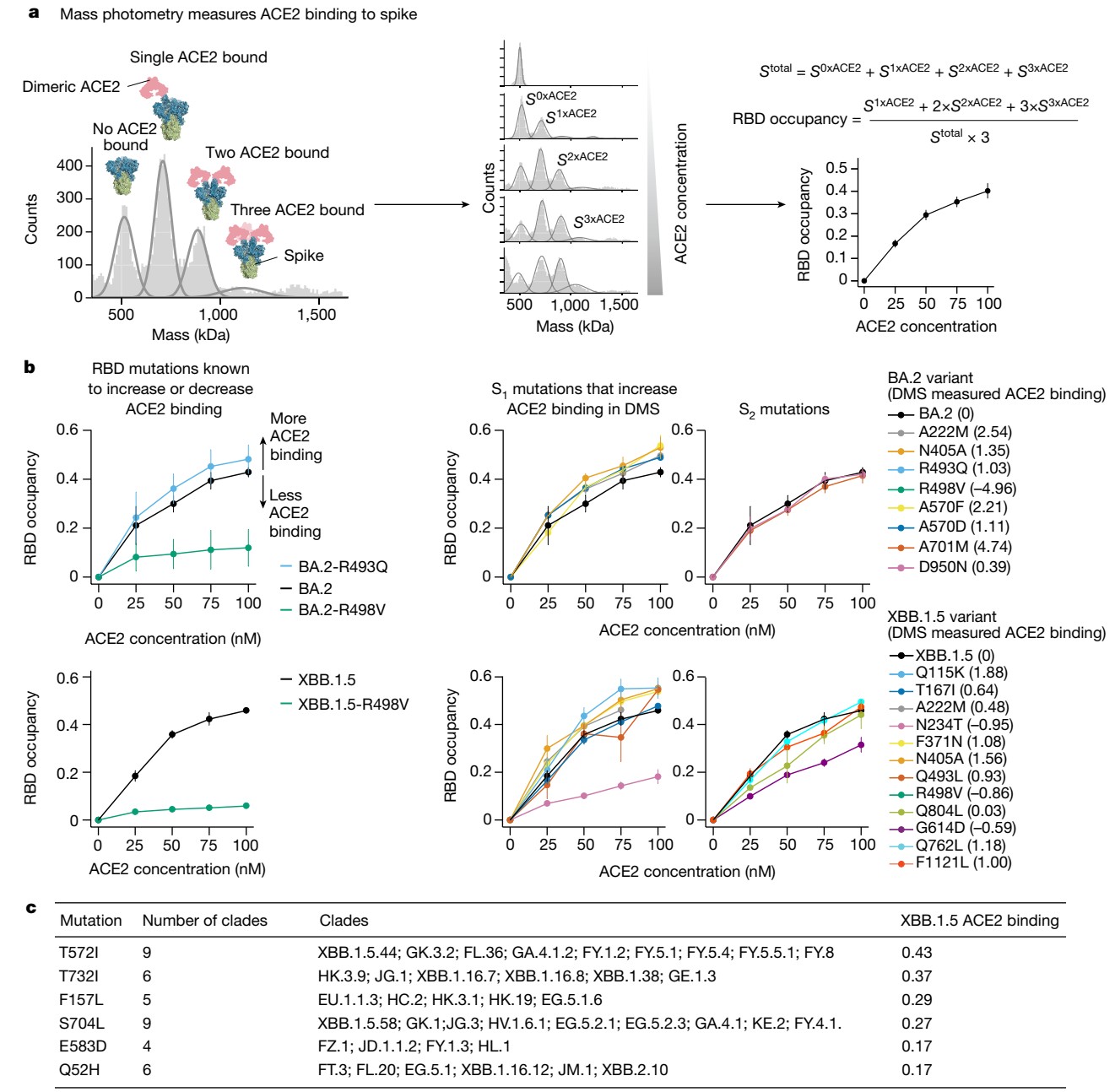

**a** Mass photometry measures ACE2 binding to spike

$$S^{total} = S^{0xACE2} + S^{1xACE2} + S^{2xACE2} + S^{3xACE2}$$

$$RBD\ occupancy = \frac{S^{1xACE2} + 2 \times S^{2xACE2} + 3 \times S^{3xACE2}}{S^{total} \times 3}$$

**b** RBD mutations known to increase or decrease ACE2 binding

BA.2 variant (DMS measured ACE2 binding)
- BA.2 (0)
- A222M (2.54)
- N405A (1.35)
- R493Q (1.03)
- R498V (−4.96)
- A570F (2.21)
- A570D (1.11)
- A701M (4.74)
- D950N (0.39)

XBB.1.5 variant (DMS measured ACE2 binding)
- XBB.1.5 (0)
- Q115K (1.88)
- T167I (0.64)
- A222M (0.48)
- N234T (−0.95)
- F371N (1.08)
- N405A (1.56)
- Q493L (0.93)
- R498V (−0.86)
- Q804L (0.03)
- G614D (−0.59)
- Q762L (1.18)
- F1121L (1.00)

**c**

| Mutation | Number of clades | Clades | XBB.1.5 ACE2 binding |
|---|---|---|---|
| T572I | 9 | XBB.1.5.44; GK.3.2; FL.36; GA.4.1.2; FY.1.2; FY.5.1; FY.5.4; FY.5.5.1; FY.8 | 0.43 |
| T732I | 6 | HK.3.9; JG.1; XBB.1.16.7; XBB.1.16.8; XBB.1.38; GE.1.3 | 0.37 |
| F157L | 5 | EU.1.1.3; HC.2; HK.3.1; HK.19; EG.5.1.6 | 0.29 |
| S704L | 9 | XBB.1.5.58; GK.1;JG.3; HV.1.6.1; EG.5.2.1; EG.5.2.3; GA.4.1; KE.2; FY.4.1. | 0.27 |
| E583D | 4 | FZ.1; JD.1.1.2; FY.1.3; HL.1 | 0.17 |
| Q52H | 6 | FT.3; FL.20; EG.5.1; XBB.1.16.12; JM.1; XBB.2.10 | 0.17 |

**Fig. 3 | Non-RBD mutations affect ACE2 binding. a**, ACE2 binding measurements using mass photometry. The histogram on the left shows distribution of spike molecular mass when no ($S^{0xACE2}$), one ($S^{1xACE2}$), two ($S^{2xACE2}$) or three ($S^{3xACE2}$) ACE2 molecules are bound. We measure how this mass distribution changes as spike is incubated with increasing concentrations of soluble dimeric ACE2. RBD occupancy is the fraction of RBDs bound to ACE2, calculated using Gaussian components for $S^{0xACE}$, $S^{1xACE2}$, $S^{2xACE2}$ and $S^{3xACE2}$ at each ACE2 concentration. **b**, RBD occupancy measured using mass photometry for different BA.2 and XBB.1.5 spike variants. Top left panel shows that a BA.2 spike mutation known to increase ACE2 binding (R493Q/blue) has greater RBD occupancy relative to unmutated BA.2 (black) spike, by contrast a mutation known to decrease ACE2 binding (R498V/green) has lower RBD occupancy in both BA.2 (top left panel) and XBB.1.5 (bottom left panel) backgrounds. Panels on the right show RBD occupancy for BA.2 (top right) and XBB.1.5 (bottom right) spike variants with mutations in $S_1$ or $S_2$ subunits measured to increase ACE2 binding in the deep mutational scanning. Values shown in parentheses after the mutation in the legend are the effect on ACE2 binding measured by deep mutational scanning. Error bars in plots **a** and **b** indicate standard error between two replicates. **c**, Non-RBD mutations measured to increase ACE2 binding in deep mutational scanning experiments that have arisen independently as defining mutations in at least four XBB-descended clades.

sites restores neutralization by antibodies elicited by infection or vaccination with earlier viral strains. To confirm that the sensitizing mutations identified in the deep mutational scanning actually increased neutralization, we validated the sensitizing effects of R403K and N405K in standard pseudovirus neutralization assay (Fig. 4d). In addition, some sensitizing mutations seem to act by placing the RBD in a more up conformation as discussed in the next subsection.

## RBD conformation affects serum escape

Most sites of strong escape described in the previous section are proximal to the ACE2-binding motif in the RBD that is the target of many potent neutralizing antibodies[34,35]. However, the deep mutational scanning also reveals individual mutations at non-RBD or ACE2-distal RBD sites that strongly escape neutralization. Some of these sites,

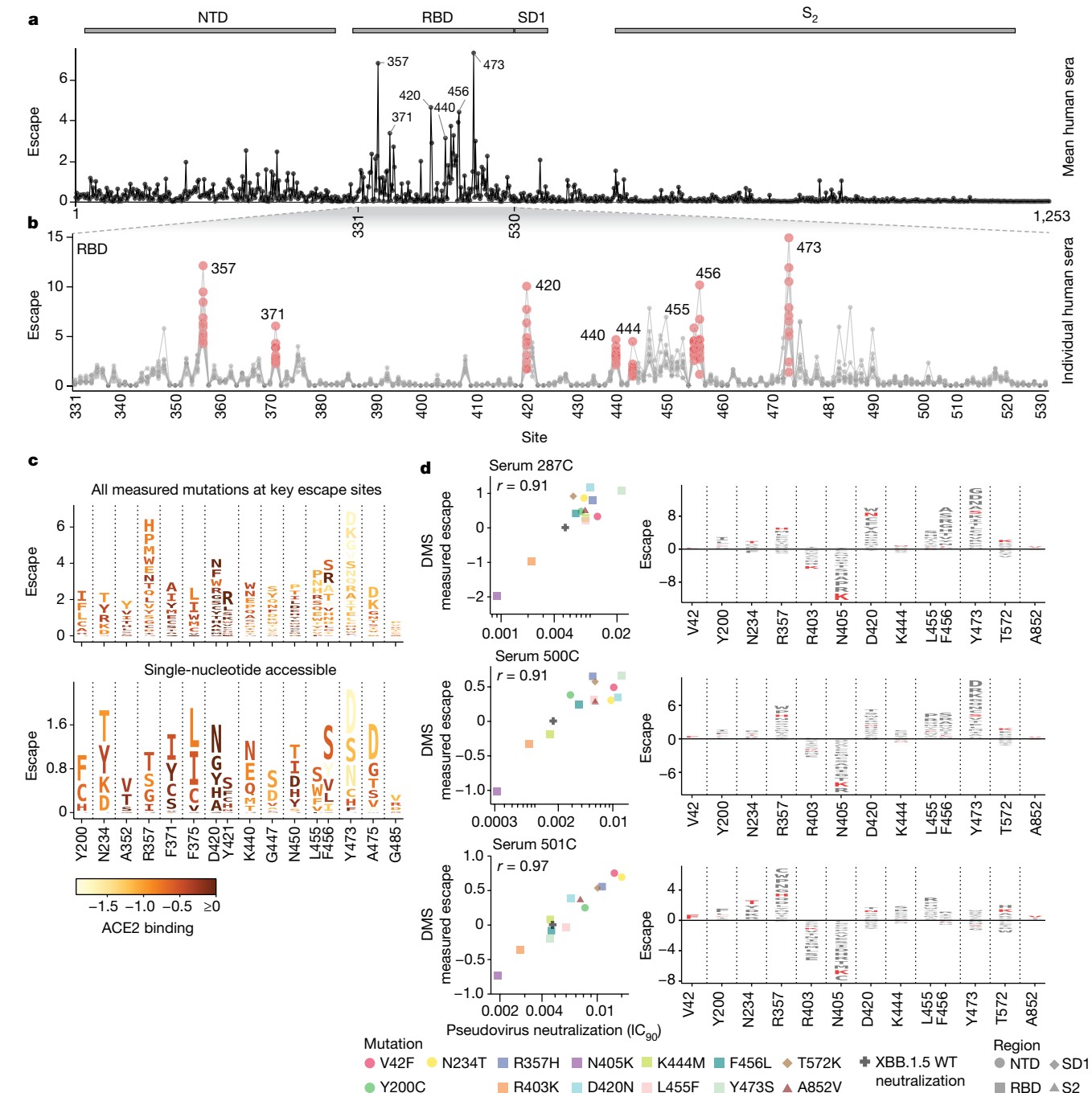

**Fig. 4 | Serum antibody escape mutations for individuals with previous XBB\* infections. a**, Escape at each site in the XBB.1.5 spike averaged across ten sera collected from individuals with previous XBB* infections. The points indicate the total positive escape caused by all mutations at each site. See https://dms-vep.github.io/SARS-CoV-2_XBB.1.5_spike_DMS/htmls/summary_overlaid.html for an interactive version of this plot with extra mutation-level data. **b**, Enlarged view of the escape at each site in RBD with each line representing one of the ten sera. Key sites are labelled with red circles indicating escape for each of the ten sera. Red data points indicate escape for each individual at select RBD positions. **c**, Logo plots showing the 16 sites of greatest total escape after averaging across the sera. Letter heights indicate escape caused by mutation to that amino acid, and letters are coloured light yellow to dark brown depending on the impact of that mutation on ACE2 binding (see colour key). The top plot shows all amino-acid mutations measured, and the bottom plot shows only amino acids accessible by a single-nucleotide mutation to the XBB.1.5 spike. **d**, The left shows a correlation between DMS escape scores and pseudovirus neutralization assay IC90 values for three sera. The right is a logo plot showing escape for all sites with mutations validated in the neutralization assays, with the specific validated mutations in red.

such as 42, 200 and 234 in the NTD, 572 in SD1 and 852 in S2 have mutations that cause as much escape as ACE2-proximal RBD mutations, decreasing serum neutralization by as much as sixfold (Fig. 4a,d). Whereas most mutations at any given site have similar effects on escape (that is, either promoting or sensitizing) at many ACE2-proximal RBD sites, different mutations at the same non-RBD or ACE2-distal

RBD site can have opposing effects on neutralization (Fig. 5a–c). Furthermore, there is a strong correlation between mutational effects on neutralization and ACE2 binding at these sites: mutations that reduce neutralization also reduce ACE2 binding, and mutations that increase neutralization also increase ACE2 binding (Fig. 5a,b). No such consistent correlation exists between neutralization and

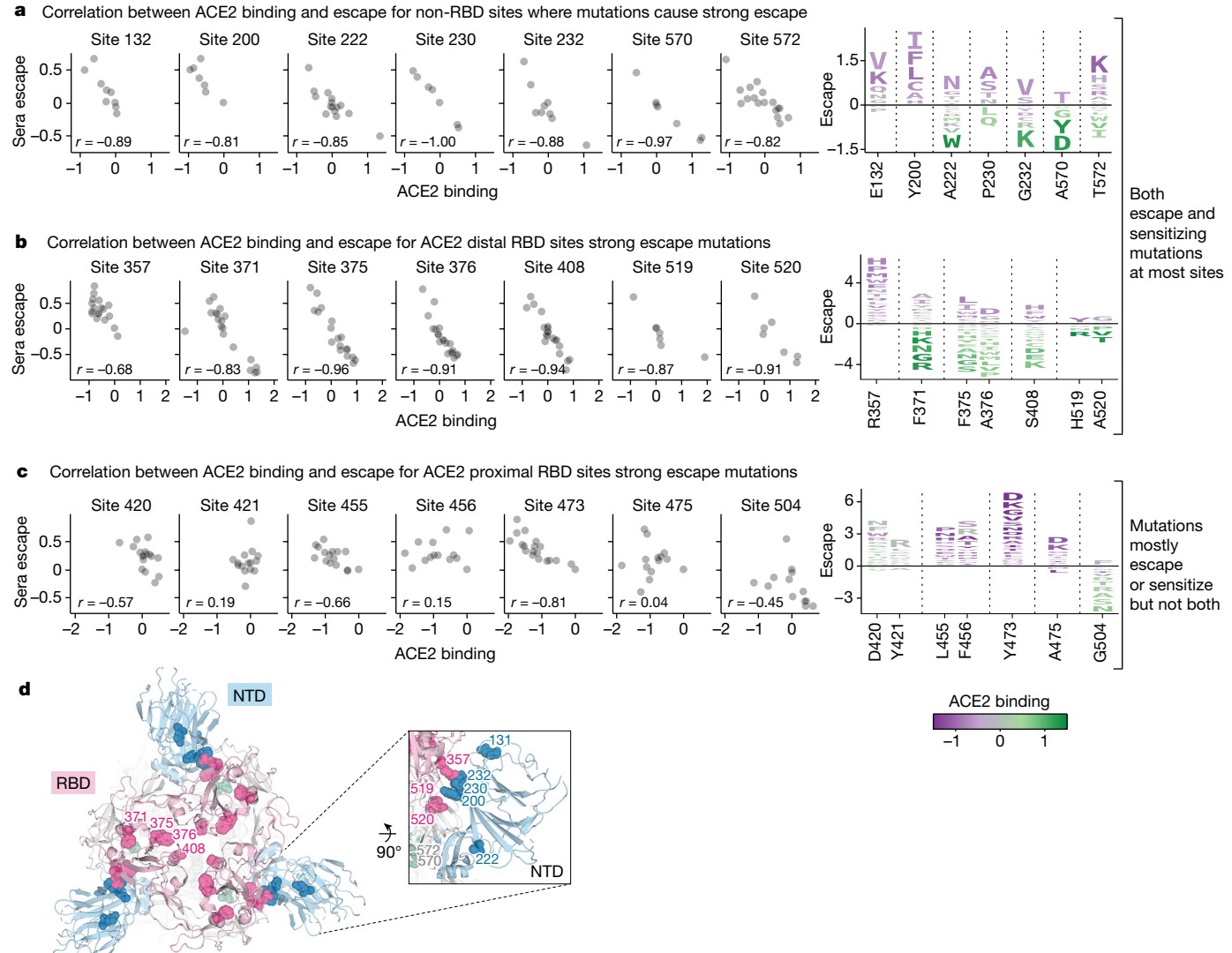

**Fig. 5 | Sera escape and ACE2 binding are inversely correlated for non-RBD and ACE2-distal RBD sites. a**, The left shows a correlation between ACE2 binding and sera escape for amino-acid mutations at non-RBD sites with the highest mutation-level sera escape (each point is a distinct amino-acid mutation). Average escape for each mutation across all sera is shown. The right shows a logo plot for the same sites, with letter heights proportional to escape caused by that mutation (negative heights mean more neutralization), and letter colours indicating effect on ACE2 binding (green means better binding). **b**, A similar plot for RBD sites that are distal (at least 15 Å) from ACE2. **c**, A similar plot for RBD sites proximal (within 15 Å) to ACE2. Only sites with at least seven different mutations measured are included in the logo plots. **d**, Top-down view of XBB spike (Protein Data Bank ID 8IOT) with the non-RBD and ACE2-distal sites shown in **a** and **b** highlighted as spheres. The RBD is pink, the NTD is blue and sites in SD1 are green.

ACE2 binding for RBD escape sites in close proximity of ACE2 binding interface (Fig. 5c).

We propose that non-RBD and ACE2-distal RBD mutations that increase both neutralization and ACE2 binding do so by shifting the RBD to a more upright position, whereas those that decrease neutralization and ACE2 binding do so by shifting the RBD to a more downwards position[36–38]. Previous work has shown that mutations that put the RBD in a down position reduce neutralization by antibodies that target RBD residues only accessible in the up position, whereas antibodies that can bind both the up and down RBD are unaffected by such mutations[15,39]. Consistent with this previous work, we confirmed that the mutations at ACE2-distal sites identified in our full-spike deep mutational scanning as probably affecting RBD conformation only affect neutralization by monoclonal antibodies that bind only to the up conformation of the RBD (Extended Data Fig. 8).

Our results show that mutations that affect neutralization and ACE2 binding by modulating RBD conformation are common in certain regions of spike: a result that makes structural sense, because most of these mutations are located near the interfaces between the RBD and other spike domains (Fig. 5d and Extended Data Fig. 9). Furthermore, many of these strong escape sites, including N234, F371, P373, F375, A376, S408, A570 and T572, have been previously shown by structural methods to affect RBD conformation[22,36–38,40–43] or the conformation of key RBD epitopes[19,44].

## Spike phenotypes and clade growth

SARS-CoV-2 evolution in humans is characterized by the repeated emergence of new viral clades, which often possess extra amino-acid mutations in spike relative to their predecessors. To test whether our deep mutational scanning measurements could help explain which clades are evolutionarily successful, we estimated the relative growth rates in humans of sufficiently-sampled SARS-CoV-2 clades using multinomial logistic regression[45] (Extended Data Fig. 10a–c). As expected,

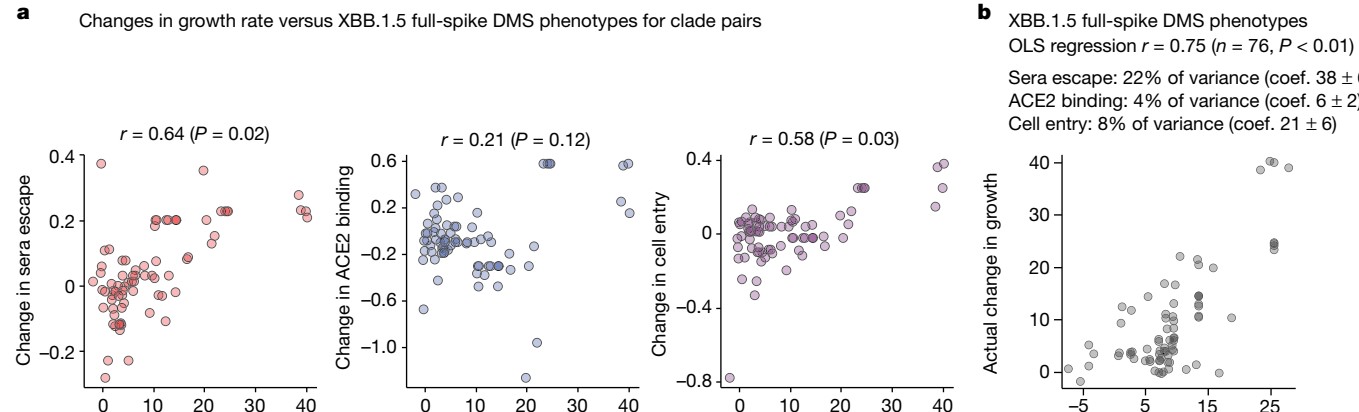

**Fig. 6 | Spike phenotypes measured by deep mutational scanning partially predict the evolutionary success of SARS-CoV-2 clades. a**, Correlation between the changes in growth rate for parent–descendant clade pairs versus the change in each spike phenotype measured in the XBB.1.5 full-spike deep mutational scanning (several mutations are assumed to have additive effects). The text above each plot shows the Pearson correlation ($r$) and a $P$ value computed by comparing the actual correlation to that for 100 randomizations of the experimental data among mutations. **b**, Ordinary least-squares multiple linear regression of changes in growth rate versus all three measured spike phenotypes. The small text indicates the unique variance explained by each

variable, as well as the coefficients (coef.) in the regression. See https://dms-vep.github.io/SARS-CoV-2_XBB.1.5_spike_DMS/htmls/current_dms_clade_pair_growth.html and https://dms-vep.github.io/SARS-CoV-2_XBB.1.5_spike_DMS/htmls/current_dms_ols_clade_pair_growth.html for interactive versions of both panels in which points can be hovered over for details on clades and their mutations. $P$ values are for one-sided tests of the hypothesis that the tested predictor outperforms randomizations, and are reported individually for each comparison. See Extended Data Fig. 12 for a similar analysis that also includes BA.2 and BA.5 descended clades.

more recent clades generally had higher growth rates, consistent with evolution selecting for viral clades that are more fit (Extended Data Fig. 10a), presumably in part due to further mutations in spike[46].

We sought to determine whether the growth of clades could be predicted from how their mutations affect the spike phenotypes measured by deep mutational scanning. Note that almost any mutation-based measurement (such as just counting mutations) trivially correlates with clade growth because newer clades typically have both better growth rates and more spike mutations (Extended Data Fig. 10a,d). For instance, clade growth rates strongly correlate with the number of spike mutations relative to the early Wuhan-Hu-1 sequence (Extended Data Fig. 10e). But this correlation is not informative because the question of evolutionary interest is not whether SARS-CoV-2's spike will acquire more mutations over time (we already know this will happen), but rather which of the various mutant viruses present at any given time will spread. Furthermore, phylogenetic correlations can exaggerate associations between mutations and clade growth[47]. Therefore, we focused on predicting changes in clade growth for each pair of parent–descendant clades separated by at least one spike mutation (Extended Data Fig. 10b). This approach eliminates the confounding effects of phylogenetic relatedness and the accumulation of mutations over time (Extended Data Fig. 10e,f), and better answers the question of how specific mutations affect clade growth.

Changes in growth between parent–descendant clade pairs were positively correlated with all three experimentally measured spike phenotypes both among just XBB-descended clades (Fig. 6a and Extended Data Fig. 11) and among all BA.2, BA.5 and XBB-descended clades (Extended Data Fig. 12). The correlations were statistically significant for sera escape and cell entry as assessed by randomization of the measurements among mutations. However, these univariate correlations do not fully capture the information in the experiments, as the effects of mutations on the spike phenotypes are themselves correlated (for example, mutations that cause sera escape sometimes decrease ACE2 binding). We therefore performed ordinary least-squares multiple linear regression of changes in clade growth versus all three phenotypes. The predictions of this regression correlated with changes in clade growth better than any individual phenotype, and were highly

statistically significant as assessed by randomization of the measurements among mutations (Fig. 6b and Extended Data Fig. 12). Sera escape uniquely explained the largest fraction of the variance in changes in clade growth, but ACE2 binding and cell entry effects also explained some variance. By contrast, neither RBD yeast-display deep mutational scanning of antibody escape[8,48] and ACE2 affinity[20] nor the EVEscape deep learning model[49] were consistently better than randomized data at predicting changes in clade growth at a significance level of $P = 0.05$ (Extended Data Figs. 11 and 12).

We also sought to test the ability of full-spike deep mutational scanning to explain the high fitness of BA.2.86 and its descendant clades (for example, JN.1), which were identified after the completion of the experiments described in this study[50]. Because there are not yet sufficient distinct BA.2.86-descended clades to make meaningful comparisons with clade growth, instead we performed a different test inspired by Thadani et al.[49]: we generated sequences with random sets of naturally observed spike amino-acid mutations that had the same number of differences relative to BA.2 as did BA.2.86, or relative to BA.2.86 as all designated BA.2.86-descended clades. Our XBB.1.5-based full-spike deep mutational scanning could distinguish the true BA.2.86 and BA.2.86-descended clades from sequences with the same number of mutations with high statistical significance, and did so better than RBD yeast-display deep mutational scanning or EVEscape (Supplementary Fig. 4).

## Discussion

More than 16 million human SARS-CoV-2 genomes have been sequenced to date, enabling rapid identification of variants with new mutations at the sequence level. However, interpreting the consequences of these mutations on viral spread in a partially immune population remains a major challenge. Here we show how pseudovirus-based deep mutational scanning can characterize the effects of mutations throughout spike on three distinct phenotypes critical to viral fitness: cell entry, ACE2 binding and serum antibody escape.

The full-spike pseudovirus data we generate enables several key insights that were not apparent from previous yeast-display RBD

deep mutational scanning approaches[1,2,48]. Most obviously, the data encompass all spike domains, not just the RBD. These data show that non-RBD mutations can affect ACE2 binding, probably by altering the conformation of the RBD in the context of the spike trimer (for example, in up versus down position). Such mutations are highly relevant for SARS-CoV-2 evolution—for instance, enhancement of ACE2 binding by non-RBD mutations appears to explain why EG.5.1 spread so rapidly after it acquired Q52H, why A222V subvariants of Delta spread widely, why A570D was selected in Alpha, and why T572I is now arising so frequently in BA.2.86-descended variants.

Pseudovirus deep mutational scanning also enables us to directly measure how mutations affect neutralization by polyclonal sera. By contrast, previous RBD-display deep mutational scanning could only measure how mutations affect antibody binding[51], and so to estimate mutational effects on serum neutralization escape it was necessary to characterize hundreds of individual antibodies assumed to represent the polyclonal neutralizing repertoire of humans[1,8]. The ability to directly map how mutations affect serum neutralization leads to two new insights. First, it reveals the heterogeneity in how mutations affect neutralization by sera from different individuals. For instance, we characterize sera from XBB* infected individuals that are both strongly affected and almost completely unaffected by mutations at key sites such as 456 or 473. The sera examined in this study came from individuals with varied immunization and infection histories, which probably contributes to observed escape heterogeneity, although individual-to-individual variation in humoral response may also play a role. This person-to-person heterogeneity in the antigenic effects of spike mutations will increase as individuals accumulate increasingly distinct exposure histories, and could come to play an important role in shaping SARS-CoV-2 evolution and disease susceptibility as it does for influenza virus[52–54].

The second major insight from direct mapping of serum escape is that mutations outside the RBD can have marked effects on neutralization. For instance, NTD mutations such as Y42F and N234T decrease neutralization by some sera by nearly sixfold. The existence of such strong non-RBD escape mutations may seem surprising given that most neutralizing activity in human sera come from antibodies that bind the RBD[9,32,51,55]. However, our data indicate that the strongest non-RBD serum escape mutations act primarily by shifting the RBD to the down conformation, thereby indirectly escaping class 1 and 4 antibodies that bind to RBD surfaces only accessible in the up conformation[15,39]. Of course, such mutations come at a cost to ACE2 binding, because the RBD cannot bind receptor in the down conformation[56,57]. Nonetheless, the ubiquity of such mutations suggests that this mechanism of escape merits monitoring and is in line with previous observations made with endemic human coronaviruses[58–60]. For instance, the RBD of the CoV-229E spike has never been observed in the up conformation[61,62] despite the fact that this spike somehow manages to bind its receptor during infection. Whether SARS-CoV-2's spike could eventually evolve to also far more strongly favour a down RBD conformation is unknown.

The most important indication of the relevance of our work is that our measurements of spike phenotypes partially explain the evolutionary success of different SARS-CoV-2 clades in humans. A longstanding goal of evolutionary biology is to understand the molecular phenotypes that contribute to fitness[63], and then measure them with sufficient accuracy to predict which mutants will actually spread in the real world. We have taken a real step towards this goal, because the spike phenotypes measured by our deep mutational scanning explain a substantial amount of the changes in growth rates of recent SARS-CoV-2 clades. Of course, pseudovirus spike deep mutational scanning will never perfectly predict SARS-CoV-2 evolution: evolution itself is partially stochastic[64], pseudovirus experiments do not capture all phenotypes of spike relevant to transmission or multicycle replication and our experiments completely ignore mutations to non-spike genes that contribute to

fitness[14,65]. Furthermore, it remains technically challenging for deep mutational scanning to account for epistatic interactions among mutations[66], and we need modelling approaches that better account for how person-to-person heterogeneity in immune-escape mutations shape viral evolution[52]. However, the fact that our deep mutational scanning has substantial power to explain clade growth shows that we have reached the point at which experiments can enable useful predictions about SARS-CoV-2 evolution. An important area of future work will be integrating these highly informative experimental measurements into even more sophisticated models of viral evolution[49,67,68].

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

## Reporting summary

Further information on research design is available in the Nature Portfolio Reporting Summary linked to this article.

## Data availability

The data described in this paper are available in both interactive and numerical form in various levels of detail. For easy interactive visualization of the data, we suggest the following interactive charts of how mutations affect all measured phenotypes after applying a reasonable set of filters to remove lower confidence measurements: XBB.1.5 spike, https://dms-vep.github.io/SARS-CoV-2_XBB.1.5_spike_DMS/htmls/summary_overlaid.html; BA.2 spike, https://dms-vep.github.io/SARS-CoV-2_Omicron_BA.2_spike_ACE2_binding/htmls/summary_overlaid.html and XBB.1.5 RBD, https://dms-vep.github.io/SARS-CoV-2_XBB.1.5_RBD_DMS/htmls/summary_overlaid.html. For numerical data on mutational effects on all measured phenotypes after applying the same reasonable set of filters, see XBB.1.5 spike, https://github.com/dms-vep/SARS-CoV-2_XBB.1.5_spike_DMS/blob/main/results/summaries/summary.csv; XBB.1.5, spike, per-serum escape: https://github.com/dms-vep/SARS-CoV-2_XBB.1.5_spike_DMS/blob/main/results/summaries/per_antibody_escape.csv; BA.2 spike, https://github.com/dms-vep/SARS-CoV-2_Omicron_BA.2_spike_ACE2_binding/blob/main/results/summaries/summary.csv and XBB.1.5 RBD, https://github.com/dms-vep/SARS-CoV-2_XBB.1.5_RBD_DMS/blob/main/results/summaries/summary.csv. Raw sequencing data files have been uploaded to BioProjects under the following accession codes: PRJNA1034580 for the XBB.1.5 full-spike library, PRJNA1035795 for the XBB.1.5 RBD-only library and PRJNA1035933 for the BA.2 full-spike library.

## Code availability

In addition to the above interactive charts and numerical data, the entire computational pipelines are available along with rich interactive HTML displays of results. These numerical data and HTML displays include extra options to filter the data for higher and lower confidence values, such as by examining the measurements in each of the two replicate libraries or filtering measurements by how many variants a mutation is seen in. Specifically, full interactive HTML documentation for each deep mutational scanning experiment are rendered on GitHub Pages as follows: XBB.1.5 full spike, https://dms-vep.github.io/SARS-CoV-2_XBB.1.5_spike_DMS/; BA.2 full spike, https://dms-vep.github.io/SARS-CoV-2_Omicron_BA.2_spike_ACE2_binding/ and XBB.1.5 RBD, https://dms-vep.github.io/SARS-CoV-2_XBB.1.5_RBD_DMS/. GitHub repositories with the actual computer code, as well as numerical data are at: XBB.1.5 spike, https://github.com/dms-vep/SARS-CoV-2_XBB.1.5_spike_DMS; BA.2 spike, https://github.com/dms-vep/SARS-CoV-2_Omicron_BA.2_spike_ACE2_binding and XBB.1.5 RBD, https://github.com/dms-vep/SARS-CoV-2_XBB.1.5_RBD_DMS. XBB.1.5 full spike, XBB.1.5 RBD spike and BA.2 full spike repositories are published via Zenodo at https://doi.org/10.5281/zenodo.10981249 (ref. 69), https://doi.org/10.5281/zenodo.10981257 (ref. 70) and https://doi.org/10.5281/zenodo.10981262 (ref. 71), respectively. Note that most of the analysis in these GitHub repositories is performed using dms-vep-pipeline-3 (https://github.com/dms-vep/dms-vep-pipeline-3), v.3.5.3. Python notebooks and raw event data used for mass photometry analysis are available at https://github.com/JackTaylorBrown/massphotometry.

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

**Acknowledgements** We thank R. Hisner for helpful comments during library design. We thank C. Roemer and R. Neher for helpful comments on interpreting the effects of mutations in the context of SARS-CoV-2 evolution. We thank T. Peacock for useful discussions on ACE2 binding. We gratefully acknowledge all data contributors, including the authors and their originating laboratories responsible for obtaining the specimens, and their submitting laboratories that generated the genetic sequence and metadata and shared via the GISAID Initiative, the data on which part of this research is based. J.D.B. is an investigator at the Howard Hughes Medical Institute. D.V. is an Investigator of the Howard Hughes Medical Institute and the Hans Neurath Endowed Chair in Biochemistry at the University of Washington. This work was supported by the grant no. NIH/NIAID R01AI141707 and contract no. 75N93021C00015 to J.D.B., as well as grant nos. NIH/NIAID P01AI167966 and DP1AI158186 and contract no. 75N93022C00036 to D.V.), a Pew Biomedical Scholars Award (D.V.), an Investigators in the Pathogenesis of Infectious Disease Awards from the Burroughs Wellcome Fund (D.V.), the University of Washington Arnold and Mabel Beckman cryoEM centre and the National Institute of Health grant no. S10OD032290 (to D.V.). This research was also supported by the Genomics & Bioinformatics Shared Resource, RRID:SCR_022606, of the Fred Hutchinson Cancer Center/University of Washington Cancer Consortium (grant no. P30 CA015704), by the Flow Cytometry Shared Resource, RRID:SCR_022613, of the Fred Hutchinson Cancer Center/University of Washington/Seattle Children's Cancer Consortium (grant no. P30 CA015704), and by Fred Hutchinson Cancer Center Scientific Computing, NIH grant nos. S10-OD-020069 and S10-OD-028685. B.M. was funded by SciLifeLab's Pandemic Laboratory Preparedness programme (grant no. VC-2022-0028) and the Erling Persson Foundation (grant no. 2021 0125). T.B. is an investigator at the Howard Hughes Medical Institute.

**Author contributions** Conceptualization was the responsibility of B.D. and J.D.B. Methodology was developed by B.D., J.B., B.M., D.V. and J.D.B. Experiments were carried out by B.D., J.B., T.E.M., J.L., A.G.F., D.A. and C.S. Computational analysis was done by B.D., J.B., B.M., D.A., D.V., T.B., M.D.F. and J.B.D. Writing of the original draft was done by B.D. and J.D.B. Writing, review and editing were carried out by B.D., D.V. and J.D.B. Resources came from J.L. and H.Y.C. Supervision of the project was by J.D.B. Funding was acquired by J.D.B.

**Competing interests** J.D.B. and B.D. are inventors on Fred Hutchinson Cancer Center licensed patents related to the pseudovirus deep mutational scanning system used in this paper. J.D.B. consults for Apriori Bio, Invivyd, Aerium Therapeutics, GlaxoSmithKline and the Vaccine Company on topics related to viral evolution. H.Y.C. reports consulting with Ellume, Pfizer and the Bill and Melinda Gates Foundation. She has served on advisory boards for Vir, Merck and Abbvie. She has conducted continuing medical education teaching with Medscape, Vindico and Clinical Care Options. She has received research funding from Gates Ventures, and support and reagents from Ellume and Cepheid, all outside the submitted work. D.V. is named as inventor on patents for coronavirus vaccines filed by the University of Washington. The other authors declare no competing interests.

**Additional information**
**Correspondence and requests for materials** should be addressed to Jesse D. Bloom.

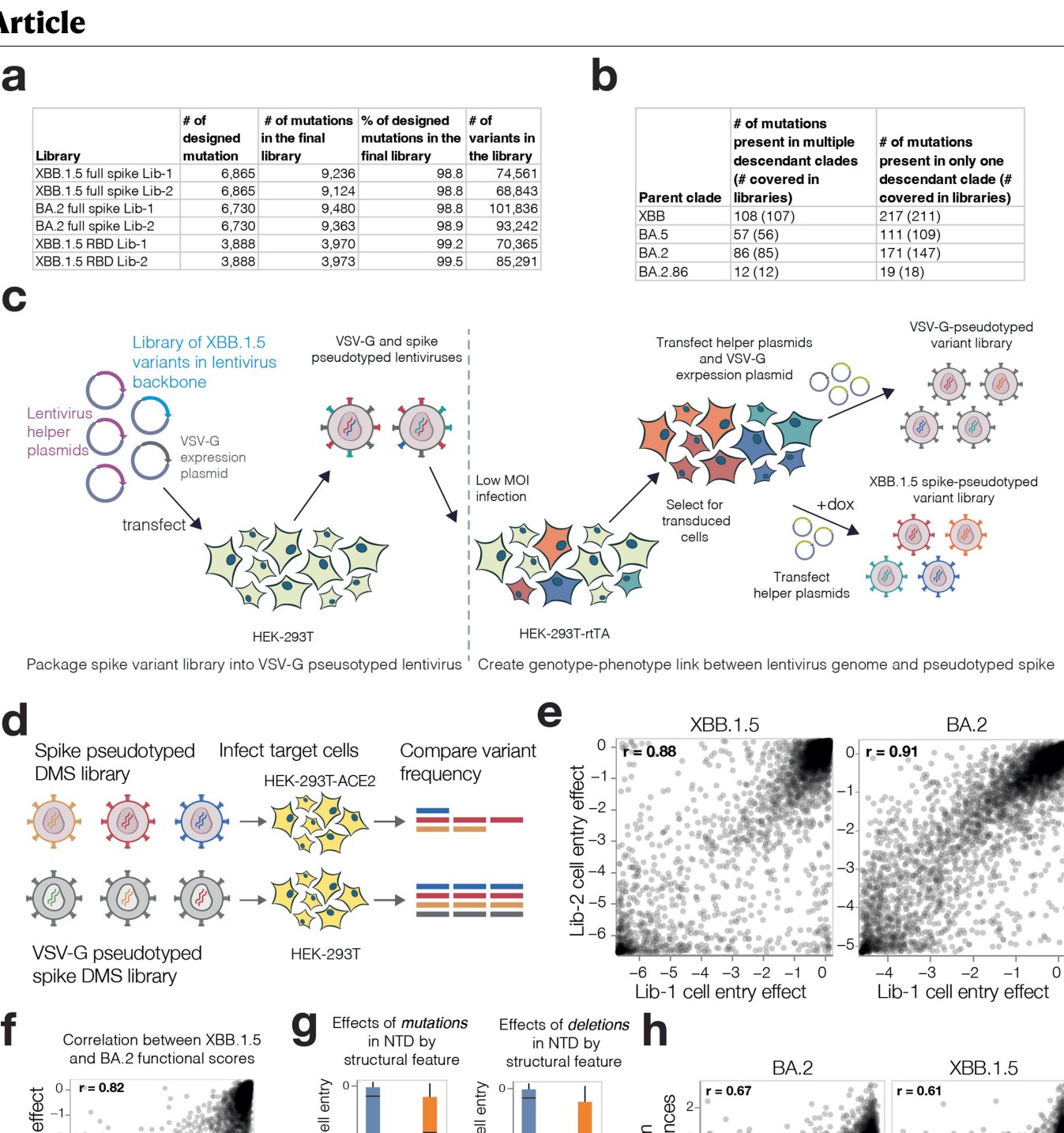

**Extended Data Fig. 1** | See next page for caption.

**Extended Data Fig. 1 | XBB.1.5 and BA.2 spike deep mutational scanning libraries. a**, Number of targeted and final number of mutations and barcoded variants in the XBB.1.5 and BA.2 full spike and XBB.1.5 RBD pseudovirus-based deep mutational scanning libraries. **b**, Total number of unique spike amino-acid mutations present in BA.2, BA.5, BA.2.86, and XBB descended Pango clades and the number of those mutations that are present in at least three barcoded variants in each replicate of the XBB.1.5 full spike libraries, which was the minimum number of occurrences we needed to make high-confidence estimates of the mutational effects on cell entry. The first number is the total number of mutations meeting the criteria and the number in parentheses is the number of these mutations covered in the libraries: for example, there are 108 spike amino-acid mutations that occur in more than one XBB-descended clade, and 107 of those mutations are well covered in our XBB.1.5 full spike libraries. **c**, Method for creating genotype-phenotype linked spike deep mutational scanning libraries, as previously described in Dadonaite et al.[4]. Lentivirus backbone plasmids encoding barcoded mutagenised spike genes together with helper and VSV-G expression plasmids are transfected into 293 T cells to make VSV-G pseudotyped virus. These viruses are used to infect 293T-rtTA cells at MOI < 0.01 so that no more than one spike variant is integrated into each cell. Transduced cells are selected for lentiviral integration, and spike pseudotyped virus libraries are produced from these cells by transfecting helper plasmids in the presence of doxycycline to induce spike expression. In the absence of doxycycline and with added VSV-G expression plasmid, VSV-G pseudotyped virus libraries are also produced from the same cell lines; these VSV-G pseudotyped viruses are used to help estimate effects of spike mutations on cell entry as described in the next panel. **d**, Method used to measure effects of mutations in spike on cell entry. The ability of each spike variant to mediate cell entry is assessed by quantifying its relative frequency in 293T-ACE2 cells infected with spike-pseudotyped versus VSV-G pseudotyped libraries. **e**, Correlations between the effects of mutations on cell entry measured using each of the two independent full spike libraries of XBB.1.5 or BA.2. Throughout the rest of this paper, we report the mean value between the two libraries. **f**, Correlation between mutational effects on cell entry measured for the XBB.1.5 versus BA.2 full spike libraries. **g**, Cell-entry effects as measured in the deep mutational scanning of mutations in either the flexible loops or core β-sheets of the NTD. The left plot shows the effects of amino-acid mutations; the right plot shows the effects of single-residue deletions. The black line indicates the median entry effect, and the boxes indicate the interquartile range. Mutational effects are the median of two biological replicates. Whiskers indicate 0.75 of the interquartile range plotted from the smallest value of the 1st and highest value of the 3rd quartile. **h**, Correlation between mutational effects measured with the XBB.1.5 or BA.2 full spike libraries and fitness effects of those mutations estimated from actual human SARS-CoV-2 sequences[14].

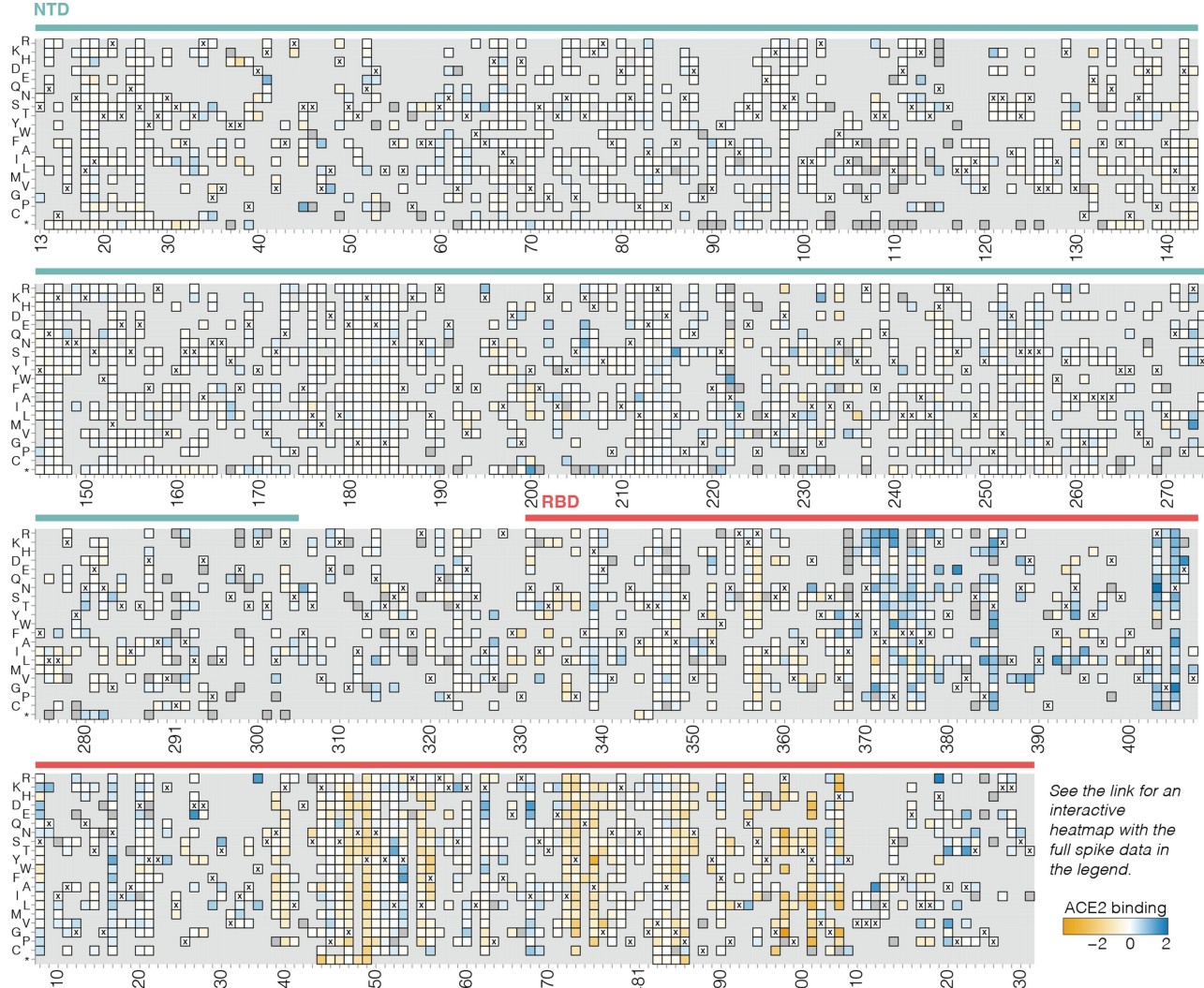

See the link for an interactive heatmap with the full spike data in the legend.

ACE2 binding
−2 0 2

**Extended Data Fig. 2 | Correlations among measured mutational effects on ACE2 binding. a**, Correlation between effects of mutations on ACE2 binding measured with XBB.1.5 full spike and XBB.1.5 RBD pseudovirus libraries. **b**, Correlation between effects of mutations on ACE2 binding measured using XBB.1.5 RBD pseudovirus library with monomeric and dimeric ACE2. Heatmaps with the XBB.1.5 RBD pseudovirus measurements made using monomeric and dimeric ACE2 are at https://dms-vep.github.io/SARS-CoV-2_XBB.1.5_RBD_DMS/htmls/monomeric_ACE2_mut_effect.html and https://dms-vep.github.io/SARS-CoV-2_XBB.1.5_RBD_DMS/htmls/dimeric_ACE2_mut_effect.html,

respectively **c**, Correlation between effects of mutations on ACE2 binding and spike-mediated cell entry for different libraries. **d**, ACE2 binding heat map showing key non-RBD sites that have mutated in the past major SARS-CoV-2 variants. Specific variant mutations are highlighted in red outline. Table on the right indicates variants in which these mutations occurred. See https://dms-vep.github.io/SARS-CoV-2_XBB.1.5_spike_DMS/htmls/monomeric_ACE2_mut_effect.html for an interactive plot showing ACE2 binding for all mutations measured in spike is at.

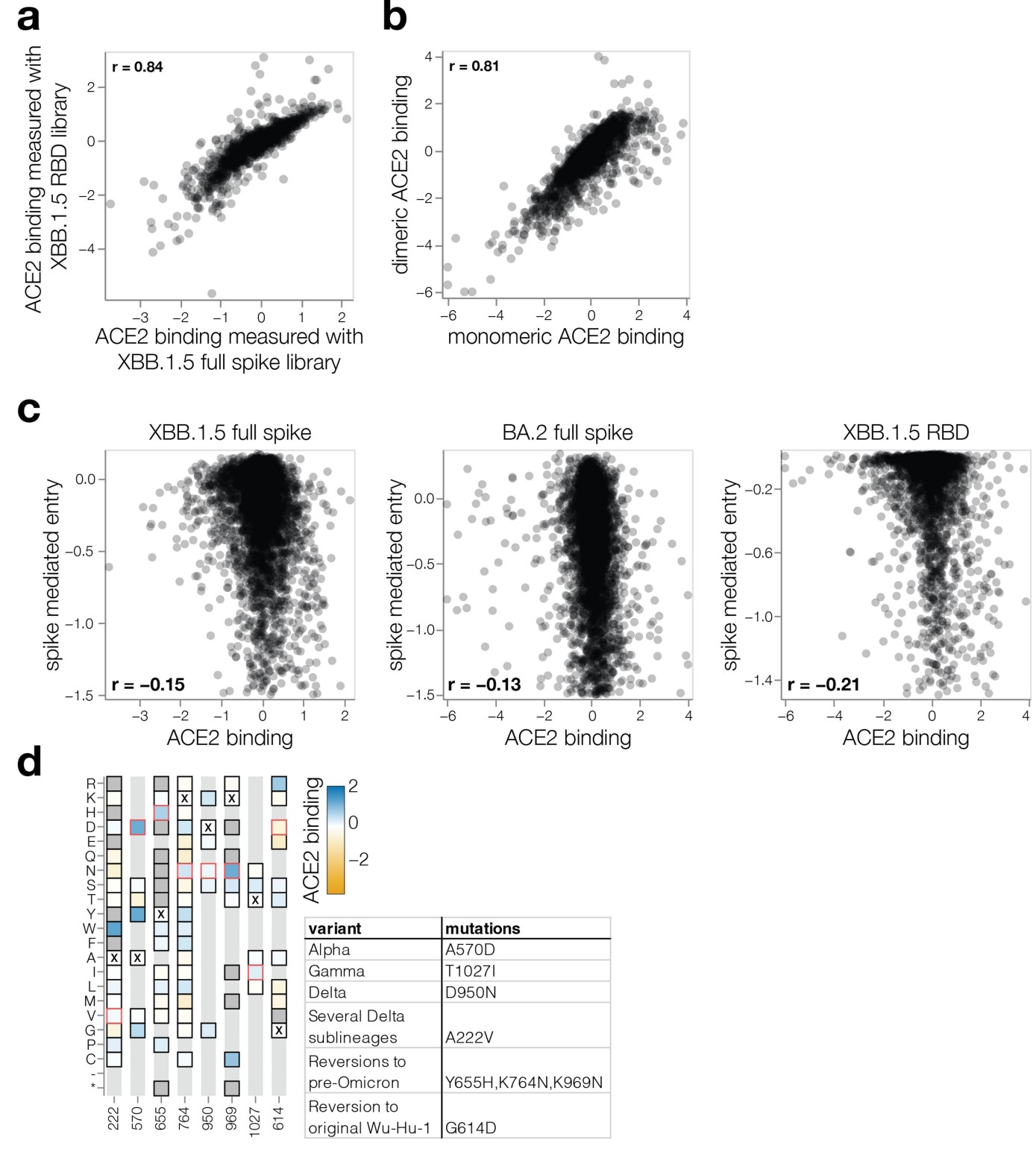

**Extended Data Fig. 3 | Effects of NTD and RBD mutations on full-spike ACE2 binding.** Mutations that enhance ACE2 binding are shaded blue, mutations that decrease affinity are shaded orange, mutations that are too deleterious for cell entry to be measured in the binding assay are dark gray, and light gray indicates mutations not present in our libraries. Interactive heatmaps showing mutational effects on ACE2 binding for the full XBB.1.5 and BA.2 spikes are at https://dms-vep.github.io/SARS-CoV-2_XBB.1.5_spike_DMS/htmls/monomeric_ACE2_mut_effect.html and https://dms-vep.org/SARS-CoV-2_Omicron_BA.2_spike_ACE2_binding/htmls/monomeric_ACE2_mut_effect.html. Note that a few sites are missing in the static heatmap in this figure due to lack of coverage or deletions in the XBB.1.5 spike; see the interactive heatmaps for per-site numbering.

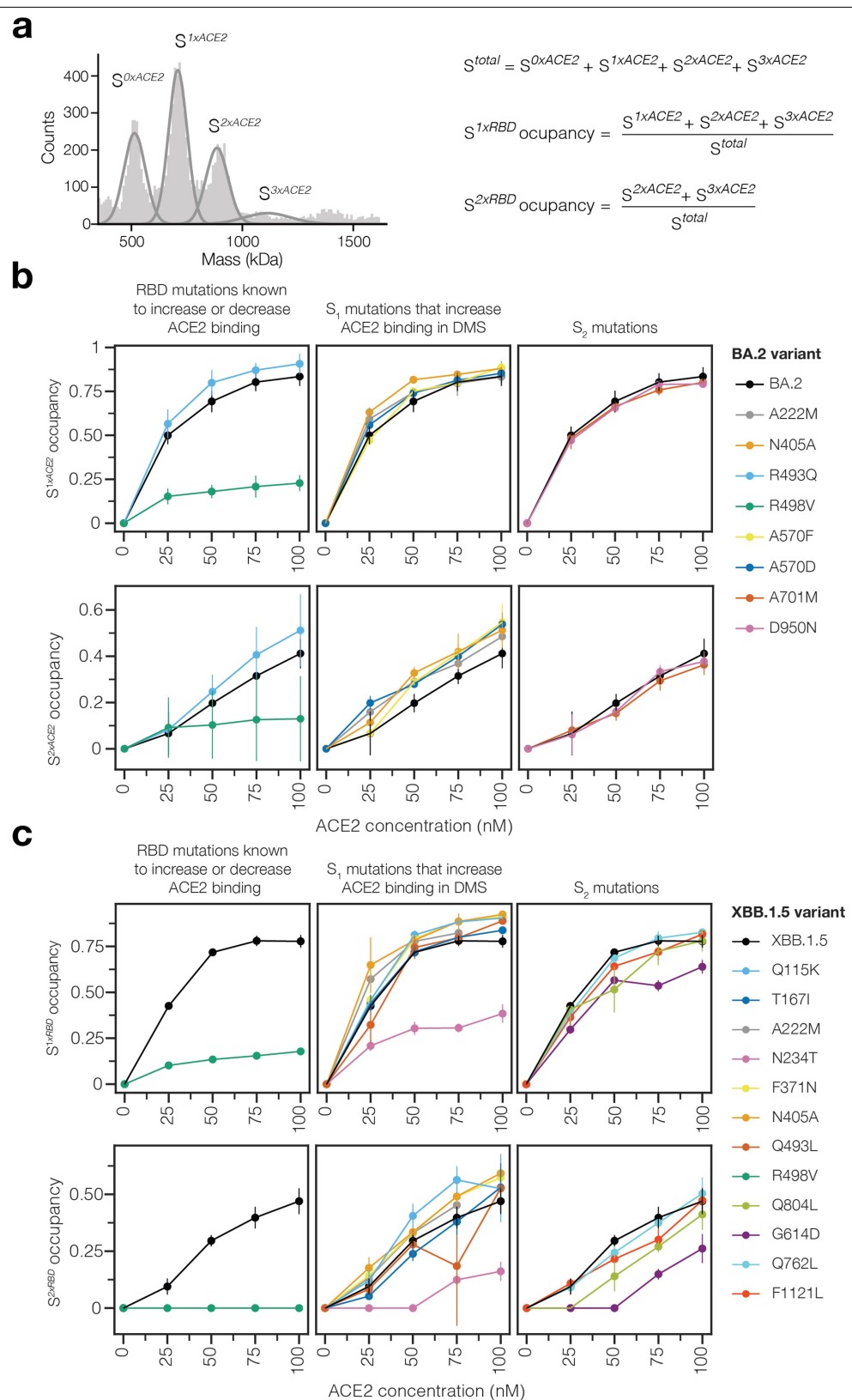

**Extended Data Fig. 4 | Mass photometry measurements for S₁ and S₂ occupancy. a**, Illustration of Gaussian components for no ($S^{0xACE2}$), one ($S^{1xACE2}$), two ($S^{2xACE2}$), or three ($S^{3xACE2}$) ACE2-bound spikes. $S^{1xRBD}$ occupancy is the fraction of spikes bound by one ACE2 molecule and $S^{2xRBD}$ occupancy is the fraction of spikes bound by two ACE2 molecules. **b**, Top row - $S^{1xRBD}$ occupancy measured using mass photometry for different BA.2 spike mutants. Bottom row - $S^{2xRBD}$ occupancy for different BA.2 spike mutants. **c**, Top row - $S^{1xRBD}$ occupancy for different XBB.1.5 spike mutants. Bottom row - $S^{2xRBD}$ occupancy for different XBB.1.5 spike mutants. Error bars in plots b-c indicate standard error between two biological replicates.

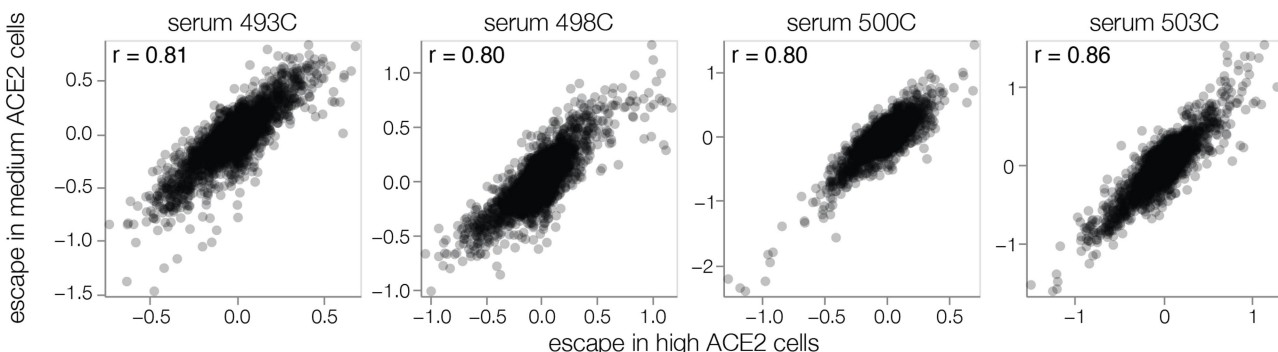

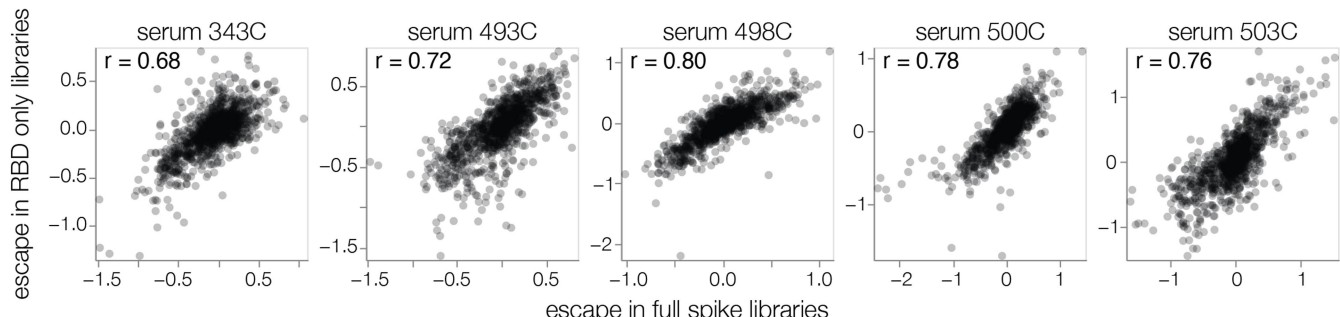

**Extended Data Fig. 5 | Correlation among serum escape mapping experiments. a**, Correlation between mutation escape scores for experiments using the full-spike XBB.1.5 libraries performed on 293 T cells expressing high or medium amounts of ACE2 for four sera. Note that the medium cells were used for all other figures shown in this paper. **b**, Correlation between mutation escape scores for mutations in the XBB.1.5 full spike and RBD-only libraries. See https://dms-vep.github.io/SARS-CoV-2_XBB.1.5_spike_DMS/htmls/compare_high_medium_ace2_escape.html and https://dms-vep.github.io/SARS-CoV-2_XBB.1.5_spike_DMS/htmls/compare_spike_rbd_escape.html for interactive versions of these scatter plots that also show line plots of per-site escape values for each serum.

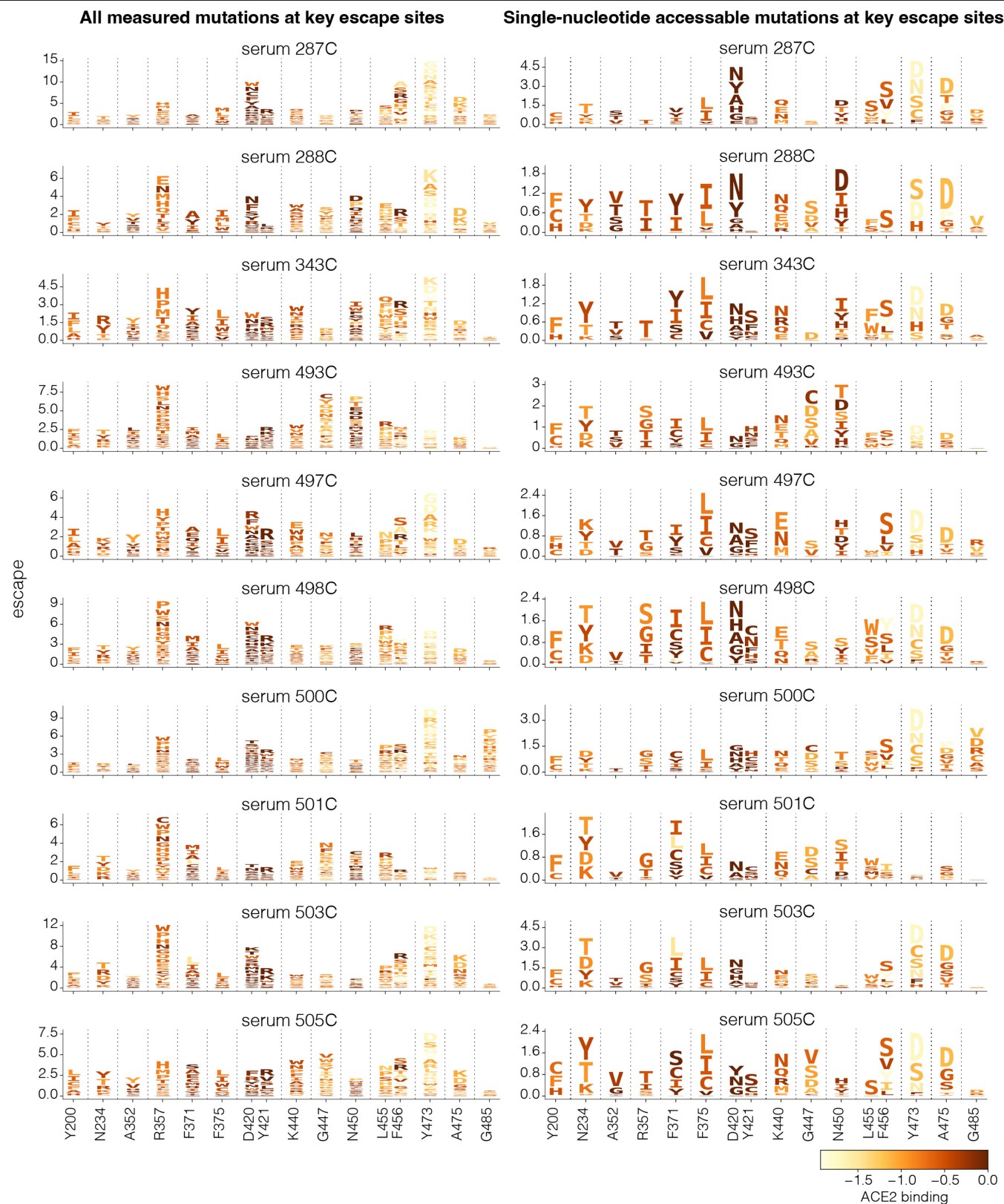

**Extended Data Fig. 6 | Escape at key sites for each serum.** Logoplots showing XBB.1.5 spike escape at 16 highest escape sites for each of the 10 sera measured. Letter heights indicate the escape caused by mutation to that amino acid. Letters are colored light yellow to dark brown depending on mutation effect on ACE2 binding. Left: all mutations measured. Right: mutations accessible with a single-nucleotide substitution.

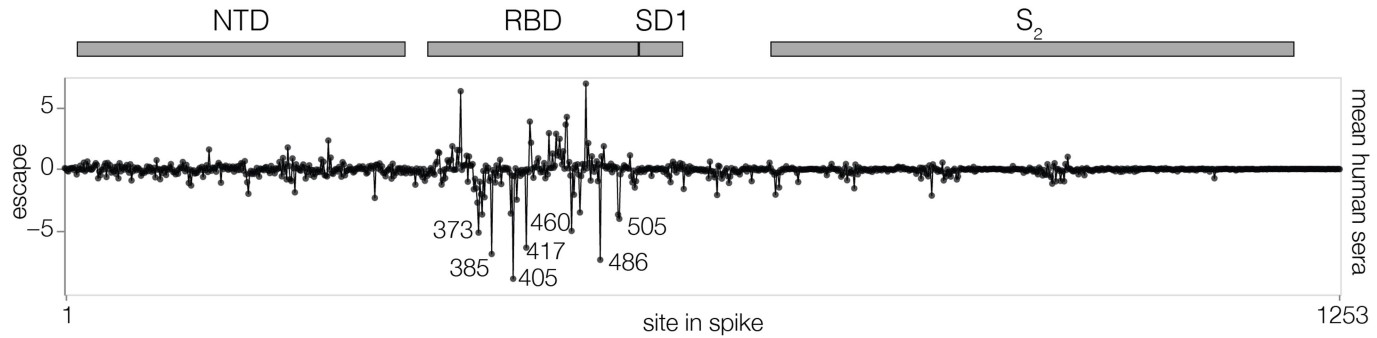

**Extended Data Fig. 7 | Mutations in XBB.1.5 spike that increase serum neutralization.** Escape at each site in the XBB.1.5 spike averaged across the 10 sera from individuals with prior XBB* infections, showing negative as well as positive values (Fig. 4 only shows positive values). Sites with negative escape in this plot are ones where many mutations make spike more sensitive to neutralization. Interactive plots with site and mutation-level escape are at https://dms-vep.github.io/SARS-CoV-2_XBB.1.5_spike_DMS/htmls/summary_overlaid.html (set 'floor escape at zero' at the bottom of the chart to false to show negative escape).

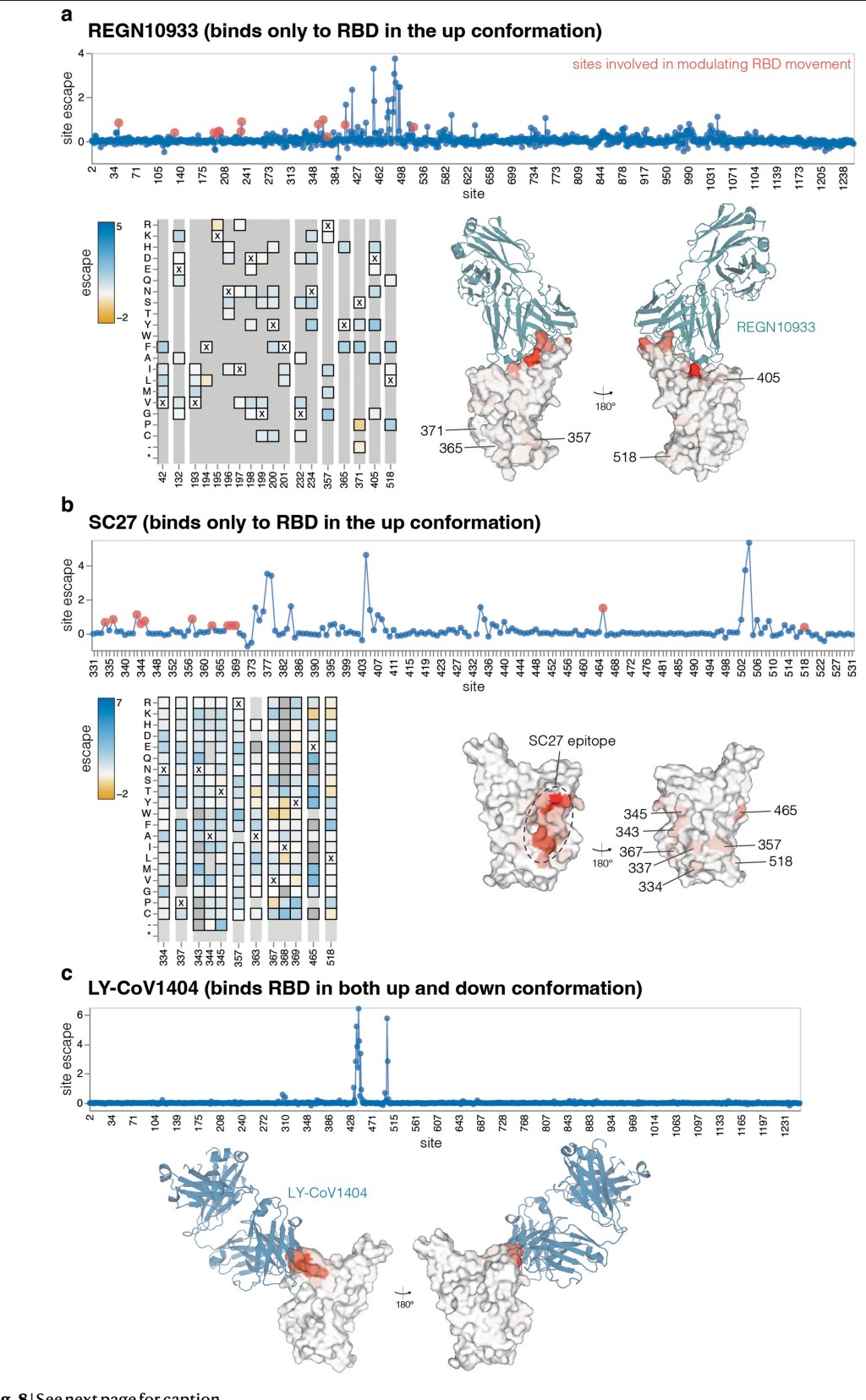

**a** REGN10933 (binds only to RBD in the up conformation)

**b** SC27 (binds only to RBD in the up conformation)

**c** LY-CoV1404 (binds RBD in both up and down conformation)

**Extended Data Fig. 8** | See next page for caption.

**Extended Data Fig. 8 | Only antibodies that bind RBD in the up conformation are escaped by mutations outside the structural epitope.** This figure shows previously generated and published deep mutational scanning escape maps for three monoclonal antibodies, two of which bind to RBD only in the up conformation (REGN10933 and SC27) and one of which binds to the RBD in both the up and down conformation (LY-CoV1404). All antibodies are escaped by mutations in their direct structural epitope, but only the antibodies that bind only the up conformation are escaped by ACE2-distal mutations outside their epitope that affect RBD up/down conformation. **a**, REGN10933 escape profile mapped using a Delta full spike deep mutational scanning library[4]. REGN10933 only binds RBD in the up position[72,73]. Line plot shows mean escape at each position in Delta spike with sites that modulate RBD movement highlighted in red. Heatmap shows mutation escape scores for sites highlighted in red on the line plot. Surface representation of RBD is coloured by site mean escape score with sites showing escape in the RBD outside the main antibody labeled (PDB ID: 6XDG). **b**, SC27 antibody escape profile mapped using the XBB.1.5 saturated RBD deep mutational scanning library[74]. SC27 only binds RBD in the up conformation. (PDB ID: 7MMO). **c**, LY-CoV1404 escape profile mapped using the BA.1 full spike deep mutational scanning library[4]. LY-CoV1404 binds RBD in both up and down conformations[75]. (PDB ID: 7MMO).

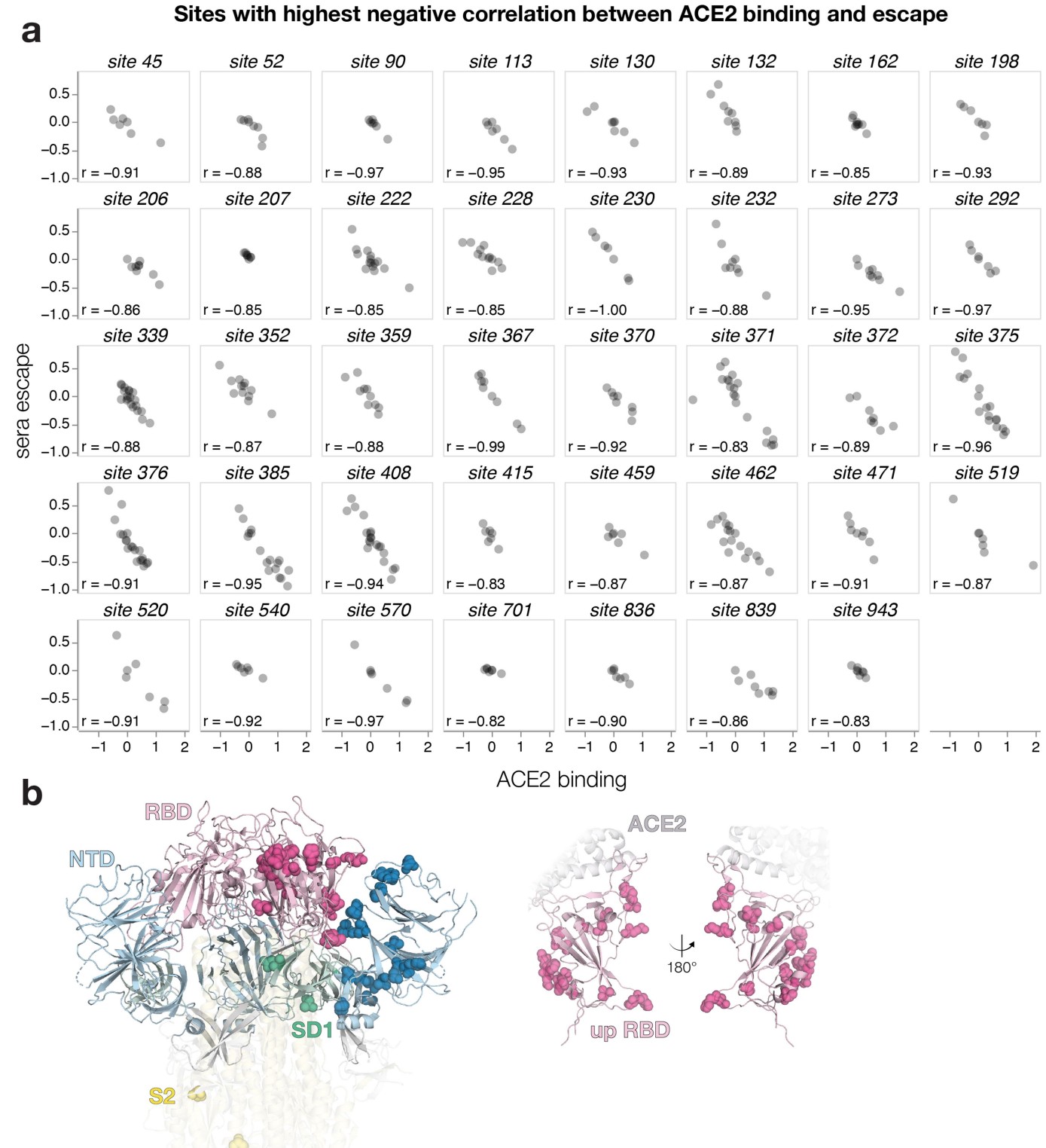

**a** Sites with highest negative correlation between ACE2 binding and escape

**Extended Data Fig. 9 | Sites with highest inverse correlation between ACE2 binding and serum escape. a**, Correlation between ACE2 binding and serum escape for sites in XBB.1.5 spike. Only sites with at least 7 mutations measured and Pearson r < 0.82 are shown. **b**, Most sites with strongly negative correlations between mutational effects on ACE2 binding and escape are at positions that could plausibly impact the RBD conformation in the context of the full spike,

since they tend to be at the interface of the RBD and other spike domains. Left: all sites from **a** shown on spike structure as spheres. RBD is colored in light pink, NTD light blue, SD1 green and the $S_2$ subunit in yellow. Spheres are shown on only one chain for each domain for clarity (PDB ID: 8IOU). Right: RBD sites from **a** shown on RBD in up position engaged with ACE2. RBD is colored in light pink and ACE2 is gray.

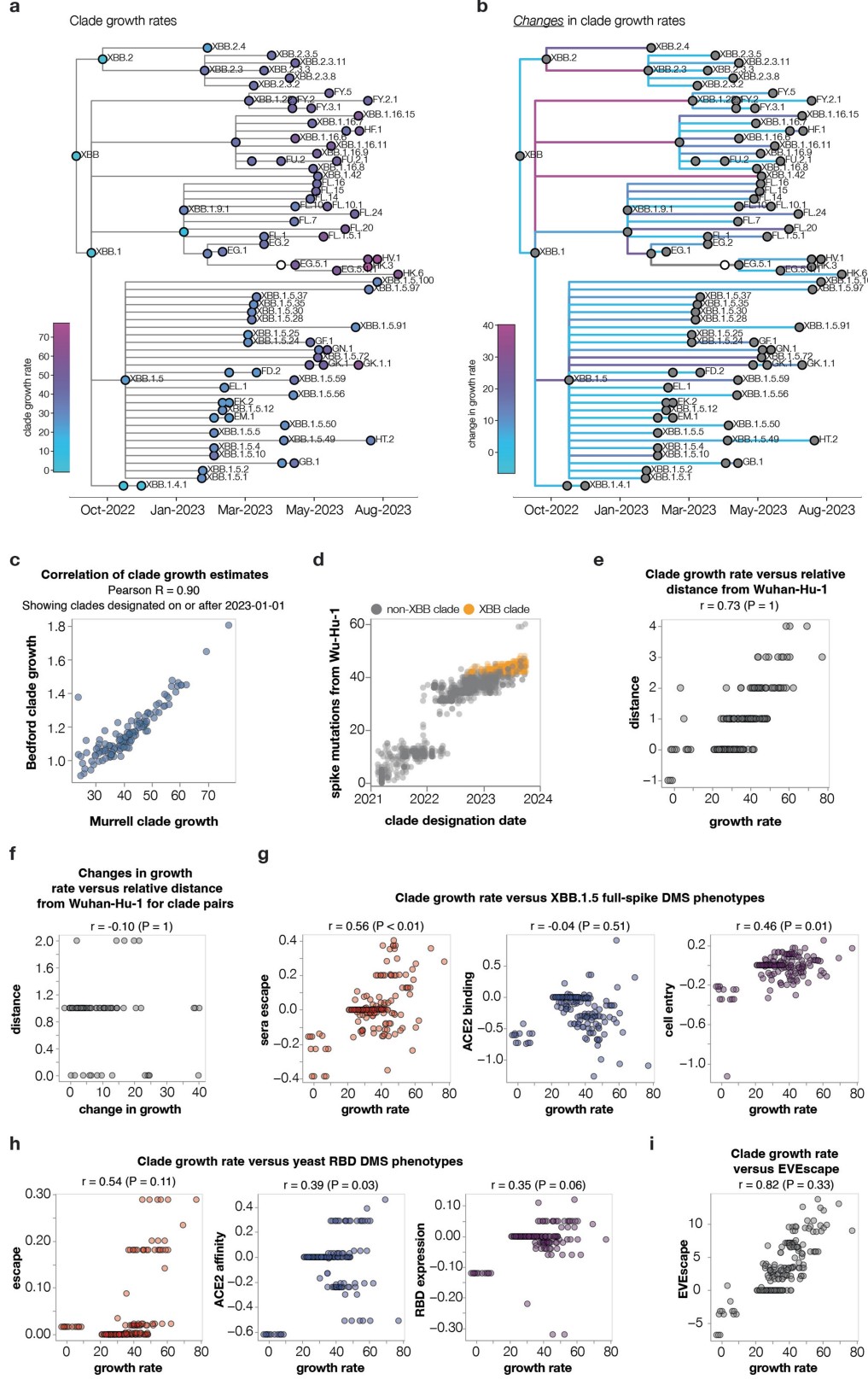

**Extended Data Fig. 10** | See next page for caption.

**Extended Data Fig. 10 | Correlations in absolute clade growth with absolute clade phenotypes. a**, Phylogenetic tree of XBB-descended Pango clades, colored by their relative growth rates. The tree shows only clades with at least 400 sequences and at least one new spike mutation, and their ancestors. Ancestor clades with insufficient sequences for growth rate estimates are in white. **b**, The same phylogeny but with branches colored by the change in growth rate between parent-descendant clade pairs. **c**, Correlation between clade growth estimates made using the Murrell lab multinomial logistic regression model (see methods) or a hierarchical multinomial logistic regression implemented by the Bedford lab[68] (see https://github.com/nextstrain/forecasts-ncov/). Both sets of estimates are for clades designated after Jan-1-2023 and use the data available as of Oct-2-2023. The estimates are highly correlated, and everywhere else in this paper we report analyses using the Murrell lab estimates. **d**, Number of spike amino-acid mutations relative to the early Wuhan-Hu-1 virus in all SARS-CoV-2 Pango clades versus the clade designation dates. XBB-descended clades are in orange. As can be seen from this plot, newer clades tend to have more spike mutations. **e**, Because newer clades tend to have both more mutations and better growth, clade growth rate is trivially correlated with a clade's relative distance (number of spike mutations) from Wuhan-Hu-1. However, this correlation is not informative as it is already known that new clades tend to have more mutations. **f**, If we instead correlate the change in growth rate between parent-descendant clade pairs

separated by at least one spike mutation (Fig. 6b) with the change in spike mutational distance to Wuhan-Hu-1 there is no correlation, since this approach removes the co-variation with total mutation count. Therefore, simple mutation counting is not informative for predicting changes in clade growth. **g**, Correlations for the phenotypes measured by the full spike deep mutational scanning in the current paper; **h**, the phenotypes measured in yeast display RBD deep mutational scanning; **i**, predicted by the EVEscape method. These plots differ from Fig. 6a and Extended Data Fig. 11 in that they show the correlations in absolute clade growth with the absolute clade phenotypes, rather than comparing the changes in both for each parent-descendant clade pair. Absolute clade phenotypes are computed as the sum of mutation effects. The P-values above the plots is a one sided test that computes the fraction of times the correlation is greater than that for the actual data after randomizing the phenotypic effects among mutations. Note that the correlations are not reflective of the P-values (there can be high correlations but non-significant P-values) for the reasons noted in the main text and in **e**—phylogenetic correlations, and the fact that new clades have both more mutations and higher growth so that any "phenotype" that amounts to counting mutations gives a correlation in these plots. For this reason, comparing changes in clade growth to changes in spike phenotypes as done in Fig. 6a and Extended Data Fig. 11 is the correct approach to test whether a method can actually predict which new clades will be successful.

**a** Changes in growth rate versus XBB.1.5 full-spike DMS phenotypes for clade pairs

r = 0.64 (P = 0.02)  r = 0.21 (P = 0.12)  r = 0.58 (P = 0.03)

**b** Changes in growth rate versus yeast RBD DMS phenotypes for clade pairs

r = 0.21 (P = 0.18)  r = 0.58 (P = 0.08)  r = 0.47 (P = 0.11)

**c** Changes in growth rate versus EVEscape for clade pairs

r = 0.56 (P = 0.2)

**d** yeast RBD DMS phenotypes
OLS regression r = 0.62 (n = 80, P = 0.37)
escape: 3% of variance (coef 27 ± 13)
ACE2 affinity: 14% of variance (coef 19 ± 5)
RBD expression: 1% of variance (coef 20 ± 20)

**f** XBB.1.5 full-spike DMS phenotypes
OLS regression r = 0.75 (n = 76, P < 0.01)
sera escape: 22% of variance (coef 38 ± 6)
ACE2 binding: 4% of variance (coef 6 ± 2)
cell entry: 8% of variance (coef 21 ± 6)

**Extended Data Fig. 11** | See next page for caption.

**Extended Data Fig. 11 | Correlations of changes in growth with various other properties of spike for XBB descended clades.** This figure shows the *change* in growth rate between parent-descendant clade pairs versus the *change* in various spike phenotypes, rather than showing the absolute clade growth and absolute spike phenotypes as in Extended Data Fig. 10. Comparing the changes removes phylogenetic correlations as discussed in the main text. **a**, Correlation between the changes in growth rate for parent-descendant clade pairs versus the change in each spike phenotype measured in the XBB.1.5 full-spike deep mutational scanning described in the current paper (multiple mutations are assumed to have additive effects). These panels are the same as those shown in Fig. 6a, and are re-printed here to enable easier comparison to other panels in this figure. **b**, Correlations of changes in clade growth with changes in site-level antibody escape, ACE2 affinity, and RBD expression measured for RBD mutations in yeast-display deep mutational scanning. **c**, Correlation of changes in the EVEscape score with changes in clade growth. **d**, Ordinary least-squares regression of changes in the yeast-display RBD deep mutational scanning phenotypes versus changes in XBB-descendant clade growth. The small text indicates the unique variance explained by each variable as well as the coefficients in the regression. **e**, Ordinary least squares multiple linear regression of changes in XBB-descendant clade growth rate versus all three measured spike phenotypes using the XBB.1.5 full spike deep mutational scanning. This panel is the same as Fig. 6b, and is re-printed here to enable easier comparison to other panels in this figure. All panels are labeled with the Pearson correlation (r) and a P-value which is a one-sided test determined by computing how many randomizations of the mutational data yield correlations as large as the actual one.

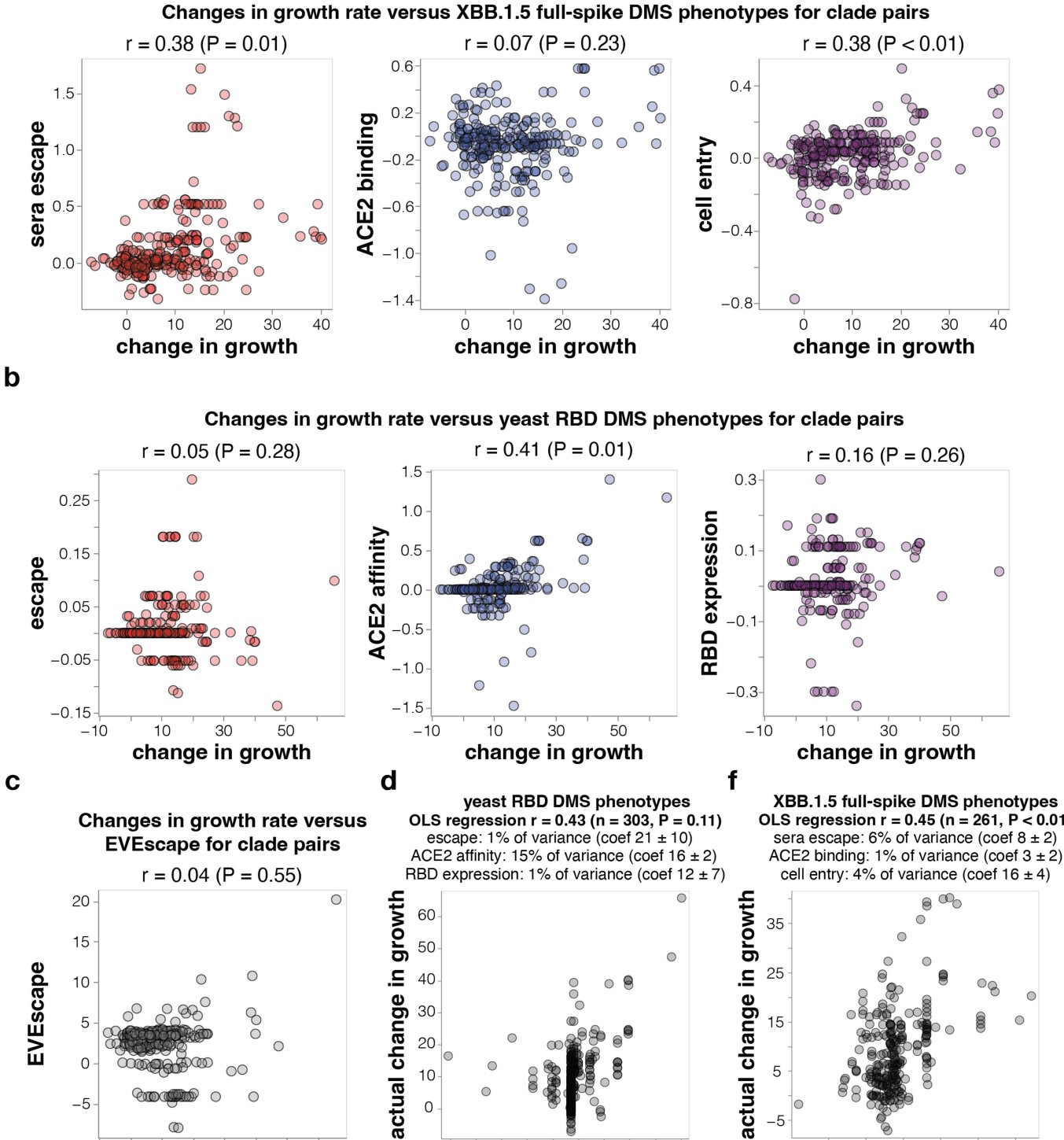

**Extended Data Fig. 12 | Correlations of changes in growth with various other properties of spike for BA.2, BA.5, and XBB descended clades.** This figure is the same as Extended Data Fig. 11 except that it includes clades descended from any of BA.2, BA.5, and XBB whereas Extended Data Fig. 11 includes just clades descended from XBB.

# Reporting Summary

## Statistics

For all statistical analyses, confirm that the following items are present in the figure legend, table legend, main text, or Methods section.

| n/a | Confirmed | |
|---|---|---|
| ☐ | ☒ | The exact sample size (*n*) for each experimental group/condition, given as a discrete number and unit of measurement |
| ☐ | ☒ | A statement on whether measurements were taken from distinct samples or whether the same sample was measured repeatedly |
| ☐ | ☒ | The statistical test(s) used AND whether they are one- or two-sided *Only common tests should be described solely by name; describe more complex techniques in the Methods section.* |
| ☒ | ☐ | A description of all covariates tested |
| ☐ | ☒ | A description of any assumptions or corrections, such as tests of normality and adjustment for multiple comparisons |
| ☐ | ☒ | A full description of the statistical parameters including central tendency (e.g. means) or other basic estimates (e.g. regression coefficient) AND variation (e.g. standard deviation) or associated estimates of uncertainty (e.g. confidence intervals) |
| ☐ | ☒ | For null hypothesis testing, the test statistic (e.g. *F*, *t*, *r*) with confidence intervals, effect sizes, degrees of freedom and *P* value noted *Give P values as exact values whenever suitable.* |
| ☒ | ☐ | For Bayesian analysis, information on the choice of priors and Markov chain Monte Carlo settings |
| ☒ | ☐ | For hierarchical and complex designs, identification of the appropriate level for tests and full reporting of outcomes |
| ☐ | ☒ | Estimates of effect sizes (e.g. Cohen's *d*, Pearson's *r*), indicating how they were calculated |

*Our web collection on statistics for biologists contains articles on many of the points above.*

## Software and code

Policy information about availability of computer code

| | |
|---|---|
| Data collection | Data described in the manuscript was acquired experimentally without specific software. Clade growth data is accessible at https://github.com/MurrellGroup/MultinomialLogisticGrowth/blob/main/model_fits/rates.csv . |
| Data analysis | Dada analysis code is available at: XBB.1.5 spike: https://github.com/dms-vep/SARS-CoV-2_XBB.1.5_spike_DMS BA.2 spike: https://github.com/dms-vep/SARS-CoV-2_Omicron_BA.2_spike_ACE2_binding XBB.1.5 RBD: https://github.com/dms-vep/SARS-CoV-2_XBB.1.5_RBD_DMS XBB.1.5 full spike, XBB.1.5 RBD spike and BA.2 full spike repositories are published via Zenodo with DOIs: 10.5281/zenodo.10981249 , 10.5281/zenodo.10981257 and 10.5281/zenodo.10981262, respectively. Note that most of the analysis in these GitHub repos is performed using dms-vep-pipeline-3 (https://github.com/dms-vep/dms-vep-pipeline-3), version 3.5.3. Python notebooks and raw event data used for mass photometry analysis are available at https://github.com/JackTaylorBrown/massphotometry . |

For manuscripts utilizing custom algorithms or software that are central to the research but not yet described in published literature, software must be made available to editors and reviewers. We strongly encourage code deposition in a community repository (e.g. GitHub). See the Nature Portfolio guidelines for submitting code & software for further information.

## Data

Policy information about [availability of data](availability of data)

All manuscripts must include a [data availability statement](data availability statement). This statement should provide the following information, where applicable:

- Accession codes, unique identifiers, or web links for publicly available datasets
- A description of any restrictions on data availability
- For clinical datasets or third party data, please ensure that the statement adheres to our [policy](policy)

Raw sequencing files have been uploaded under BioProjects: PRJNA1034580 for XBB.1.5 full spike library, PRJNA1035795 for XBB.1.5 RBD-only library, PRJNA1035933 for BA.2 full spike library.

Data generated by the analysis is available in the following directories:
XBB.1.5 spike: https://github.com/dms-vep/SARS-CoV-2_XBB.1.5_spike_DMS/results
BA.2 spike: https://github.com/dms-vep/SARS-CoV-2_Omicron_BA.2_spike_ACE2_binding/results
XBB.1.5 RBD: https://github.com/dms-vep/SARS-CoV-2_XBB.1.5_RBD_DMS/results

## Research involving human participants, their data, or biological material

Policy information about studies with [human participants or human data](human participants or human data). See also policy information about [sex, gender (identity/presentation), and sexual orientation](sex, gender) and [race, ethnicity and racism](race, ethnicity and racism).

| | |
|---|---|
| Reporting on sex and gender | *Use the terms sex (biological attribute) and gender (shaped by social and cultural circumstances) carefully in order to avoid confusing both terms. Indicate if findings apply to only one sex or gender; describe whether sex and gender were considered in study design; whether sex and/or gender was determined based on self-reporting or assigned and methods used. Provide in the source data disaggregated sex and gender data, where this information has been collected, and if consent has been obtained for sharing of individual-level data; provide overall numbers in this Reporting Summary. Please state if this information has not been collected. Report sex- and gender-based analyses where performed, justify reasons for lack of sex- and gender-based analysis.* |
| Reporting on race, ethnicity, or other socially relevant groupings | *Please specify the socially constructed or socially relevant categorization variable(s) used in your manuscript and explain why they were used. Please note that such variables should not be used as proxies for other socially constructed/relevant variables (for example, race or ethnicity should not be used as a proxy for socioeconomic status). Provide clear definitions of the relevant terms used, how they were provided (by the participants/respondents, the researchers, or third parties), and the method(s) used to classify people into the different categories (e.g. self-report, census or administrative data, social media data, etc.) Please provide details about how you controlled for confounding variables in your analyses.* |
| Population characteristics | *Describe the covariate-relevant population characteristics of the human research participants (e.g. age, genotypic information, past and current diagnosis and treatment categories). If you filled out the behavioural & social sciences study design questions and have nothing to add here, write "See above."* |
| Recruitment | *Describe how participants were recruited. Outline any potential self-selection bias or other biases that may be present and how these are likely to impact results.* |
| Ethics oversight | XBB* infection sera used in this manuscript were collected as part of prospective longitudinal Hospitalized or Ambulatory Adults with Respiratory Viral Infections (HAARVI) study after informed consent from participants. |

Note that full information on the approval of the study protocol must also be provided in the manuscript.

# Field-specific reporting

Please select the one below that is the best fit for your research. If you are not sure, read the appropriate sections before making your selection.

☒ Life sciences　　　☐ Behavioural & social sciences　　　☐ Ecological, evolutionary & environmental sciences

For a reference copy of the document with all sections, see [nature.com/documents/nr-reporting-summary-flat.pdf](nature.com/documents/nr-reporting-summary-flat.pdf)

# Life sciences study design

All studies must disclose on these points even when the disclosure is negative.

| | |
|---|---|
| Sample size | Two independently produced libraries were used for all DMS experiments to capture experimental noise. 10 randomly selected sera were used to perform serum escape experiments we used 10 sera because we considered this number to be sufficient to get a representative escape profile for population that had diverse virus exposures and vaccinations and was feasible to do withing experimental limitations. |

| Data exclusions | No data was excluded |

| Replication | All deep mutational scanning experiments were performed using 2 biological replicates (independently generated DMS libraries). Mass photometry experiments were done using two independently produced protein samples for each spike variant. Neutralization assays were performed using two technical replicates. Reproducibility of experiments was assessed by correlation between independent experiments , which can be seen in the correlation plots and standard errors provided in the manuscript. |

| Randomization | No specific randomization of samples was performed because the study does not include any comparison between experimental groups. |

| Blinding | No blinding was performed as the study as the study does not include comparison between experimental groups and is not a trial study. |

# Reporting for specific materials, systems and methods

We require information from authors about some types of materials, experimental systems and methods used in many studies. Here, indicate whether each material, system or method listed is relevant to your study. If you are not sure if a list item applies to your research, read the appropriate section before selecting a response.

## Materials & experimental systems

| n/a | Involved in the study |
|-----|------------------------|
| ☒ | Antibodies |
| ☐ | ☒ Eukaryotic cell lines |
| ☒ | Palaeontology and archaeology |
| ☒ | Animals and other organisms |
| ☒ | Clinical data |
| ☒ | Dual use research of concern |
| ☒ | Plants |

## Methods

| n/a | Involved in the study |
|-----|------------------------|
| ☒ | ChIP-seq |
| ☒ | Flow cytometry |
| ☒ | MRI-based neuroimaging |

## Eukaryotic cell lines

Policy information about cell lines and Sex and Gender in Research

| Cell line source(s) | 293T cells purchased from ATCC, Expi293F cells were purchased from Thermo Fisher Scientific |

| Authentication | none were authenticated |

| Mycoplasma contamination | not contaminated |

| Commonly misidentified lines (See ICLAC register) | N/A |

## Plants

| Seed stocks | Report on the source of all seed stocks or other plant material used. If applicable, state the seed stock centre and catalogue number. If plant specimens were collected from the field, describe the collection location, date and sampling procedures. |

| Novel plant genotypes | Describe the methods by which all novel plant genotypes were produced. This includes those generated by transgenic approaches, gene editing, chemical/radiation-based mutagenesis and hybridization. For transgenic lines, describe the transformation method, the number of independent lines analyzed and the generation upon which experiments were performed. For gene-edited lines, describe the editor used, the endogenous sequence targeted for editing, the targeting guide RNA sequence (if applicable) and how the editor was applied. |

| Authentication | Describe any authentication procedures for each seed stock used or novel genotype generated. Describe any experiments used to assess the effect of a mutation and, where applicable, how potential secondary effects (e.g. second site T-DNA insertions, mosiacism, off-target gene editing) were examined. |

