## [Peer Review file · Nature]

Manuscript Title: Spike deep mutational scanning helps predict success of SARS CoV-2 clades

Reviewer Comments & Author Rebuttals

Reviewer Reports on the Initial Version:

Referees' comments:

Referee #1 (Remarks to the Author):

This study profiled the impact of XBB.1.5 and BA.2 Spike mutations on cell entry, ACE2 binding, and sera escape based on previously reported pseudovirus deep mutational scanning (DMS) system, demonstrated that mutations in non-RBD region could notably impact the ACE2 binding and serum escape by modulating RBD conformation. Utilizing Spike phenotype data on ACE2 binding, cell entry, and serum evasion obtained by the pseudovirus DMS system, this study established a model to predict SARS-CoV-2 evolution. I believe this full Spike DMS platform could improve our understanding of the impacts and functions of SARS-CoV-2 Spike mutations on non-RBD sites, which are not well-studied and of great importance. However, based on the proposed methods and models, this study only retrospectively analyzed the growth rate changes of several XBB-descended clades over the last year, which is inconsistent with the declared original intent of the model's establishment. As a paper to be considered for publication in Nature, the proposed methods and models are expected to give predictive results of substantial confidence and the comparison among various prediction models should be more detailed. Thus, the below issues should be addressed before further consideration of acceptance.

Major points:

1. The study developed a model to predict the growth rate changes of the SARS-CoV-2 variant based on phenotypes of Spike mutation measured by the pseudovirus DMS system. However, this study only conducted a retrospective analysis by explaining the evolution of XBB-descended clades in the past year and did not provide remarkable predictions on the emergence and prevalence of future SARS-CoV-2 mutants. It is expected that the authors could not only provide a comprehensive retrospective analysis on the recent emergence of Spike mutations (especially non-RBD mutations that were less frequently investigated by previous studies), to rationally explain the high prevalence of non-RBD mutation hotspots on Spike (such as the NTD mutations observed in BA.2.86/JN.1 lineage), but also provide a reasonable prediction on the future evolutionary trends of the SARS-CoV-2 Spike protein, which would be critical for the evaluation and development of vaccine boosters.
2. Fig. 1b shows that NTD substitutions are generally less deleterious than RBD substitutions for cell entry, while many NTD deletions could be highly deleterious. We usually consider NTD as a more flexible region compared to RBD and NTD deletions are more common in previous SARS-CoV-2 variants. Please briefly discuss this phenomenon. Do these results indicate any unknown critical NTD sites for viral function?
3. In the section titled "Mapping escape from XBB* infection sera reveals heterogeneity among individuals", the study manifested the existence of inter-individual heterogeneity by differences in

mutation escape maps from diverse serum samples. However, it neglected to take into account the distinctions in the immunological background of serum samples, particularly in relation to vaccination histories. Should the observed distinct escape spectrum be attributed to their distinct immune histories, or other internal heterogeneity of humoral immune response? Moreover, the vaccination histories corresponding to each serum sample were not explicitly delineated in Supplementary Table 1.

4. The discussion of sensitizing mutations in the “Mapping escape from XBB* infection sera reveals heterogeneity among individuals” section is not sufficient. The capability of identifying these sensitizing mutations (mainly reversions) is an advantage of the pseudovirus-based full Spike DMS system. This should be considered a significant constraint for the prediction of viral evolution. And, do you observe any non-RBD sites that could facilitate the neutralization of serum samples?

It is concluded that many mutations could allosterically affect ACE2 and NAb binding by affecting the Spike up-down conformation dynamics. Is it possible to validate the results by testing several mAbs that could either bind up/down Spike (Class 2/3), or bind up Spike only (Class 1/4, and ACE2).

5. In the last section and Discussion, this study offered a comparative analysis between the pseudovirus-based full spike DMS system and the yeast display-based RBD DMS system. However, the comparison is too brief and not convincing enough. Please define the points in Extended Data Fig. 10d-f. Do they represent different clades, variants, residues, or mutations? Different methods might exhibit various efficiencies for predicting mutations on different regions of the Spike. Is it possible to analyze the performance the efficiency of predicting the prevalence of mutations) of each method for different regions on the Spike separately (NTD, RBD, S2 ...)?

6. The pseudovirus-based DMS system incorporates only roughly 7000 naturally occurring, manually curated mutations, excluding the scope of every single possible mutation within the spike protein, especially the non-RBD regions. The potential ramifications of this limitation on the model's predictive power for future viral evolution warrant further examination, and should be declared in the manuscript.

Minor Points:

1. Could you provide further explanation on why the designed mutation of XBB.1.5 RBD libraries, which saturates all amino acid mutations, is denoted as 3888 in Extended Data Fig. 1a?

2. The accompanying legend in Extended Data Fig. 1b is unclear and initially provokes confusion. A more explicit clarification of the numbers within and beyond the brackets would be beneficial.

3. It is noticed that Extended Data Fig. 1d appears identical to Figure 1B in the Cell publication titled “A pseudovirus system enables deep mutational scanning of the full SARS-CoV-2 spike”.

4. Figure 2d elucidates the correlation between the effects of mutations on ACE2 affinity according to the pseudovirus-based full spike DMS system and the yeast display-based RBD DMS system. XBB.1.5 libraries exhibit a poorer correlation compared to the BA.2 libraries. Can you shed light on the potential reasons behind this discrepancy?

5. Extended Data Fig. 2d only presents a heatmap illustrating the impact of non-RBD mutations on ACE2 affinity, which should be clarified in the figure legend.

6. The relevance of the numbers presented within brackets in the right panel of Fig. 3b should be detailed in the figure legend.

7. Changes have been made in the design and screening of mutations in the XBB.1.5 spike library relative to the BA.2 spike library. Could you expatiate the motivation behind these modifications?

8. A revised mutagenesis PCR protocol was implemented for the XBB.1.5 spike library as opposed to the BA.2 library, which seemingly resulted in a diminished average number of mutations per spike

variant in the XBB.1.5 library, as well as decreased mutations in the final obtained library (Extended Data Fig. 1a). Can you share any additional observations resulting from these modifications?

9. In Extended Data Fig. 4a, could you confirm if the lane on the far left represents a marker? Clear labeling of this segment in the image would be beneficial.

10. It would be preferable for all the Negative stain electron microscopy images in Extended Data Fig. 4b to include a scale bar.

11. Could you explain why the non-neutralizing control for the ACE2 binding ability experiment was not similarly adjusted to the RDPPro glycoprotein for experimental uniformity?

12. Before implementing the escape mapping for serum samples using the pseudovirus DMS system, is there a requirement for pre-treatment to inactivate the sera specimens? This is not clearly elucidated in the provided "Methods" section.

13. In the left panels of Figures 5a to 5c, does each dot represent a distinct amino acid substitution? If this is the case, could you clarify why the number of dots does not correspond with the number of amino acids illustrated in the logo plot of the right panel?

14. The citations for reference 28 in the "Recombinant Protein Production" subsection of the Methods are rendered in red.

15. There is a noticeable absence of the Author contributions section.

Referee #2 (Remarks to the Author):

The paper develops integrative deep mutational scanning as a method to measure changes in three molecular phenotypes of SARS-CoV-2 BA.2 and XBB.1.5 (cell entry, ACE2 binding, and neutralization by human sera) and applies these data to evolutionary predictions. The authors score mutations that escape human sera of multiple vaccinated and/or recently with XBB infected individuals. The analysis confirms that humans are heterogeneous with respect to antigenic changes. A novel result is that ACE2 binding can be substantially affected by mutations outside the RBD. Finally, an interesting observation is that the authors see changes in viral growth rates among different XBB descendants, which can be explained by results using deep mutational scanning which could aid as predictive approach. The principal claim of the paper is that deep mutational scanning data enable substantially improved evolutionary predictions. The authors present a correlation analysis of escape scores with growth rate advances of various sub-clades of XBB inferred using a multinomial logistic regression model. The paper raises a number of specific questions on the deep mutational scanning analysis (1,2) and, more fundamentally, on the application to predictions (3-5):

1. In the analysis of cell entry, the authors write: "Negative values indicate worse cell entry than the unmutated parental spike. Note that the library design favored introduction of substitutions and deletions that are well tolerated by spike, explaining why many mutations of these types have neutral to only modestly deleterious impacts on cell entry". In Fig 1b, indeed nearly all mutations have a neutral or negative impact. The authors should discuss this fact more, because it is a crucial part of cell infection. Also, it seems that no deletions were introduced in the RBD based on the BA.2 background. This should be discussed as well.

2. Using the new DMS approach for predicting ACE2 binding is an interesting idea. In Fig 3b, the

authors show RBD occupancy vs. ACE2 concentration for a known increasing or decreasing mutations in the RBD compared to a BA.2 background. S1 and S2 mutations predicted to increase ACE2 binding do not always show this effect. While S1 changes seem to have an effect increasing ACE2 binding, S2 changes always appear to reduce ACE2 binding. The authors explain that this effect could be due to the experimental setup and could be explained with effects on spike fusion. Would that mean that this approach does not work for all mutations affecting spike fusion? The authors should discuss this point more.

3. The growth advance analysis of the paper falls short of the stated goal of predictions. This is for two reasons.

(a) Using only the growth rate of the clade as a fitness measure, the authors look at a very short time-span locally in one part of the tree (the XBB subclade). Evolutionary success depends on the whole viral population over an intermediate time span, predictions require to assess the difference of its fitness with the average fitness in the population, taking into account the competition between clades. It is not sufficient to use only the difference in the growth rate compared with only the parent.

(b) The additive escape score ignores epistatic effects between multiple mutations. These are especially important for SARS, where new variants of concern often show multiple additional mutations; see, e.g., Thadani et al. (Nature 2023) for a recent treatment of such effects.

4. The statistical analysis of predictions is not state of the art.

(a) As the authors exclude clades that have fewer than 200 sequences, they bias against clades with decreased fitness, i.e., underweigh false positives.

(b) The data sample where the predictive power is compared to other methods seems to be too small. I understand that because of the full-spike DMS technique, an analysis of the full phylogenetic tree is not plausible. However, why is no analysis on the growth rates of the subclades in the BA.2 subtree included?

(c) There is no separation of training and validation data.

5. The analysis of empirical growth rates is unclear.

(a) As I understand, the inferred growth rate is the exponential growth of the number of sequences for a lineage per day. At the same time, the difference in growth rates between two variants is the selection coefficient and reflects the change in the relative frequencies of the two variants. While the growth rates are variant-specific, there is also a constant that is country-specific for each pair of variants that determines the intercepts. Could you give a more complete description of the method with the explicit likelihood function that is optimised over? What are the parameters that are being optimised?

(b) The inferred growth rates grow almost monotonically with time (fig. 6, fig.S10b). This does not include the expected time-dependence of growth within one clade; see, e.g., Yan et al. (eLife 2019).

(c) The growth analysis gives some surprising results: (1) XBB.1.5 is not inferred with a high clade growth in comparison with other foreground clades, even though XBB.1.5 did come up very fast. (2) A comparison of inferred growth values between HV.1 and HK.6 shows similar growth, which is different from what other people observe (<https://cov-spectrum.org>). The authors should explain the growth analysis in more detail and compare to other established methods.

In summary, this paper presents a convincing deep-mutational scanning analysis of multiple growth-relevant viral phenotypes of SARS-CoV-2 that is a promising avenue for future applications. The application to predictions, presented as the central message of the paper, is far less convincing. I appreciate that data of multiple phenotypes can help predictions, but the quantitative improvement is a bit incremental. On the conceptual and methodological side, the paper is rather a step back compared to the standard of recent publications (Thadani et al., Nature 2023; Meijers et al., Cell 2023).

Referee #3 (Remarks to the Author):

In this manuscript, the authors deploy a technology they recently developed, unveiled at the end of 2022, for performing deep mutational scanning of full-length SARS-CoV-2 spike proteins, to two important SARS-CoV-2 genomic variants, BA.2 and XBB.1.5, and further develop it to extend the phenotypes that it can measure. Using these two variants as backgrounds allows them to effectively cover the ancestors of the major variants that circulated in 2023 (and in 2024 to date, though inevitably circulating virus is now quite diverged from these ancestors). The authors provide phenotyping in several different dimensions, and then use genomic surveillance data from circulating SARS-CoV-2 sequences to measure the biological relevance of these screening data from the laboratory, finding that the screen data have significant power to explain evolution in the world. The combined analysis provides a tremendous wealth of data for those studying SARS-CoV-2's ongoing evolution, and the authors also develop important new tools that can be applied to future variants and other viruses. There is a very substantial amount of work described, bringing together deep mutational scanning, in different lineages and different contexts, with genomic surveillance data and the recently-developed analytical approach of mass photometry.

This approach to performing deep mutational scanning across the entire spike protein remains novel and impactful. The most important outcome of this work is the dataset produced, which will be well-used by the community. This dataset allows any researcher coming across a new lineage to assess whether the mutations it possesses are (on their own) likely to alter cell entry properties, to alter ACE2 binding, or to evade immunity induced by vaccination and infection. The authors' approach to documenting their analytical workflows is exemplary, with notebooks available to reproduce every step of the analytical workflows, and outputs available in convenient interactive forms and data files. The availability and documentation of code and workflows is at the very highest level for any analyses in this area, and to be commended. The Github URLs liberally distributed through the text are very helpful to the reader seeking additional data on any point.

The assay the authors are using measures the efficiency with which pseudoviruses of different genotypes can enter cells, but they exploit the fact that the neutralisation of this entry by exogenous ACE2 differs according to the strength with which each pseudovirus binds to ACE2 to assess the ACE2-binding of each mutant in their pool. The authors give some examples of the insights that this approach provides, including sites such as A222V, which emerged repeatedly in Delta and which causes increased ACE2 binding despite not being found in the spike RBD.

The utility of the dataset as a whole is brought home by the authors' demonstration using genomic

surveillance data that these mutational-scanning data alone are sufficient to predict (imperfectly, as they acknowledge) the change in clade growth rate between a parent clade and its descendant, created by the mutations that it acquired. The analysis here is careful and elegant, and is a useful contribution in providing benchmarking data against other approaches (which here are less effective) and as a benchmark for future approaches.

The main limitation of the deep mutation scanning approach, acknowledged in the Discussion, is that it looks (in practice) at each mutation in turn, while epistasis appears to also play a substantial role in the paths SARS-CoV-2 takes through its fitness landscape. Nevertheless, these analyses show that mutations analysed in isolation can still tell us a great deal.

The manuscript is easy to read and carefully written. All conclusions are robust and well-supported. Clarity has been prioritised and overstatement avoided. Statistical analysis is sound, figures are clear, and I find nothing of any substance to fault. I list some small stylistic suggestions below that occurred to me during my reading.

- "mutations observed at an appreciable number of times" - I found the 'at' here unexpected to my ear.

- I was initially somewhat surprised by the authors' decisions to bias their libraries so much in favour of mutations that have been observed occurring multiple times in GISAID, given that it is also interesting to understand why mutations are selected against. Reading their original methods paper gave me a clearer idea of the rationale (a high proportion of deleterious mutations can be problematic when many of the pseudoviruses carry multiple mutations). It could be worth reiterating this reasoning in the methods.

- "highly correlated between the replicate libraries for each spike, indicating the experiments have good precision (Extended Data Fig. 1e)." - 'precision' can have a technical connotation of relating to the degree of resolution in an assay, (i.e. something like the number of decimal places). Duplicates would not speak directly to that, and regardless here the data in the figure looks very bimodal, so repeatability or accuracy might be a better word?

Referee #1 (Remarks to the Author):

This study profiled the impact of XBB.1.5 and BA.2 Spike mutations on cell entry, ACE2 binding, and sera escape based on previously reported pseudovirus deep mutational scanning (DMS) system, demonstrated that mutations in non-RBD region could notably impact the ACE2 binding and serum escape by modulating RBD conformation. Utilizing Spike phenotype data on ACE2 binding, cell entry, and serum evasion obtained by the pseudovirus DMS system, this study established a model to predict SARS-CoV-2 evolution. I believe this full Spike DMS platform could improve our understanding of the impacts and functions of SARS-CoV-2 Spike mutations on non-RBD sites, which are not well-studied and of great importance. However, based on the proposed methods and models, this study only retrospectively analyzed the growth rate changes of several XBB-descended clades over the last year, which is inconsistent with the declared original intent of the model's establishment. As a paper to be considered for publication in Nature, the proposed methods and models are expected to give predictive results of substantial confidence and the comparison among various prediction models should be more detailed. Thus, the below issues should be addressed before further consideration of acceptance.

Thanks for the accurate summary of our work. We agree that the measurements made by deep mutational scanning improve our understanding of spike mutations. As detailed below, we have added new analyses to show how these measurements help predict viral evolution, both by adding analyses of the evolution of BA.2.86-descended clades that arose in the future relative to the time of our experiments, and also by expanding the retrospective analysis to include a broader set of clades.

Major points:

1. The study developed a model to predict the growth rate changes of the SARS-CoV-2 variant based on phenotypes of Spike mutation measured by the pseudovirus DMS system. However, this study only conducted a retrospective analysis by explaining the evolution of XBB-descended clades in the past year and did not provide remarkable predictions on the emergence and prevalence of future SARS-CoV-2 mutants. It is expected that the authors could not only provide a comprehensive retrospective analysis on the recent emergence of Spike mutations (especially non-RBD mutations that were less frequently investigated by previous studies), to rationally explain the high prevalence of non-RBD mutation hotspots on Spike (such as the NTD mutations observed in BA.2.86/JN.1 lineage), but also provide a reasonable prediction on the future evolutionary trends of the SARS-CoV-2 Spike protein, which would be critical for the evaluation and development of vaccine boosters.

This is a good comment, and we have added additional analyses that address the reviewer's point.

First, we have added analyses showing that our full-spike deep mutational scanning outperforms other approaches for predicting both the high fitness of the BA.2.86 lineage, which had not been

identified yet at the time we performed our experiments in the summer of 2023 (although it had been identified by the time we submitted our original paper). We also show that the deep mutational scanning outperformed other methods for identifying the high fitness of clades descended from BA.2.86 such as JN.1 (which mostly had not yet emerged even at the time of

original submission of our manuscript). For these analyses, we have used an approach similar to that of Thadani et al (2023) (ninth extended data figure of their paper) that involves generating random sequences with mutations observed at least a modest number of times across all SARS-CoV-2 sequences in GISAID, and then comparing the the actual sequence (either BA.2.86 or its descendant Pango clades) to randomly mutated sequences with the same number of mutations. This new analysis is in Extended Data Figure 15 of our revised manuscript, and part of the analysis is shown at left. For instance, the plot at left shows that according to the spike pseudovirus deep mutational scanning, BA.2.86 has a much more favorable phenotype than 1000 randomly generated spike sequences with the same number of mutations, both if we look at the linear model of all full-spike deep mutational scanning phenotypes parameterized on the XBB-descended clades, or if we look just directly at the measured cell entry and sera escape phenotypes. Looking at the actual phenotype of BA.2.86 versus the randomly generated mutants further shows that the full-spike deep mutational scanning has better predictive power than EVEscape or RBD yeast-display deep mutational scanning. Extended Data Figure 15 of our revised manuscript has a comparable plot showing that the same is true if we test the ability of the

different methods on descendant Pango clades of BA.2.86 relative to the original BA.2.86. **Overall, these new analyses show that our full-spike deep mutational scanning has better predictive power on the BA.2.86 clades that emerged after our experiments than alternative methods.**

Second, the reviewer notes that our retrospective analysis in the original manuscript only included XBB-descended clades. In the revised manuscript, we have also added a second retrospective analysis with expanded scope that includes all BA.2, BA.5, and XBB-descended clades with growth estimates. This new analysis is in Extended Data Figure 14 of the revised manuscript. The results on this expanded set of clades are essentially the same as for just XBB-descended clades: the full-spike deep mutational scanning does a reasonable job of predicting changes in clade

growth, and outperforms both RBD yeast-display deep mutational scanning and EVEscape.

Overall, this new analysis broadens the retrospective validation of the predictive power of the deep mutational scanning to include all clades evolutionarily downstream of BA.2.

Finally, the reviewer asks if we can predict future evolution. As mentioned above, we have shown that our measurements have substantial predictive power with respect to BA.2.86 (which emerged after our experiments were completed) and BA.2.86-descended clades (which mostly emerged after our original manuscript was submitted). Of course it is impossible for us to benchmark any predictions about evolution in the future to the time of writing of this response. However, we can describe how our data is being used to interpret / predict the current near-term evolution of SARS-CoV-2. Currently, there are three main mutations convergently appearing in the most rapidly growing SARS-CoV-2 variants: R346S/T, F456L, and T572I (see https://github.com/neherlab/SARS-CoV-2_variant-reports/blob/10458d09afa9dae9ffc5ef522078fe0769c209f1/reports/variant_report_latest_draft.md). Our data have been useful for understanding all of them as follows, and predicting they were important before they became widespread:

- R346S/T: On Dec-4-2023, a R346S Pango clade was designated (JN.1.1). In that Pango GitHub issue, we immediately identified that R346S was a significant serum-escape mutation based on the full-spike deep mutational scanning (see the here: <https://github.com/sars-cov-2-variants/lineage-proposals/issues/1148#issuecomment-1838996345>). Indeed, R346S/T mutations have since become common in most of the fastest growing new clades, demonstrating the predictive power of our deep mutational scanning.
- T572I: By late Dec-2023, many of the fastest growing clades were JN.1 descendants with T572I, which prior to our full-spike deep mutational scanning had no known phenotypic effect. We were able immediately explain the reason for spread of this mutation by noting that our deep mutational scanning showed T572I increased ACE2 binding by modulating RBD conformation (see https://twitter.com/jbloom_lab/status/1741199536280989821). Indeed, T572I is currently one of the three most important convergent mutations.
- F456L: This mutation is spreading rapidly, and is predicted by our deep mutational scanning to confer substantial serum escape with just modest cost to ACE2 binding and spike-mediated cell entry.

Overall, these examples demonstrate how our deep mutational scanning measurements have provided predictive power with respect to the three most important SARS-CoV-2 mutations that have emerged over the last few months.

2. Fig. 1b shows that NTD substitutions are generally less deleterious than RBD substitutions for cell entry, while many NTD deletions could be highly deleterious. We usually consider NTD as a more flexible region compared to RBD and NTD deletions are more common in previous SARS-CoV-2 variants. Please briefly discuss this phenomenon. Do these results indicate any unknown critical NTD sites for viral function?

The reviewer is correct in this comment—however, the distinction (which we failed to make in the original data display) is that the flexible loops but not the core β -sheets of the NTD are highly tolerant of deletions. Specifically, our libraries include single-residue deletions at most of NTD sites, and Fig.1b shows the effects of these deletions aggregated across the entire NTD. The “flexibility” of NTD that the reviewer notes mostly lies in its ability to accommodate indels and substitutions in the flexible loops that connect the β -sandwich that makes up the core of NTD. If we separate NTD

mutations by whether they occur in the core β -sheets or the flexible loops, we see that the NTD is more tolerant of both substitutions and deletions in the loops than in the core β -sheets (see plot at right). This difference is especially notable for deletions: the median effect of a single-residue deletion in the NTD loops is only slightly negative (median effect of about -0.25), but the median effect of a deletion in the core β -sheets is highly negative (median effect of about -5). Therefore, the NTD is really only tolerant of deletions in its flexible loops. We have added a sentence to the text explaining this fact, as well as adding the plot at right as Extended Data Fig. 1g of the revised manuscript.

3. In the section titled “Mapping escape from XBB* infection sera reveals heterogeneity among individuals”, the study manifested the existence of inter-individual heterogeneity by differences in mutation escape maps from diverse serum samples. However, it neglected to take into account the distinctions in the immunological background of serum samples, particularly in relation to vaccination histories. Should the observed distinct escape spectrum be attributed to their distinct immune histories, or other internal heterogeneity of humoral immune response? Moreover, the vaccination histories corresponding to each serum sample were not explicitly delineated in Supplementary Table 1.

The reviewer is correct that heterogeneity in sera escape can be due to vaccination/infection histories of the individuals from which these sera were obtained, or could be due to intrinsic differences in individuals’ humoral responses. We have added additional discussion of these possibilities in the Discussion section: we suspect vaccination/infection history may be a major factor based on other published work. We have also updated Supplementary Table 1 to give detailed vaccination and infection histories for the individuals from which each serum was collected.

However, we are not able to confidently assess whether exposure history is the main factor shaping the heterogeneity in the samples we analyzed. All of the samples we analyzed are from individuals who were imprinted by vaccination with the original vaccine prior to any infection. The individuals then have different numbers of infections. As shown in the plot below (which mimics Fig. 4a from the paper but is stratified by one versus multiple infections), there are some modest differences in key sites of escape among sera from individuals with one or multiple infections. However, the number of samples in our study is not sufficient to be confident that these differences are significant. In addition, because our study did not draw sera from defined cohorts with highly similar exposure histories, it is difficult to perform this analysis because every individual has a somewhat different history. We agree that future deep mutational scanning studies of serum escape, such as ones comparing individuals imprinted with the ancestral vaccine versus Omicron infection, could shed more light on this question. But for now all we can confidently conclude is that there is substantial heterogeneity across the study subjects, but can only speculate on the cause.

4. The discussion of sensitizing mutations in the “Mapping escape from XBB* infection sera reveals heterogeneity among individuals” section is not sufficient. The capability of identifying these sensitizing mutations (mainly reversions) is an advantage of the pseudovirus-based full Spike DMS system. This should be considered a significant constraint for the prediction of viral evolution. And, do you observe any non-RBD sites that could facilitate the neutralization of serum samples?

We agree with the reviewer that the ability to also detect sensitizing mutations that increase neutralization is a major strength of our lentivirus-based deep mutational scanning system, and provides useful information on SARS-CoV-2 evolution. We have expanded discussion of these mutations in the manuscript.

Most sensitizing mutations occur at sites which were mutated in previous major SARS-CoV-2 variants indicating that sera have antibodies targeting the original unmutated epitopes. As the reviewer notes, such reversions are less likely to occur in future variants because they increase sensitivity to neutralization by pre-existing antibodies targeting those sites. Furthermore, our ability to map sensitizing mutations also is of utility when predicting the evolution of non-XBB clades that start with different genetic backgrounds. For instance, XBB.1.5 already had the R346T serum-escape mutation, and so the T346R reversion is measured to be sensitizing in our XBB.1.5 deep mutational scanning. But JN.1 variants lack R346T, and so that mutation is now rising rapidly in JN.1 descendants—indicating that R346T is an escape mutation in JN.1, exactly as predicted by the fact that we measure T346R (the reverse mutation) to be sensitizing in our deep mutational scanning.

The interactive serum-escape maps we provide with our data (see here https://dms-vep.org/SARS-CoV-2_XBB.1.5_spike_DMS/htmls/summary_overlaid.html) allow visualization of sensitizing mutations if you set the ‘*floor escape at zero*’ toggle below the heatmap to ‘*false*’. In addition, in Extended Figure 9 we do analyze sensitizing mutations at sites outside the RBD. Most of the sensitizing mutations at non-RBD sites appear to put the RBD in a more up conformation, which makes it more accessible to neutralization by antibodies that bind to that can only bind to the RBD in the up conformation.

It is concluded that many mutations could allosterically affect ACE2 and NAb binding by affecting the Spike up-down conformation dynamics. Is it possible to validate the results by testing several mAbs that could either bind up/down Spike (Class 2/3), or bind up Spike only (Class 1/4, and ACE2).

This is an excellent suggestion, and we have added new data in the Extended Data Fig. 10 that validates the effects as suggested by the reviewer. Specifically, we have previously performed full-spike deep mutational scanning of monoclonal antibodies that can either only bind the RBD in the up conformation (REGN10933 and SC27) or can bind to the RBD in both the up and down conformation (LY-CoV1404). The prediction would be that the antibodies that can bind only in the up conformation would be affected both by mutations directly in their epitopes and by mutations that modulate RBD up-down conformation, whereas the antibody that can bind in both the up and down conformation would only be affected by mutations directly in its epitope. Indeed, the monoclonal antibody deep mutational scanning data are fully consistent with this prediction. We have added these data in new Extended Data Fig. 10, and have also added explanatory text in the legend to that figure and the main text. These new data strengthen our conclusions.

We have also updated the mass photometry analysis and added new mass photometry data for the XBB.1.5 strain (Figure 3, Extended Data Fig. 3-6) that further provide direct biophysical support for non-RBD mutations affecting RBD up-down conformation.

5. In the last section and Discussion, this study offered a comparative analysis between the pseudovirus-based full spike DMS system and the yeast display-based RBD DMS system. However, the comparison is too brief and not convincing enough. Please define the points in Extended Data Fig. 10d-f. Do they represent different clades, variants, residues, or mutations? Different methods might exhibit various efficiencies for predicting mutations on different regions of the Spike. Is it possible to analyze the performance the efficiency of predicting the prevalence of mutations) of each method for different regions on the Spike separately (NTD, RBD, S2 ...)?

As described above, we have greatly expanded the comparisons between different methods for predicting clade growth, including the full-spike versus yeast-display RBD deep mutational scanning. This includes adding several new supplementary figures (Extended Data Figures 12 to 15 in the revised manuscript). We have clarified in these revisions that each point in the plots referenced by the reviewer represents a different SARS-CoV-2 clade.

Unfortunately, we cannot compare clade growth on a per-spike-domain (eg, NTD, RBD, etc) basis, as clades can acquire mutations throughout the spike (and even the rest of the genome) that influence their growth, and in general it is not possible to strictly separate the effects of these mutations. However, we do clearly show that full-spike deep mutational scanning (which covers mutations at all spike domains) is better than yeast-display RBD deep mutational scanning (which only covers RBD mutations) at predicting clade growth. This fact suggests that mutations outside the RBD do make meaningful contributions to clade fitness. Indeed, in Fig. 3C we list some non-RBD mutations (T572I, T732I, G157L, S704L, E583D, Q52H) that likely enhance fitness (since they arise repeatedly in different clades) and that our experiments measure to have a beneficial effect.

As far as direct comparison between the full-spike and yeast-display RBD measurements, Fig. 2d shows that the effects of RBD mutations on ACE2 binding made using the two methods are quite

similar. Therefore, one main difference may be that the full-spike deep mutational scanning also captures the effect of non-RBD mutations, and these in turn sometimes make important contributions to clade fitness. The other major strength of the full-spike deep mutational scanning is that it directly measures escape from neutralization by polyclonal sera, whereas the yeast-display RBD deep mutational scanning only measures escape from binding by monoclonal antibodies. The former (polyclonal sera neutralization escape) likely better represents the true antigenic selection on SARS-CoV-2 during its evolution in the human population.

6. The pseudovirus-based DMS system incorporates only roughly 7000 naturally occurring, manually curated mutations, excluding the scope of every single possible mutation within the spike protein, especially the non-RBD regions. The potential ramifications of this limitation on the model's predictive power for future viral evolution warrant further examination, and should be declared in the manuscript.

The reviewer is correct that our libraries contain only a subset of spike mutations: namely those that are observed at least some small number of times in the ~16-million human SARS-CoV-2 sequences, plus all mutations at sites that are of clear evolutionary importance. However, this library-design strategy effectively covers nearly all relevant evolutionary mutations even into the future. The reason is that nearly every mutation that ends up emerging in any significant SARS-CoV-2 clade is already sampled in the ~16-million human SARS-CoV-2 sequences, since among this many sequences every evolutionary accessible and tolerated mutation is expected to be observed dozens to hundreds of times (see DOI 10.1093/ve/vead055 for exact statistics on how we expect to observe every tolerated mutation many times). Therefore, nearly all the mutations that we exclude from our libraries are ones that are not evolutionarily relevant (at least in the near- to mid-term future) because they are not accessible by single-nucleotide mutations and/or are highly deleterious.

To better demonstrate how our libraries effectively sample all relevant mutations, we have updated Extended Data Figure 1b to show how many of the mutations that have occurred in natural Pango clades are covered in our libraries. Even though our libraries were designed nearly a year ago (when we started the experiments), nearly all mutations that have occurred in natural Pango clades even up to the current date are well represented in our libraries. This includes even Pango clades descended from BA.2.86, which had not even been identified at the time we designed our library. This fact reflects the evolutionary dynamics of SARS-CoV-2: the mutations that end up spreading are almost always ones that are being extensively sampled in the millions of human SARS-CoV-2 sequences in GISAID—since for a virus with a high mutation rate and large population size like SARS-CoV-2, the key dynamic in evolution is generally not the waiting time for occurrence of a mutation, but rather the competition among different mutant clades that generally leads to only a small number of clades spreading widely at any given time.

In addition, we note that even very rare mutations in BA.2.86 (like the deletion of V483) were covered in our library as we saturated key sites. In fact, that was one of the facts that enabled us to correctly predict when BA.2.86 emerged in the late summer of 2023 that this deletion would have only a small adverse effect on ACE2 binding. Specifically, slide 9 at https://slides.com/jbloom/new_2nd_gen_ba2_variant shows how we were able to correctly assess the effect of delV483 in BA.2.86 when it emerged using our deep mutational scanning, despite the fact that we designed our library long before BA.2.86 was discovered.

For these reasons, the restriction of our libraries to the chosen ~7,000 mutations actually is not a major limitation for evolutionary prediction, as there is enough data to determine with fairly high accuracy which SARS-CoV-2 mutations will be relevant in near- to mid-term future evolution. However, the reviewer is correct that this library design does in principle introduce a limitation, and we now also state this more clearly.

Minor Points

1. Could you provide further explanation on why the designed mutation of XBB.1.5 RBD libraries, which saturates all amino acid mutations, is denoted as 3888 in Extended Data Fig. 1a?

RBD is 201 amino acids (positions 331-531), so a fully saturated library covers $201 \times 19 = 3,819$ different amino acid mutations (since each of the 201 residues can be mutated an of 19 other amino-acid identities). In addition to these amino acid mutations, we designed our libraries to include stop-codon mutations at 51 sites (as a negative control, as we know this will be deleterious), and single-residue deletions at another 18 sites (where we thought such deletions could plausibly occur), bringing the total number of distinct mutations to $3,819 + 51 + 18 = 3,888$.

2. The accompanying legend in Extended Data Fig. 1b is unclear and initially provokes confusion. A more explicit clarification of the numbers within and beyond the brackets would be beneficial.

We have added more explanatory text to the legend for Extended Data Fig. 1b. In particular, we now explain that the numbers outside the bracket indicate the total number of mutations observed in natural Pango clades, and the number inside the brackets indicate how many of these mutations are well covered in our libraries.

3. It is noticed that Extended Data Fig. 1d appears identical to Figure 1B in the Cell publication titled "A pseudovirus system enables deep mutational scanning of the full SARS-CoV-2 spike".

It is indeed the same figure and we explicitly cite our previous Dadonaite et al. (2023) paper in the legend for Extended Data Fig. 1d. The method for library production between that paper and the current study, but since understanding the method is crucial for understanding the experiments we felt it was important to re-include the experimental schematic since we know it is difficult for readers when they are forced to go read a separate reference to understand a key aspect of a study.

4. Figure 2d elucidates the correlation between the effects of mutations on ACE2 affinity according to the pseudovirus-based full spike DMS system and the yeast display-based RBD DMS system. XBB.1.5 libraries exhibit a poorer correlation compared to the BA.2 libraries. Can you shed light on the potential reasons behind this discrepancy?

The reviewer is correct to note that ACE2 binding measurements done with XBB.1.5 libraries correlate less well with yeast DMS-based measurements than BA.2 libraries. For most part this is because the XBB.1.5 full-spike libraries contain more mutations that are involved in RBD modulation than BA.2 libraries do. This is illustrated in the plot at left, an interactive version of which is now also included as part of analysis pipeline at

https://dms-vep.org/SARS-CoV-2_XBB.1.5_spike_DMS/htmls/binding_corr.html

Correlation between yeast DMS and full spike DMS for **all mutations**

Correlation between yeast DMS and full spike DMS for **only ACE2 proximal mutations**

Correlation between yeast DMS and full spike DMS for **only ACE2 distal mutations**

Correlation between yeast DMS and full spike DMS for **ACE2 proximal mutations shared between BA.2 and XBB.1.5 libraries**

Specifically, the plot included in this response shows that when only sites proximal to the ACE2 are considered, correlations between yeast display and full spike DMS systems increase considerably. Notably, measurements for ACE2 distal sites, that may be involved in modulating RBD movement, show no correlation with yeast display system because yeast display can only measure one-to-one ACE2-RBD interaction and does to capture effects of mutation in full trimeric spike context. In addition, the BA.2 libraries contain several mutations at R493 site that are some of the strongest binding increasing mutations in the RBD, which further increases correlation to the yeast system. XBB.1.5 lacks these mutations because it already has the R493Q mutation. Subsetting to show sites that are measured in all libraries further increases correlation

between XBB.1.5 yeast and lentivirus measurements to $r=0.86$. This correlation is still slightly lower than that of BA.2 ($r=0.9$) which may be due to measurement noise in either system.

5. Extended Data Fig. 2d only presents a heatmap illustrating the impact of non-RBD mutations on ACE2 affinity, which should be clarified in the figure legend.

The reviewer is correct. The figure panel only shows non-RBD mutations as that is what we are discussing. We have clarified the legend to clearly state that the panel only shows non-RBD mutations.

6. The relevance of the numbers presented within brackets in the right panel of Fig. 3b should be detailed in the figure legend.

We have clarified the legend to indicate that the numbers in the brackets reflect the deep mutational scanning measured value for each mutation.

7. Changes have been made in the design and screening of mutations in the XBB.1.5 spike library relative to the BA.2 spike library. Could you explain the motivation behind these modifications?

We made modest changes to the library design for XBB.1.5 versus BA.2 to better capture evolutionarily relevant mutations in our libraries. As described in the answer to the reviewer's major point (6) above, these changes helped us successfully capture relevant mutations.

Specifically, changes to the library design were made to reflect (i) increased number of available sequences on GISAID, (ii) newly available experimental measures of antigenically important sites and (iii) observations made by individuals tracking variants of unique mutations with potentially interesting phenotypes, and (iv) to ensure we included mutations that were at low frequencies but present in recent important variants. To this end we increased requirements for minimum mutation count on GISAID from 16 to 50. We also saturated sites that are important antigenically based on antibody escape calculator (https://ibloomlab.github.io/SARS2_RBD_Ab_escape_maps/escape-calc/) and RBD deep mutational scanning data from Yunlong Cao's group (<https://pubmed.ncbi.nlm.nih.gov/36535326/>); we cite these resources in the methods. In addition, Ryan Hisner, who tracks variants suggested including several deletions in the RBD that he has observed in a couple sequences from Ukraine that seemed to have transmitted, although were very rare on GISAID (notably del483 site that emerged in BA.2.86 was included in this set). Finally we set lower requirements for mutations counted on UShER tree if they were present in BA.2.75, BQ.1.1, XBB or XBB.1.5 clade sequences given the dominance of these clades at the time of library design.

8. A revised mutagenesis PCR protocol was implemented for the XBB.1.5 spike library as opposed to the BA.2 library, which seemingly resulted in a diminished average number of mutations per spike variant in the XBB.1.5 library, as well as decreased mutations in the final obtained library (Extended Data Fig. 1a). Can you share any additional observations resulting from these modifications?

The PCR library mutagenesis protocol for the XBB.1.5 and BA.2 libraries was actually the same. The reviewer is correct that there was a modest difference in the average number of mutations per barcoded variant (1.9 mutations per variant in XBB.1.5 library versus 2.3 mutations per variant in BA.2 library). In truth, this type of modest difference is within the range of differences that could have just occurred in day-to-day variation in PCR mutagenesis, since the two libraries were constructed months apart. However, there were also some small differences in primer design that could have contributed to the differences. The BA.2 library design had a set of primers that included pairs of mutations that are in close proximity, which we reasoned is important for producing variants with two mutations nearby. The revised primer design for the XBB.1.5 libraries used only single-mutation containing primers but for sites that were saturated we designed pairs of primers that target the same site but are offset from each other slightly, which we hoped would decrease the chances of sites dropping out due to poor primer properties (which occasionally happens). These adjustments in primer design could possibly have changed mutation frequencies per spike when the same PCR mutagenesis protocol was used.

The total number of variants per library is not determined by PCR mutagenesis protocol but by the low MOI infection step in library production protocol that stores each proviral genome in 293T-rTTA

cells. We build libraries by infecting cells at MOI of < 0.01 and the exact number of infectious units required to get to this MOI is determined by the VSV-G pseudotyped virus titers and the number of cells per plate. Both VSV-G titration and cell counting steps have some amount of noise and therefore even if aiming to build libraries with the same number of variants the exact number of variants in the final libraries can reasonably vary by 2-fold and that would still be within experimental noise. And again XBB.1.5 and BA.2 libraries were made months apart, which also contributed to a different number of variants in the final libraries.

9. In Extended Data Fig. 4a, could you confirm if the lane on the far left represents a marker? Clear labeling of this segment in the image would be beneficial.

Yes, that lane on the far left is a ladder. We have now labeled the protein ladder on the gel, which is in Extended Data Fig. 5a in the revised manuscript.

10. It would be preferable for all the Negative stain electron microscopy images in Extended Data Fig. 4b to include a scale bar.

We have added scale bars to all negative stain images, which are in which is in Extended Data Fig. 5b of the revised manuscript.

11. Could you explain why the non-neutralizing control for the ACE2 binding ability experiment was not similarly adjusted to the RDPro glycoprotein for experimental uniformity?

Because RDPro pseudotypes lentivirus worse than VSV-G it is more laborious to produce and requires an additional concentration step to increase titers. We therefore tend to only use RDPro as a non-neutralizing standard if VSV-G is being neutralized by whatever neutralization agent we are using in experiments, as is the case for human sera (which sometimes weakly inhibits VSV-G but not RDPro infection). As ACE2 does not interact with VSV-G and therefore does not inhibit it, we used VSV-G standard for the ACE2 binding experiments. We have added text in the methods explaining this.

12. Before implementing the escape mapping for serum samples using the pseudovirus DMS system, is there a requirement for pre-treatment to inactivate the sera specimens? This is not clearly elucidated in the provided "Methods" section.

We have added a statement to the methods section clarifying that sera were inactivated for 1h at 56°C before being used in any experiments. This inactivation should remove any complement activity.

13. In the left panels of Figures 5a to 5c, does each dot represent a distinct amino acid substitution? If this is the case, could you clarify why the number of dots does not correspond with the number of amino acids illustrated in the logo plot of the right panel?

Yes the points in the scatter plots in the left panels of Figure 5a-c do represent distinct substitutions, and we have clarified that in the legend. All the mutations are shown in the logo plots in the right panels as well. However, the height of each letter in the logo plot is proportional to the escape caused by that mutation—so if the escape from that amino-acid mutation is very close to zero, then the letter height for that amino acid is very small and therefore not visible on the logo plot.

14. The citations for reference 28 in the "Recombinant Protein Production" subsection of the Methods are rendered in red.

Thanks for catching the error, we have fixed the color.

15. There is a noticeable absence of the Author contributions section.

Thanks for noting this oversight, we have added an Author Contributions section.

Referee #2 (Remarks to the Author):

The paper develops integrative deep mutational scanning as a method to measure changes in three molecular phenotypes of SARS-CoV-2 BA.2 and XBB.1.5 (cell entry, ACE2 binding, and neutralization by human sera) and applies these data to evolutionary predictions. The authors score mutations that escape human sera of multiple vaccinated and/or recently with XBB infected individuals. The analysis confirms that humans are heterogeneous with respect to antigenic changes. A novel result is that ACE2 binding can be substantially affected by mutations outside the RBD. Finally, an interesting observation is that the authors see changes in viral growth rates among different XBB descendants, which can be explained by results using deep mutational scanning which could aid as predictive approach. The principal claim of the paper is that deep mutational scanning data enable substantially improved evolutionary predictions. The authors present a correlation analysis of escape scores with growth rate advances of various sub-clades of XBB inferred using a multinomial logistic regression model. The paper raises a number of specific questions on the deep mutational scanning analysis (1,2) and, more fundamentally, on the application to predictions (3-5):

Thanks for the excellent summary of our paper. We respond to the specific questions below.

1. In the analysis of cell entry, the authors write: "Negative values indicate worse cell entry than the unmutated parental spike. Note that the library design favored introduction of substitutions and deletions that are well tolerated by spike, explaining why many mutations of these types have neutral to only modestly deleterious impacts on cell entry". In Fig 1b, indeed nearly all mutations have a neutral or negative impact. The authors should discuss this fact more, because it is a crucial part of cell infection. Also, it seems that no deletions were introduced in the RBD based on the BA.2 background. This should be discussed as well.

The reviewer is correct that both the XBB.1.5 and BA.2 libraries were designed with the goal of including tolerated mutations that could conceivably be relevant in SARS-CoV-2 evolution, but to minimize the inclusion of highly deleterious mutations expected to completely disrupt spike function. As noted in our response to Major Point 6 of Reviewer 1, we were largely successful in the goal of including all mutations likely to be relevant in near-term SARS-CoV-2 evolution, as nearly all mutations that have appeared in new viral clades (including both XBB and BA.2.86 descendants) are well sampled in our library.

As the reviewer notes, even with the library design that tries to favor tolerated mutations, most mutations are neutral or deleterious for cell entry. This is because the vast majority of mutations to any well-adapted protein are deleterious or neutral, and so even with a library design that disfavors the most deleterious mutations, the overall distribution of effects is slanted towards deleterious /

neutral. To make this point more clearly, we have now also included a plot of the distribution of effects of all RBD mutations in our XBB.1.5 RBD-only libraries, which include *all* RBD mutations. As can be seen in that plot, which is included as Extended Figure 1g and shown at right, the distribution of mutation effects is substantially more deleterious in the all-RBD mutations library than for RBD mutations in the libraries designed to include only tolerated mutations.

Additionally, the reviewer is correct that we included RBD deletions only in the XBB.1.5 libraries. The reason is that we designed the BA.2 libraries first, and had not really considered the possibility that deletions in the RBD could be important at that point. But by the time we designed the XBB.1.5 libraries, deletions had begun to be observed in some chronic infection sequences, so we decided to include RBD deletions at key sites. This turned out to be a good decision, as delV483 (which is included in our XBB.1.5 libraries) is found in BA.2.86.

We have added additional text to the manuscript elaborating on the above issues as suggested by the reviewer.

Effects of mutations in the saturated RBD library

2. Using the new DMS approach for predicting ACE2 binding is an interesting idea. In Fig 3b, the authors show RBD occupancy vs. ACE2 concentration for known increasing or decreasing mutations in the RBD compared to a BA.2 background. S1 and S2 mutations predicted to increase ACE2 binding do not always show this effect. While S1 changes seem to have an effect increasing ACE2 binding, S2 changes always appear to reduce ACE2 binding. The authors explain that this effect could be due to the experimental setup and could be explained with effects on spike fusion. Would that mean that this approach does not work for all mutations affecting spike fusion? The authors should discuss this point more.

The reviewer is correct that the mass photometry data we report (note that we have added additional validation data in revision, as described elsewhere in this response) provide excellent support for the effects of S1 mutations on ACE2 binding as identified in the deep mutational scanning. This biophysically validates how NTD, SD1, and non-ACE2-proximal RBD mutations affect ACE2 binding.

But the reviewer is also correct that the S2 mutations identified in the deep mutational scanning as affecting ACE2 binding generally do not validate in the mass photometry. We think that the reason S2 mutations do not validate well as ACE2 binding increasing mutations is mostly due to technical aspects of the constructs used for mass photometry experiments. Namely, in order to produce high amounts of stable spike protein, the spike needs to be pre-fusion stabilized (just like the constructs in vaccines). In our case we employ a widely used HexaPro spike construct, which contains 6 proline substitutions in S2. While stabilization helps with recombinant protein expression it restricts the conformational changes spike can undergo in the S2 domain, which in non-stabilized spikes undergoes a large conformational change to mediate membrane fusion. Concurring with this hypothesis, we previously showed that ACE2-induced allosteric conformational changes exposing the fusion peptide were inhibited by the prefusion-stabilizing 2P mutations (PMID: 35857703).

Therefore, the prefusion stabilizing mutations in the HexaPro constructs used in our mass photometry likely mask the effects that other S2 mutations would normally have on spike dynamics by locking S2 into a more rigid prefusion conformation. We are endeavoring to test this hypothesis by producing sufficient amounts of non-stabilized recombinant spike for the mass photometry experiments, but have not yet succeeded in performing experiments with non-stabilized spike.

We have added text to the revised manuscript more completely explaining these points.

3. The growth advance analysis of the paper falls short of the stated goal of predictions. This is for two reasons. (a) Using only the growth rate of the clade as a fitness measure, the authors look at a very short time-span locally in one part of the tree (the XBB subclade). Evolutionary success depends on the whole viral population over an intermediate time span, predictions require to assess the difference of its fitness with the average fitness in the population, taking into account the competition between clades. It is not sufficient to use only the difference in the growth rate compared with only the parent.

These are excellent comments, and we have added substantial new material in the revised manuscript to address them.

First, the reviewer is correct that in our original submission we compared the deep mutational scanning measurements only to the growth of XBB-descended clades. In the revised manuscript, we retain the comparison of deep mutational scanning measurements to growth of XBB-descended clades (revised Fig. 6 and Extended Data Fig. 13) but **also** add comparison of the deep mutational scanning measurements to the growth of all BA.2-, BA.5-, and XBB-descended clades (the new Extended Data Fig. 14). This comparison addresses the reviewer's comments about the comparison in the original paper being over only a portion of the tree. As shown in the new Extended Data Fig. 14, our deep mutational scanning measurements still have a significant ability to predict the growth of the broader set of clades (all BA.2, BA.5, and XBB descendants), and outperforms the other methods we test (yeast display RBD deep mutational scanning and EVEscape). The ability of our deep mutational scanning measurements to predict the growth of the broader set of clades is slightly worse than for the XBB-descended clades, perhaps reflecting slightly different antigenicity across clades (e.g., different mutations causing the most antibody escape due to changes in the specificity of human immunity over time due to exposures / vaccinations with new strains)—but the deep mutational scanning still outperforms any other method on this broader set of clades.

In addition, as described earlier in this response, we have also added an evaluation of the ability of our deep mutational scanning (as well as other methods) to predict the evolution of the highly divergent BA.2.86 clade, which emerged after completion of our experiments and really only dramatically increased in frequency (in the form of its JN.1 descendant) after submission of our original manuscript. Because there is not enough time / subclades to perform a meaningful analysis of clade growth within JN.1 clades, we instead used an alternative approach inspired by Thadani et al (2023) of evaluating how well our deep mutational scanning could distinguish the favorable properties of the clades that actually emerged versus sequences with the same number of random mutations. As shown in the new Extended Data Fig. 15, this new analysis again shows our deep mutational scanning has significant ability to distinguish the clades that actually emerge from sequences with the same number of random mutations, and that it outperforms other methods (eg, EVEscape, yeast-display deep mutational scanning) in this respect.

Overall, the above two additions address the reviewer's comments by expanding the scope of comparison from just XBB-descended clades (original manuscript) to also include a much broader set of clades, as well as the BA.2.86-descended clades that are now dominant.

The reviewer also notes that clade growth needs to be estimated over the entire viral population. In fact, this is exactly how we have estimated clade growth, and this was perhaps not adequately explained in the original version (see response to this reviewer's comment 5 for more details on how clade growth was estimated). However, although clade growth is estimated over the entire viral population, comparison of mutation-based scoring metrics to clade growth is best done by then making the comparisons at the level of parent-daughter clade differences rather than absolute values over the entire population. The reason that this is the better approach is explained in general terms in Felsenstein's famous paper on phylogenetic contrasts (Felsenstein, 1985, *The American Naturalist*, 125:1-15, <https://www.jstor.org/stable/pdf/2461605.pdf>). Briefly, as Felsenstein explains, the nodes on a phylogenetic tree are not independent. So it is not correct to treat different nodes (in this case, clades) on the tree as independent data points in a correlation analysis, because they share most of their evolutionary history. However, the branches on the tree (differences between a node and its parent) can be treated as independent data points, since each parent-descendant branch represents independent evolution. For instance, right now the fastest growing viral clades are all JN.1 descendants (JN.1.4.3, JN.1.7, JN.1.18, etc). But these are not the fastest growing clades because JN.1.4.3, JN.1.7, and JN.1.18 all independently acquired the ~30 mutations that distinguish them from XBB.1.5. Rather, most of the growth advantage of these three clades is due to the shared mutations they acquired due to their recent shared common ancestry from JN.1, plus a bit of additional advantage from the specific mutations each acquired relative to JN.1. It would therefore be inappropriate to treat the high clade growth of JN.1.4.3, JN.1.7, and JN.1.18 as three independent data points when correlating mutation-based metrics to clade growth, since most of the mutations are shared due to common ancestry.

This problem is seen most clearly in Extended Data Fig. 12c, which simply shows the correlation of clade growth to Hamming distance (number of mutations) from Wuhan-Hu-1. Absolute clade growth is highly correlated with Hamming distance from Wuhan-Hu-1 because newer clades tend to have more mutations (in part because they are descended from previously successful highly mutated clades). But this observation is not really of meaningful predictive power, since at any given time there are always a variety of new clades with more mutations, and the relevant question is *which of these new clades will be successful?* So what we really want to do is assess which of the new clades that arise at any time will be more successful than their current parents. Indeed, if we correlate *change* in clade growth between parent-descendant clade pairs with change in Hamming distance, then Hamming distance is no longer a meaningful predictor (Extended Data Fig 12d). Another way to look at the same thing is to test if various potential predictors correlate with growth better than a null distribution generated by randomizing the predictors among mutations. As can be seen in Extended Data Fig. 12c-g, predictors like Hamming distance from Wuhan-Hu-1 or EVEscape correlate with absolute clade growth, but the correlation is no better than a null distribution generated by randomizing the predictions among mutations. Therefore, the apparent correlation of Hamming distance with clade growth (and much of the correlation of EVEscape with clade growth) is just due to shared phylogenetic ancestry and the fact that new clades have more mutations. In contrast, the deep mutational scanning correlations with clade growth *are* higher than those generated by randomizing the data, as indicated by the P-values in Extended Data Fig. 12e,g. The advantage of correlating changes in clade growth is that it avoids this issue where

things like just counting mutations correlate with clade growth, for the reason explained in Felstenstein's paper on phylogenetic contrasts. That is what is done in the main text figures in the paper. However, the supplement (Extended Data Fig. 12) also contains the correlations with absolute clade growth although it is then necessary to look at the P-values to assess how meaningful correlations are—and when that is done, it is clear that both methods clearly show that the full-spike deep mutational scanning provides the best predictions.

Finally, we also note that for BA.2.86 (where there are not enough subclades for a meaningful clade-growth based estimate), we have used the approach of Thadani et al (2023) of making predictions for actual versus random-mutation sequences. As shown in Extended Data Fig. 15, that analysis again shows that the full-spike deep mutational scanning provides the best predictions.

(b) The additive escape score ignores epistatic effects between multiple mutations. These are especially important for SARS, where new variants of concern often show multiple additional mutations; see, e.g., Thadani et al. (Nature 2023) for a recent treatment of such effects.

The effectiveness of predictions depends on two factors: the amount of relevant information carried in the data used to generate the predictions, and accuracy of the model that converts that information into the predictions. The major contribution of our work is not to formulate a better or more complex model, but rather to generate experimental data that is extremely informative about SARS-CoV-2 evolution.

In particular, in Fig. 6 and Extended Data Figs. 13, 14 and 15 we show that our direct measurements of serum escape and cell entry from our experiments are quite informative about the success of SARS-CoV-2 clades, substantially more so than the models of Thadani et al (2023). We fully agree that the models of Thadani et al accommodate more complexity, for instance by allowing the potential for epistasis among mutations. But in practice, as shown by a variety of experimental work¹⁻³ including our own, the main pressure driving the evolution of human SARS-CoV-2 is pressure to escape from human neutralizing antibodies. This is the reason that even simple univariate correlations of our serum-escape measurements predict SARS-CoV-2 clade success better than any other metric. In the case of Thadani et al, they have a detailed model for the effects of mutations on protein functionality (their EVE model), but the escape portion essentially just quantifies the extent to which spike substitutions are on the protein's surface and cause a substantial change in amino-acid physicochemical properties. Empirically, the results presented in our paper show that the direct measurements of serum escape made in our deep mutational scanning are far more informative.

Therefore, we agree that our modeling is much simpler than that of Thadani et al. Our paper reports predictors that either involve no model at all (just direct measurements) or a simple linear model of three different measurements. The fact that these measurements outperform models such as those of Thadani et al show that our deep mutational scanning is capturing highly biologically relevant information. This does not mean one strategy is “better” than the other— future work should try to merge the data advances in our paper with modeling advances from others. We have made modifications to the text to stress this point. For instance, we have added a new last sentence to the Discussion: **“An important area of future work will be integrating these highly informative experimental measurements into more sophisticated models of viral evolution (Meijers et al, 2023; Thadani et al, 2023; Abousamra et al, 2023).”**

4. The statistical analysis of predictions is not state of the art. (a) As the authors exclude clades that have fewer than 200 sequences, they bias against clades with decreased fitness, i.e., underweigh false positives. (b) The data sample where the predictive power is compared to other methods seems to be too small. I understand that because of the full-spike DMS technique, an analysis of the full phylogenetic tree is not plausible. However, why is no analysis on the growth rates of the subclades in the BA.2 subtree included? (c) There is no separation of training and validation data.

These are all good suggestions. We have addressed them as follows:

(a) The reviewer is correct that our comparisons to clade growth exclude clades with fewer than 200 sequences. The reason for this is relatively straightforward: we are comparing against clade growth, and it is not possible to reliably estimate the growth rates of clades with very few sequences. In the revised manuscript, we have added a new analysis of BA.2.86 and its descendant clades (Extended Data Fig. 15) that uses a different metric than comparison to clade growth: instead it simply compares the scores assigned to actual observed clade sequences versus random sequences with the same number of mutations (this is the metric used by Thadani et al). We think this change should largely address the reviewer's comment. We are of course open to suggestions by the reviewer for additional comparison metrics, but not it is simply impossible for us to perform a clade-growth based analysis for clades with insufficient sequences to estimate growth.

(b) As discussed above, in the revised manuscript, we have extended the comparisons to additional sets of viral clades. Specifically, we now compare the full-spike deep mutational scanning to the natural evolution of XBB-descended clades (Fig. 6 and Extended Data Fig. 13), all clades descended from any of BA.2, BA.5 or XBB (Extended Data Fig. 14), and BA.2.86 and its descendant clades (Extended Data Fig. 15). In all cases, the full spike deep mutational scanning provides predictions that are both significant and better than any of the comparator predictors tested (small P values when compared to randomizing the predictors).

(c) The reviewer is correct that our original manuscript did not separate training and test data. This is largely because we are not doing any training. For the most part, we are directly comparing experimental measurements to viral growth, with no training or free parameters. For instance, Fig. 6c shows direct univariate correlations of our experimental measurements to viral clade growth. Since there are no free parameters, there is no need for separation of test and training data, as nothing is being trained. In other words, we are simply demonstrating that the measurements that come directly out of our experiments are substantially predictive of clade growth, without any additional modeling or parameterization. The only comparison with clade growth that involves any "training" (fitting of parameters) is the multiple-linear regression in Fig. 6d and Extended Data Fig. 13d,f, where we fit a linear model of our three experimental measurements to predict clade growth. Those linear models involve three free parameters (the coefficients in the linear model assigned to each experimentally measured phenotype). We do show that the correlations are statistically significant by randomizing the predictors and showing that the real experimental data predict clade growth better than randomized data. In addition, we have now added an analysis of a phylogenetically entirely distinct group of viruses (BA.2.86 and its descendant clades), and shown that the linear model fit on the XBB-descended clades provides the best method of distinguishing the actual BA.2.86 clades that have emerged from randomly mutated sequences (Extended Data Fig. 15). This analysis of BA.2.86 clades using a linear model trained on XBB clades represents a

stringent separation of training and test data for the only part of our analysis that involves any training (fitting of parameters).

5. The analysis of empirical growth rates is unclear. (a) As I understand, the inferred growth rate is the exponential growth of the number of sequences for a lineage per day. At the same time, the difference in growth rates between two variants is the selection coefficient and reflects the change in the relative frequencies of the two variants. While the growth rates are variant-specific, there is also a constant that is country-specific for each pair of variants that determines the intercepts. Could you give a more complete description of the method with the explicit likelihood function that is optimised over? What are the parameters that are being optimised? (b) The inferred growth rates grow almost monotonically with time (fig. 6, fig.S10b). This does not include the expected time-dependence of growth within one clade; see, e.g., Yan et al. (eLife 2019). (c) The growth analysis gives some surprising results: (1) XBB.1.5 is not inferred with a high clade growth in comparison with other foreground clades, even though XBB.1.5 did come up very fast. (2) A comparison of inferred growth values between HV.1 and HK.6 shows similar growth, which is different from what other people observe (<https://cov-spectrum.org>). The authors should explain the growth analysis in more detail and compare to other established methods.

We have added much more detailed methods on the growth rate estimates, that should address the reviewer's questions.

In addition, we have incorporated an additional set of growth estimates made by the group of Trevor Bedford using multinomial logistic regression, available at <https://nextstrain.github.io/forecasts-ncov/> and made using the methods described in Abousamra et al (<https://www.medrxiv.org/content/10.1101/2023.11.30.23299240v3>). Due to this data addition, we have also added Trevor Bedford and Marlin Figgins to the author list of our paper. As shown in the new Extended Data Fig. 12a, the growth estimates from the Bedford lab are almost perfectly correlated with the ones used in our original submission, demonstrating that those estimates are consistent with other established methods.

To address specific questions from the reviewer:

- a) The text we have added to the methods explains the growth rates better. Basically, the relative change in frequency of two clades should be proportional to the exponential of the difference in their growth rates.
- b) Yes, the inferred clade growth rates generally increase monotonically over time. This is because the clades that emerge in the future generally have higher growth rates than the clades they replace, which is why they spread. This misunderstanding could be due to confusion over what we are estimating: we are not estimating change in each clade's frequency over time (which obviously can go down as a clade is replaced), but rather its inherent growth rate relative to other clades (which is expected to generally increase for newer clades that replace old ones).
- c) The model does give XBB.1.5 a substantially higher growth rate than the clades that preceded it or occurred at about the same time. This may be somewhat difficult to see with the coloring in Fig. 6a since blues cover a wide range of growth, but can be seen especially clearly by looking at the branch to XBB.1.5 in Fig. 6b. Specifically, XBB.1.5 is estimated to have a growth rate of 29 compared to for instance 5 for XBB.1.4 and 1.2 for XBB.1. However, although XBB.1.5 had a higher growth rate than other clades designated in late

2022 (which is why it came up so quickly), it had lower growth rate than later clades like EG.5.1 and subsequently HV.1 and HK.3 that replaced it. As far as HV.1 versus HK.6, the estimates were made when HK.6 only had ~600 sequences which could contribute to some noise for those specific clades—but also, HK.6 and HV.1 do have somewhat different growth rates in our estimates (60 for HK.6, 69 for HV.1). As suggested by the reviewer, we have systematically compared our estimates to the ones from the Bedford lab (see above), and they are very similar.

In summary, this paper presents a convincing deep-mutational scanning analysis of multiple growth-relevant viral phenotypes of SARS-CoV-2 that is a promising avenue for future applications. The application to predictions, presented as the central message of the paper, is far less convincing. I appreciate that data of multiple phenotypes can help predictions, but the quantitative improvement is a bit incremental. On the conceptual and methodological side, the paper is rather a step back compared to the standard of recent publications (Thadani et al., Nature 2023; Meijers et al., Cell 2023).

We fully agree that our paper does not include advanced modeling like Thadani et al or Meijers et al. As discussed above, the major value of our paper is to generate a new type of experimental data that is tremendously informative about SARS-CoV-2 evolution. The direct experimental measurements of serum escape from our deep mutational scanning (with no modeling or free parameters) is more predictive of SARS-CoV-2 evolution than the EVEscape method of Thadani et al in all comparisons (see Extended data Figs. 12-15). This is not a criticism of the modeling frameworks in other papers, but rather just a reminder that predictions depend both on the quality of the model and the informativeness of the data. For instance, both Thadani et al or Meijers et al model the antigenic phenotypes of different clades. Our experiments provide new measurements of antigenic effects of mutations that are clearly far more informative than those used by Thadani et al, given that our direct measurements of sera escape with no fit parameters outperform the full model of Thadani et al.

Therefore, the path forward is to continue to merge models like those mentioned by the reviewer with highly informative data like what we present. It is no more fair to dismiss our highly predictive experimental measurements because we do not include a model than it would be to dismiss the modeling work of Thadani et al or Meijers et al because they do not report any new experimental measurements. Advances both in generating information and improving models are valuable, and the greatest progress will come from synthesizing advances in both. We have added text to the discussion emphasizing this point, namely the new final sentence: **“An important area of future work will be integrating these highly informative experimental measurements into more sophisticated models of viral evolution (Meijers et al, 2023; Thadani et al, 2023; Abousamra et al, 2023).”**

Referee #3 (Remarks to the Author):

In this manuscript, the authors deploy a technology they recently developed, unveiled at the end of 2022, for performing deep mutational scanning of full-length SARS-CoV-2 spike proteins, to two important SARS-CoV-2 genomic variants, BA.2 and XBB.1.5, and further develop it to extend the phenotypes that it can measure. Using these two variants as backgrounds allows them to effectively cover the ancestors of the major variants that circulated in 2023 (and in 2024 to date,

though inevitably circulating virus is now quite diverged from these ancestors). The authors provide phenotyping in several different dimensions, and then use genomic surveillance data from circulating SARS-CoV-2 sequences to measure the biological relevance of these screening data from the laboratory, finding that the screen data have significant power to explain evolution in the world. The combined analysis provides a tremendous wealth of data for those studying SARS-CoV-2's ongoing evolution, and the authors also develop important new tools that can be applied to future variants and other viruses. There is a very substantial amount of work described, bringing together deep mutational scanning, in different lineages and different contexts, with genomic surveillance data and the recently-developed analytical approach of mass photometry.

This approach to performing deep mutational scanning across the entire spike protein remains novel and impactful. The most important outcome of this work is the dataset produced, which will be well-used by the community. This dataset allows any researcher coming across a new lineage to assess whether the mutations it possesses are (on their own) likely to alter cell entry properties, to alter ACE2 binding, or to evade immunity induced by vaccination and infection. The authors' approach to documenting their analytical workflows is exemplary, with notebooks available to reproduce every step of the analytical workflows, and outputs available in convenient interactive forms and data files. The availability and documentation of code and workflows is at the very highest level for any analyses in this area, and to be commended. The Github URLs liberally distributed through the text are very helpful to the reader seeking additional data on any point.

We thank the reviewer for the summary, and are glad that (s)he finds the new data and associated ways of interrogating it informative for understanding SARS-CoV-2 evolution.

The assay the authors are using measures the efficiency with which pseudoviruses of different genotypes can enter cells, but they exploit the fact that the neutralisation of this entry by exogenous ACE2 differs according to the strength with which each pseudovirus binds to ACE2 to assess the ACE2-binding of each mutant in their pool. The authors give some examples of the insights that this approach provides, including sites such as A222V, which emerged repeatedly in Delta and which causes increased ACE2 binding despite not being found in the spike RBD.

We agree that the way that our data explains the benefits of mutations like A222V, and now more recently T572I in JN.1 descendant clades, is a major strength. Our paper has highlighted the significant evolutionary importance of mutations that affect ACE2 binding by altering the up-down conformation of the RBD. Note that in the revised manuscript we have added additional mass photometry data biophysically validating these observations.

The utility of the dataset as a whole is brought home by the authors' demonstration using genomic surveillance data that these mutational-scanning data alone are sufficient to predict (imperfectly, as they acknowledge) the change in clade growth rate between a parent clade and its descendant, created by the mutations that it acquired. The analysis here is careful and elegant, and is a useful contribution in providing benchmarking data against other approaches (which here are less effective) and as a benchmark for future approaches.

The main limitation of the deep mutation scanning approach, acknowledged in the Discussion, is that it looks (in practice) at each mutation in turn, while epistasis appears to also play a substantial role in the paths SARS-CoV-2 takes through its fitness landscape. Nevertheless, these analyses show that mutations analysed in isolation can still tell us a great deal.

We fully agree that (as pointed out by reviewers 1 and 2) although our experimental measurements are highly informative for understanding evolution, the modeling does not account for more complex factors like epistasis. We are glad that the reviewer appreciates the impact of our experimental data on its own for its inherent predictive power. As described in the response to reviewers 1 and 2 above, the future of this field will be to integrate powerfully informative data like that described in our paper with more advanced models like the ones being developed by others. This is obviously a long-term goal, and we think our paper represents a significant step in that direction.

The manuscript is easy to read and carefully written. All conclusions are robust and well-supported. Clarity has been prioritised and overstatement avoided. Statistical analysis is sound, figures are clear, and I find nothing of any substance to fault. I list some small stylistic suggestions below that occurred to me during my reading.

- "mutations observed at an appreciable number of times" - I found the 'at' here unexpected to my ear.

The reviewer is correct, and we have removed the "at".

- I was initially somewhat surprised by the authors' decisions to bias their libraries so much in favour of mutations that have been observed occurring multiple times in GISAID, given that it is also interesting to understand why mutations are selected against. Reading their original methods paper gave me a clearer idea of the rationale (a high proportion of deleterious mutations can be problematic when many of the pseudoviruses carry multiple mutations). It could be worth reiterating this reasoning in the methods.

This is a good point, and was also raised by Reviewer 1. As described in the response to major comment 6 of Reviewer 1, we have added additional analyses showing that this library design strategy still effectively covered all mutations that have been relevant not only to XBB-descended clades but also in clades of the more recent JN.1 lineage. We have added additional text and explanation in the methods as suggested by the reviewer.

- "highly correlated between the replicate libraries for each spike, indicating the experiments have good precision (Extended Data Fig. 1e)." - 'precision' can have a technical connotation of relating to the degree of resolution in an assay, (i.e. something like the number of decimal places). Duplicates would not speak directly to that, and regardless here the data in the figure looks very bimodal, so repeatability or accuracy might be a better word?

This is a good suggestion, and we have changed this text to read "repeatability."

Reviewer Reports on the First Revision:

Referees' comments:

Referee #1 (Remarks to the Author):

In the revised manuscript, the authors have addressed most of the concerns. The additional analyses further demonstrate the strength of the proposed XBB.1.5 pseudovirus-based DMS in explaining and predicting SARS-CoV-2 Spike evolution. Regarding the authors' responses and the modifications to the figures and manuscript, I suggest the following additional points to improve the clarity, soundness, and significance of the manuscript:

1. Some ACE2-distal mutations significantly affect ACE2 binding. This could be attributed to the fact that your measurements, based on pseudovirus libraries, actually detected a combination of mutation impacts on both cell entry (contributed by impacts on RBD up/down conformation, fusion, S1 shedding, and other factors) and RBD-ACE2 binding (the inhibition activity is related to both the internal ACE2 binding capability and the infection efficiency), which could hardly be avoided despite the usage of VSV-G or RDPro standard. Is it useful and possible to deconvolute them according to your initial DMS results on cell entry? This analysis may improve your correlation between pseudovirus and yeast display-based DMS on ACE2 binding. If this deconvolution is difficult, I suggest using a term like "apparent ACE2 binding" when first introducing this method, and emphasizing its differences from real monomeric RBD-ACE2 binding affinity (including RBD yeast display-based DMS results).
2. Fig 5c and Extended Data Fig 9 show that many G504 mutations are sensitizing and enhance ACE2 binding, which is inconsistent with the XBB.1.5 RBD yeast display-based DMS results from T. Starr Lab (https://tstarrlab.github.io/SARS-CoV-2-RBD_DMS_Omicron-XBB-BQ/RBD-heatmaps/). This site is not mentioned in the text related to Figure 4, where sensitizing mutations are discussed, either. Most sensitizing mutations on the RBD are on Omicron-mutated residues (especially reversion to WT) and could be well explained by the common WT vaccination history. However, it is doubtful why mutations on G504, which is a conserved site in almost all existing variants and even sarbecoviruses, are sensitizing. G504 should also be a critical site of a recently approved prophylactic NAb, VYD222, further rendering the importance of additional investigation and discussion on it. Does this phenomenon vary among different human serum samples?
3. In Extended Data Fig. 10, why are the escape scores in non-RBD region not shown for the mAb SC27?
4. Are the escape scores in Fig. 5a-c average values over different sera samples? Please specify it in the Figure legend.
5. Please further explain the relationship between the effects on spike fusion, S1 shedding, and ACE2 binding, to make the text more accessible to readers who are not familiar with the SARS-CoV-2 field (Page 5, Lines 1-2).

Referee #2 (Remarks to the Author):

The revised manuscript by Dadonaite and colleagues establishes deep mutational scanning as an experimental pathway generating data for evolutionary predictions. Thanks to the authors for the thorough revision and detailed responses to the referees' queries. The major points raised included a more detailed discussion on the deep mutational scanning experiments, a better explanation of growth estimates and other aspects of their validation by comparison with relative growth differences between parent and descendant variants.

The authors have carefully addressed these points. The revised version expands explanations of how the library was constructed and how potential technical experimental constraints could impact observed properties, such as the reduction in ACE2 binding associated with S2 changes. The authors also extend their application to other variants, including growth in BA.2, BA.5 and XBB descendant clades, and an application to the emergence of BA.2.86/JN.1. Growth estimates are explained better, and it now becomes clear that the section on predictions is adequate to validate the power of DMS data. Here, I would ask the authors to formulate the statement "This approach ... answers the question of real evolutionary interest." more precisely. As I read it, it answers the question of key evolutionary interest for this work: how new mutations change the growth of variants in the actual strain population. There are other questions of real evolutionary interest, involving the competition between clades with different genetic backgrounds, which the authors' approach does not address (and does not need to for its purpose). With this clarification, the paper looks ready for publication to me.

Referee #3 (Remarks to the Author):

The authors have addressed my minor comments. My assessment of the manuscript remains, as previously, very positive.

Referee #3 (Remarks on code availability):

I have looked through the supplied scripts and notebooks which are of high quality, and offer a high degree of reproducibility compared to typical papers in the field. I have not personally run the code.

Referees' comments:

Referee #1 (Remarks to the Author):

In the revised manuscript, the authors have addressed most of the concerns. The additional analyses further demonstrate the strength of the proposed XBB.1.5 pseudovirus-based DMS in explaining and predicting SARS-CoV-2 Spike evolution. Regarding the authors' responses and the modifications to the figures and manuscript, I suggest the following additional points to improve the clarity, soundness, and significance of the manuscript:

1. Some ACE2-distal mutations significantly affect ACE2 binding. This could be attributed to the fact that your measurements, based on pseudovirus libraries, actually detected a combination of mutation impacts on both cell entry (contributed by impacts on RBD up/down conformation, fusion, S1 shedding, and other factors) and RBD-ACE2 binding (the inhibition activity is related to both the internal ACE2 binding capability and the infection efficiency), which could hardly be avoided despite the usage of VSV-G or RDPro standard. Is it useful and possible to deconvolute them according to your initial DMS results on cell entry? This analysis may improve your correlation between pseudovirus and yeast display-based DMS on ACE2 binding. If this deconvolution is difficult, I suggest using a term like “apparent ACE2 binding” when first introducing this method, and emphasizing its differences from real monomeric RBD-ACE2 binding affinity (including RBD yeast display-based DMS results).

Thank you for this suggestion. As suggested by the reviewer, we have edited the manuscript to use the term “apparent ACE2 binding” when first introducing the method and further emphasized that we are not measuring one-to-one ACE2-RBD binding in these assays but instead a combined outcome of several phenotypes.

We initially also wondered whether the effects of mutations on cell entry might have some correlation with ACE2 binding; however as is shown in Extended Data 2c of the manuscript (which is pasted below in this response), we do not see any substantial correlation between these phenotypes. This is perhaps not really surprising because the “cell entry” phenotype is really a combination of the ability of spike to fold correctly and utilize ACE2 for entry. So even if a mutation allows for favorable ACE2 binding, if its effect

on spike folding is highly deleterious it will have a bad effect on cell entry too. It could also be the case that in the context of excess ACE2 (such as infection of ACE2 overexpressing cell lines) ACE2 binding is not really a limiting factor for virus cell entry.

In addition, as noted in the previous round of reviewer response, we think that at least some of the imperfect correlation between RBD yeast-display and full-spike measures of ACE2 binding is that the former strictly measures the one-to-one affinity of the RBD to ACE2, whereas the latter is also affected by quaternary motions in spike—as shown by the fact that the worst correlation is for ACE2-distal RBD residues that are near interfaces of the RBD with other spike domains.

In addition, our mass photometry measurements provide strong evidence that ACE2-distal mutations affect ACE2 binding independent of their effects on cell entry. For example mutations Q115K, A222M and F371N – all of which increase overall spike binding to ACE2 as evidenced by increased RBD occupancy measured by mass photometry (see Figure 3b in the manuscript) – also all have deleterious effects on cell entry (entry scores of -1.59, -0.53 and -0.53, respectively). Furthermore, other mutations that we find to increase ACE2 binding but significantly decrease cell entry – such as A570D – have also been independently shown to lead to changes in RBD movement using biophysical measurements (Ke et al. 2023, DOI:10.1101/2023.12.21.572824).

2. Fig 5c and Extended Data Fig 9 show that many G504 mutations are sensitizing and enhance ACE2 binding, which is inconsistent with the XBB.1.5 RBD yeast display-based DMS results from T. Starr Lab (https://tstarrlab.github.io/SARS-CoV-2-RBD_DMS_Omicron-XBB-BQ/RBD-heatmaps/). This site is not mentioned in the text related to Figure 4, where sensitizing mutations are discussed, either. Most sensitizing mutations on the RBD are on Omicron-mutated residues (especially reversion to WT) and could be well explained by the common WT vaccination history. However, it is doubtful why mutations on G504, which is a conserved

site in almost all existing variants and even sarbecoviruses, are sensitizing. G504 should also be a critical site of a recently approved prophylactic NAb, VYD222, further rendering the importance of additional investigation and discussion on it. Does this phenomenon vary among different human serum samples?

The reviewer makes a good point about the G504 site. We think the effects on ACE2 binding for this site don't correlate well between RBD yeast-display and our full-spike DMS because G504 falls into the "gray" area of affecting **both** direct RBD-ACE2 binding and RBD up/down movement. Site G504 does not directly interact with ACE2 itself but is between residues 503 and 505 both of which interact with ACE2. For this reason, mutations to G504 probably often modestly reduce the one-to-one binding of the RBD to ACE2 by altering the conformation of adjacent ACE2-contact residues 503 and 505, as reflected in the RBD yeast-display from the Starr lab.

In addition, in the RBD-down conformation, site G504 is directly facing other RBD protomers (see image below) and therefore mutations at this site likely also impact RBD up-down movement. Mutations G504I/V/C are highly deleterious for ACE2 binding in both RBD yeast-display and our full-spike DMS, which probably indicates that these mutations directly reduce the one-to-one binding of the RBD with ACE2. But other mutations at site 505 do not correlate well between the RBD yeast-display and the full-spike DMS, but do exhibit a strong negative correlation between ACE2 binding and serum escape in the full-spike DMS, suggesting these mutations probably modulate RBD up-down conformation more than they directly affect one-to-one RBD-ACE2 binding.

In response to the question at the end of the reviewer's comment, mutations at site G504 similarly affect neutralization by all the sera we tested. Such consistent escape patterns also hint at this site being involved in modulating RBD movement as opposed to direct serum escape, since it is unlikely all sera would contain neutralizing antibodies targeting this specific residue, but it likely that all sera contain antibodies that can better neutralize spike when the RBD is in the up conformation.

The reviewer makes a good observation that the fact that most mutations to G504 increase serum neutralization is a good sign for antibodies like VYD222, which have an epitope that includes site G504 (spans residues 501-508 (Yuan et al. 2022, DOI:10.1073/pnas.2205784119)). The reason that this is a good sign is that many mutations at site G504 might come at a cost to serum neutralization if they put the RBD in a more up conformation. However, some mutations at G504, including G504F are reasonably tolerated for both cell entry and ACE2 binding suggesting that mutations at this site in future variants are possible.

3. In Extended Data Fig. 10, why are the escape scores in non-RBD region not shown for the mAb SC27?

That's because this antibody was mapped with the pseudovirus XBB.1.5 library that contained all RBD mutations but no other mutations, meaning we could not measure escape outside the RBD. We have described this in the legend but we have now added a note in the actual figure to further emphasize this point.

4. Are the escape scores in Fig. 5a-c average values over different sera samples? Please specify it in the Figure legend.

Yes these are escape values averaged over all 10 sera. We have clarified that in the figure legend. Mutation-level per sera escape can be found under 'antibody escape mutation effect plots' section at https://dms-vep.org/SARS-CoV-2_XBB.1.5_spike_DMS/.

5. Please further explain the relationship between the effects on spike fusion, S₁ shedding, and ACE2 binding, to make the text more accessible to readers who are not familiar with the SARS-CoV-2 field (Page 5, Lines 1-2).

We have added additional clarification to the main text. Specifically, spike-mediated fusion occurs after ACE2-binding and subsequent proteolytic cleavage initiates the shedding of spike's S₁ domain. If some mutations to spike increase the propensity for ACE2 binding to lead to S₁ shedding, then in our ACE2-neutralization assay those mutations would be measured as increasing ACE2 binding because they increase the efficiency with which soluble ACE2 can irreversibly inactivate spike. The extent to which mutations are having this effect is unclear, but it is a possibility.

Referee #2 (Remarks to the Author):

The revised manuscript by Dadonaite and colleagues establishes deep mutational scanning as an experimental pathway generating data for evolutionary predictions. Thanks to the authors for the thorough revision and detailed responses to the referees' queries. The major points raised included a more detailed discussion on the deep mutational scanning experiments, a better explanation of growth estimates and other aspects of their validation by comparison with relative growth differences between parent and descendant variants.

The authors have carefully addressed these points. The revised version expands explanations of how the library was constructed and how potential technical experimental constraints could impact observed properties, such as the reduction in ACE2 binding associated with S2 changes. The authors also extend their application to other variants, including growth in BA.2, BA.5 and XBB descendant clades, and an application to the emergence of BA.2.86/JN.1. Growth estimates are explained better, and it now becomes clear that the section on predictions is adequate to validate the power of DMS data. Here, I would ask the authors to formulate the statement "This approach ... answers the question of real evolutionary interest." more precisely. As I read it, it answers the question of key evolutionary interest for this work: how new mutations change the growth of variants in the actual strain population. There are other questions of real evolutionary interest, involving the competition between clades with different genetic backgrounds, which the authors' approach does not address (and does not need to for its purpose). With this clarification, the paper looks ready for publication to me.

Thank you for this suggestion, we have modified the statement to emphasize that our work only focuses on a question addressing the effects of mutations on clade growth and other questions not addressed by our study are also often of evolutionary interest.

Referee #3 (Remarks to the Author):

The authors have addressed my minor comments. My assessment of the manuscript remains, as previously, very positive.

Referee #3 (Remarks on code availability):

I have looked through the supplied scripts and notebooks which are of high quality, and offer a high degree of reproducibility compared to typical papers in the field. I have not personally run the code.

Thank you for a positive review.

Full-spike deep mutational scanning helps predict the evolutionary success of SARS-CoV-2 clades

Bernadeta Dadonaite¹, Jack Brown², Teagan E McMahon¹, Ariana G Farrell¹, Marlin D Figgins^{3,4}, Daniel Asarnow², Cameron Stewart², Jimin Lee², Jenni Logue⁵, Trevor Bedford^{3,6,8}, Ben Murrell⁷, Helen Y. Chu⁵, David Veessler^{2,8}, Jesse D Bloom^{1,8,#}

¹Basic Sciences Division and Computational Biology Program, Fred Hutchinson Cancer Center, Seattle, Washington, 98109, USA

²Department of Biochemistry, University of Washington, Seattle, Washington, USA

³Vaccine and Infectious Disease Division, Fred Hutchinson Cancer Center, Seattle, WA, USA

⁴Department of Applied Mathematics, University of Washington, Seattle, WA, USA

⁵University of Washington, Department of Medicine, Division of Allergy and Infectious Diseases, Seattle, WA

⁶Department of Epidemiology, University of Washington, Seattle, WA, USA

⁷Department of Microbiology, Tumor and Cell Biology, Karolinska Institutet, Stockholm, Sweden

⁸Howard Hughes Medical Institute, Seattle, WA, 98195, USA

#Lead Contact jbloom@fredhutch.org

Abstract

SARS-CoV-2 variants acquire mutations in spike that promote immune evasion and impact other properties that contribute to viral fitness such as ACE2 receptor binding and cell entry. Knowledge of how mutations affect these spike phenotypes can provide insight into the current and potential future evolution of the virus. Here we use pseudovirus deep mutational scanning to measure how >9,000 mutations across the full XBB.1.5 and BA.2 spikes affect ACE2 binding, cell entry, or escape from human sera. We find that mutations outside the receptor-binding domain (RBD) have meaningfully impacted ACE2 binding during SARS-CoV-2 evolution. We also measure how mutations to the XBB.1.5 spike affect neutralization by serum from individuals who recently had SARS-CoV-2 infections. The strongest serum escape mutations are in the RBD at sites 357, 420, 440, 456, and 473—however, the antigenic impacts of these mutations vary across individuals. We also identify strong escape mutations outside the RBD; however many of them decrease ACE2 binding, suggesting they act by modulating RBD conformation. Notably, the growth rates of human SARS-CoV-2 clades can be explained in substantial part by the measured effects of mutations on spike phenotypes, suggesting our data could enable better prediction of viral evolution.

Introduction

Over the last four years of SARS-CoV-2 evolution, the virus has accumulated mutations throughout its genome. The most rapid evolution has occurred in the viral spike: for instance, the XBB-descended variants that dominated in 2023 have 45–48 spike protein mutations relative to the earliest known strains from Wuhan in late 2019. The reason for this rapid evolution is that spike mutations can strongly affect both the virus's inherent transmissibility and ability to escape

pre-existing immunity¹⁻³. A crucial aspect of interpreting and forecasting SARS-CoV-2 evolution is therefore understanding the impact of current and potential future mutations to spike.

Here we measure how thousands of mutations to the spike glycoprotein of the XBB.1.5 and BA.2 SARS-CoV-2 strains impact three molecular phenotypes critical to viral evolution: cell entry, ACE2 binding, and neutralization by human polyclonal serum (**Fig. 1a**). To do this, we extend a recently described pseudotyped lentivirus deep mutational scanning system⁴ that enables safe experimental characterization of mutations throughout the spike^{5,6}. We demonstrate that mutations outside the RBD can substantially impact spike binding to ACE2. We also define the mutations that escape neutralization by sera from humans who have been multiply vaccinated and also recently infected by XBB or one its descendant lineages (XBB*), and show there is appreciable heterogeneity in the antigenic impact of mutations across individuals. Finally, we show that the spike phenotypes we measure explain much of the changes in viral growth rate among different SARS-CoV-2 clades that have emerged in humans over the last few years.

Design of deep mutational scanning libraries

We created mutant libraries of the spikes from the XBB.1.5 and BA.2 strains. We chose these strains because nearly all currently circulating human SARS-CoV-2 descends from either BA.2 or XBB.1.5's parent lineage XBB⁷, and because XBB.1.5 is the sole component of the COVID-19 booster vaccine recommended by the WHO in 2023⁸. We wanted the libraries to contain all evolutionary accessible amino-acid mutations tolerable for spike function. We therefore included all mutations observed an appreciable number of times among the millions of SARS-CoV-2 sequences in GISAID. In addition, we included all possible mutations at sites that change often during SARS-CoV-2 evolution or are antigenically important^{3,9}, and deletions at key NTD and RBD sites. These criteria led us to target ~7,000 amino-acid mutations in each of the XBB.1.5 and BA.2 libraries (**Extended Data Fig. 1a**). We created two independent libraries for each spike so we could perform all deep mutational scanning in full biological duplicate. The actual libraries contained between 69,000 and 102,000 barcoded spike variants with an average of 2 mutations per variant, and successfully covered 99% of the targeted mutations as well as some additional mutations (**Extended Data Fig. 1a**). To retrospectively validate that this library design strategy covered most evolutionarily important mutations, we confirmed that our XBB.1.5 libraries provided adequate coverage for high-confidence experimental measurements of nearly all spike mutations currently present in XBB, BA.2, and BA.2.86 descended Pango clades—despite the fact that BA.2.86 had not even emerged yet at the time we designed the library (**Extended Data Fig. 1b**). So while our libraries do not contain all spike mutations, in practice they cover nearly all mutations that are relevant in the near- to mid-term evolution of SARS-CoV-2. Because the RBD is an especially important determinant of ACE2 binding and serum antibody escape^{2,10}, we also made duplicate XBB.1.5 libraries that saturated all amino-acid mutations in the RBD only (**Extended Data Fig. 1a**).

Effects of spike mutations on cell entry

We measured the effects of all library mutations on spike-mediated cell entry in 293T-ACE2 cells⁴ (**Extended Data Fig. 1c-d** and interactive heatmaps at https://dms-vep.github.io/SARS-CoV-2_XBB.1.5_spike_DMS/htmls/293T_high_ACE2_entry_func_effects.html and https://dms-vep.github.io/SARS-CoV-2_Omicron_BA.2_spike_ACE2_binding/htmls/293T_high_ACE2_entry_func_effects.html). These measurements were highly correlated between the replicate libraries for each spike, indicating the experiments have good repeatability (**Extended Data Fig. 1e**). The effects of mutations were also well correlated between the XBB.1.5 and BA.2 spikes (**Extended Data Fig. 1f**), consistent with prior reports that most but not all mutations have similar impacts on the spikes of different SARS-CoV-2 variants^{11,12}. As expected, stop codons were highly deleterious for cell entry (**Fig. 1b**). Because our full spike library design strategy favors functionally tolerated mutations in spike, most amino-acid mutations in our libraries just slightly impaired cell entry, and some but not all single-residue deletions were also well tolerated (**Fig. 1b**). As expected, mutation effects in saturated RBD library had a substantially more deleterious distribution (**Extended Data Fig. 1g**). SARS-CoV-2 has acquired numerous deletions in the NTD's flexible loops during its evolution^{13,14}, and consistent with that evolution we find that the flexible loops but not the core β -sheets of the NTD are relatively tolerant of deletions (**Extended Data Fig. 1h**). Overall, the effects of mutations on cell entry were fairly well correlated with the effects of amino-acid mutations on viral fitness estimated from millions of natural human SARS-CoV-2 sequences (**Extended Data Fig. 1i**)¹⁵.

No individual mutation in either the XBB.1.5 or BA.2 spikes dramatically increased pseudovirus cell entry, though some mutations did marginally improve entry (**Fig. 1b** and interactive heatmaps linked in figure legend). One mutation that slightly improves pseudovirus entry in both XBB.1.5 and BA.2 is P1143L (**Fig. 1c**), which is found in the recently emerged BA.2.86 lineage¹⁶. We previously reported that mutations to P1143 also improve cell entry for BA.1 and Delta pseudoviruses⁴. The deletion mutations in our libraries are usually more deleterious for cell entry than substitutions (**Fig. 1b**); however, deletion of V483 in the RBD is well tolerated for cell entry, consistent with emergence of this mutation in the BA.2.86 variant¹⁶. The F456L mutation, which has emerged repeatedly in XBB clades after being rare in earlier BA.2-derived clades, is well tolerated for cell entry in XBB.1.5 but substantially deleterious in BA.2 (**Fig. 1c**).

Both RBD and non-RBD mutations affect spike binding to ACE2

To measure how mutations in spike affect receptor binding, we leveraged the fact that the soluble ACE2 ectodomain neutralizes spike-mediated infection with a potency proportional to the strength of spike binding to ACE2^{3,17}. Specifically, soluble ACE2 more potently blocks entry by spikes with mutations that increase binding to ACE2. To validate this fact, we made pseudoviruses with six different spike variants and quantified their neutralization by monomeric ACE2 (**Fig. 2a**). Compared to the BA.2 spike, the Wuhan-Hu-1 + D614G spike is neutralized less potently by soluble ACE2 consistent with its weaker ACE2 binding^{18,19}, whereas four mutants of BA.2 known to have higher ACE2 binding²⁰ (N417K, N417F, R493Q, and Y453F) were all neutralized more potently by soluble

ACE2 (**Fig. 2a**). The quantitative neutralization by soluble ACE2 was highly correlated with previously measured monomeric RBD ACE2 affinities^{19–21} (**Fig. 2b**).

Using this approach, we measured how mutations across both the XBB.1.5 and BA.2 spikes affect **apparent** ACE2 binding (**Fig. 2c** and interactive plots at https://dms-vep.github.io/SARS-CoV-2_XBB.1.5_spike_DMS/htmls/monomeric_ACE2_mut_effect.html and https://dms-vep.github.io/SARS-CoV-2_Omicron_BA.2_spike_ACE2_binding/htmls/monomeric_ACE2_mut_effect.html).

Because our assay **measures** ACE2 neutralization **rather than 1:1 ACE2-RBD affinity** there are several distinct mechanisms that could affect what we refer to as ACE2 binding: direct changes in 1:1 RBD-ACE2 binding affinity^{20,22}, changes in spike that modulate the conformation of the RBDs (such as up/down movements)^{23,24}, and ACE2-induced shedding of the S₁ subunit (**ACE2 binding leads to shedding of the S₁ domain from spike, and any mutations that increase the propensity for S₁ shedding might promote neutralization of pseudovirus by soluble ACE2 in our assay**)^{25,26}.

Because our assay **measures** ACE2 neutralization **rather than 1:1 ACE2-RBD affinity** there are several distinct mechanisms that could affect what we refer to as ACE2 binding: direct changes in 1:1 RBD-ACE2 binding affinity^{20,22}, changes in spike that modulate the conformation of the RBDs (such as up/down movements)^{23,24}, and ACE2-induced shedding of the S₁ subunit (**ACE2 binding leads to shedding of the S₁ domain from spike, and any mutations that increase the propensity for S₁ shedding might promote neutralization of pseudovirus by soluble ACE2 in our assay**)^{25,26}.

The effects of RBD mutations on ACE2 binding to the spike measured using pseudovirus deep mutational scanning correlate well with previously reported measurements from RBD yeast-display for both XBB.1.5 and BA.2 (**Fig. 2d**)²². We also measured ACE2 binding for the XBB.1.5 pseudovirus libraries with saturating RBD mutations using both monomeric and dimeric soluble ACE2. The RBD-only pseudovirus measurements were highly correlated with the full-spike library measurements (**Extended Data Fig. 2a**), and the measured values were highly similar for monomeric versus dimeric soluble ACE2 (**Extended Data Fig. 2b**). Importantly, ACE2 binding and pseudovirus cell entry are distinct properties, with no strong correlation between these properties among tolerated mutations (**Extended Data Fig. 2c**) – likely reflecting the fact that cell entry can be limited by factors unrelated to receptor binding, especially in target cells expressing moderate to high levels of ACE2 like those used in our experiments.

A striking observation from the deep mutational scanning is that some mutations outside the RBD appreciably affect binding to ACE2 (**Fig. 2c,e** and **Extended Data Fig. 2d**). To validate these findings, we used mass photometry to measure binding of the soluble native ACE2 dimer to the spike ectodomain trimer (**Fig. 3a**). Mass photometry measures protein-protein interactions in solution by detecting changes in light scattering that are proportional to protein molecular mass²⁷, which allows us to detect binding of one or more ACE2 molecules to the spike (**Fig. 3a**). We produced prefusion-stabilized HexaPro²⁸ BA.2 and XBB.1.5 spikes, along with mutants that our deep mutational scanning experiments showed to modulate ACE2 binding, and performed mass photometry in the presence of a series of ACE2 concentrations (**Fig. 3a-b, Extended Data Figs. 3-6**). As expected, we observed better and worse ACE2 binding for RBD mutations that have been previously identified to either increase (R493Q) or abrogate (R498V) ACE2 engagement, respectively²⁰ (**Fig. 3b**, left panels). Furthermore, we detected increased ACE2 binding for all but one of the BA.2 and XBB.1.5 spike trimers harboring S₁ subunit mutations (in NTD, RBD, and SD1 domains) that our deep mutational scanning indicated had better binding (**Fig. 3b** middle panel, **Extended Data Figs. 3-6**) as well as decreased ACE2 binding for NTD and S₂ mutations that our deep mutational scanning indicated had worse binding (**Fig. 3b**). However, mutations to the BA.2

and XBB1.5 S₂ subunit found to increase binding to ACE2 in our deep mutational scanning did not lead to increased ACE2 binding detectable by mass photometry (**Fig. 3b** right panel, **Extended Data Figs. 3, 4** and **6b-c**). Notably, some of these S₂ mutations were previously reported to affect spike fusion²⁹⁻³¹ suggesting that they may indeed affect S₁ shedding and in turn affect ACE2 binding consistent with our deep mutational scanning. However, unlike the spikes in deep mutational scanning experiments, the spikes used in mass photometry experiments are pre-fusion stabilized by introduction of the HexaPro mutations in the fusion machinery²⁸. These modifications to spike may limit the propagation of long-range allosteric changes induced by S₂ subunit mutations, possibly explaining the discrepancy between deep mutational scanning and mass photometry. Concurring with this hypothesis, we previously showed that ACE2-induced allosteric conformational changes that drive fusion peptide exposure were inhibited by the prefusion-stabilizing 2P mutations³².

Non-RBD mutations that enhance ACE2 binding have played an important role in SARS-CoV-2 evolution. The following non-RBD mutations that enhance ACE2 binding occurred in major pre-Omicron variants of concern: A570D (Alpha), A222V (several moderate-frequency Delta sublineages), T1027I (Gamma), and D950N (Delta) (**Extended Data Fig. 2d**). In addition, the following non-RBD mutations that occurred in Omicron variants, all of which represent reversions to pre-Omicron residue identities, increase ACE2 binding: K969N, K764N and Y655H. Consistent with prior studies showing that the original D614G mutation increased the proportion of RBDs in the up conformation^{23,33,34}, we find that G614D decreases full spike ACE2 binding (**Fig. 3b**, **Extended Data Fig. 2d**).

To systematically examine the recent evolutionary role of non-RBD ACE2 binding-enhancing mutations, we tabulated non-RBD mutations that enhance binding and are new mutations in at least four XBB-descended Pango clades (**Fig. 3c**). Some of these mutations may explain why certain clades had a growth advantage. For example, the NTD mutation Q52H provided the EG.5.1 lineage with a clear growth advantage over EG.5³⁵, despite not measurably affecting serum neutralization³⁶. Our deep mutational scanning provides an explanation for the success of EG.5.1 by showing that Q52H enhances ACE2 binding. Similarly, the T572I mutation is now appearing convergently in JN.1-descended lineages, and our results show that mutation enhances ACE2 binding³⁷. Overall, these results suggest that non-RBD mutations that affect ACE2 binding play an important role in SARS-CoV-2 evolution.

Mapping escape from XBB* infection sera reveals heterogeneity among individuals

We next mapped how mutations in spike affect neutralization by the polyclonal antibodies in sera from 10 vaccinated individuals who either had a confirmed XBB* infection or whose last infection was during a period when XBB lineages were the dominant circulating variants (**Supplementary Table 1**). We performed these measurements with the full spike XBB.1.5 libraries using 293T cells expressing moderate levels of ACE2 that better capture the activities of non-RBD antibodies^{38,39},

although the key sites of escape were mostly similar if we used 293T cells expressing high levels of ACE2 or the RBD-only libraries (**Extended Data Fig. 7**). The sites of greatest serum escape were mainly in the RBD (**Fig. 4a-c** and interactive plot at https://dms-vep.github.io/SARS-CoV-2_XBB.1.5_spike_DMS/htmls/summary_overlaid.html). These sites include 357, 371, 420, the 440-447 loop, 455-456, and 473, as well as a few sites in the NTD, such as positions 200 and 234. At some sites, the escape mutations are strongly deleterious to ACE2 binding (**Fig. 4c**). For instance, mutations at Y473 cause strong neutralization escape but greatly reduce ACE2 binding, likely explaining their low frequency among circulating SARS-CoV-2 variants. In addition, only some of the antibody-escape mutations mapped in our experiments are accessible by single-nucleotide mutations to XBB.1.5 (**Fig. 4c**). Several escape mutations that are single-nucleotide accessible and do not strongly impair ACE2 binding are found in recent variants, including mutations at site 456 in EG.5.1 and many other XBB variants, mutations at 455 in HK.3.1 and JN.1, mutations at 420 in GL.1, and mutations at 200 in XBB.1.22^{7,35}.

While the same mutations often escape many sera, there is also heterogeneity such that the sera-average is not fully reflective of the impacts of mutations on any individual serum (**Fig. 4b,d** and **Extended Data Fig. 8**). For example, while mutations to site Y473 strongly escape neutralization by most sera, two sera we analyzed (493C and 501C) are largely unaffected by mutations at that site. Other key sites of escape, including 420 and 456, show similar heterogeneity across sera. To validate that escape mutations can have very different effects across sera, we performed standard pseudovirus neutralization assays⁵ against a panel of point mutants to the XBB.1.5 spike (**Fig. 4d**). The changes in neutralization in these validation assays were highly correlated with the escape measured by deep mutational scanning, and confirmed the serum-to-serum heterogeneity. For example, Y473S strongly reduces neutralization by sera 287C and 500C, but actually slightly increases neutralization by serum 501C. Similarly, F456L substantially reduces neutralization by only some sera (**Fig. 4d**).

The deep mutational scanning identifies mutations that increase as well as escape neutralization (**Extended Data Fig. 9**). Sensitizing mutations often occur at sites that are mutated in XBB.1.5 relative to earlier variants, such as sites 373, 405, 417, 460, 486 and 505 (**Extended Data Fig. 9**). Presumably in many cases, reverting mutations at these sites restores neutralization by antibodies elicited by infection or vaccination with earlier viral strains. To confirm that the sensitizing mutations identified in the deep mutational scanning actually increased neutralization, we validated the sensitizing effects of R403K and N405K in standard pseudovirus neutralization assay (**Fig. 4d**). In addition, some sensitizing mutations appear to act by placing the RBD in a more up conformation as discussed in the next subsection.

Some mutations that strongly affect neutralization modulate RBD conformation rather than directly affecting antibody binding

Most sites of strong escape described in the previous section are proximal to the ACE2-binding motif in the RBD that is the target of many potent neutralizing antibodies^{40,41} (we define

ACE2-proximal RBD residues as those within 15 Å of ACE2 in the RBD-ACE2 crystal structure). However, the deep mutational scanning also reveals individual mutations at non-RBD or ACE2-distal RBD sites that strongly escape neutralization. Some of these sites, such as 42, 200, 234 in the NTD, 572 in SD1, and 852 in S2 have mutations that cause as much escape as ACE2-proximal RBD mutations, decreasing serum neutralization by as much as 6-fold (**Fig. 4a,d**). Whereas most mutations at any given site have similar effects on escape (i.e. either promoting or sensitizing) at many ACE2-proximal RBD sites, different mutations at the same non-RBD or ACE2-distal RBD site can have opposing effects on neutralization (**Fig. 5a-c**). Furthermore, there is a strong correlation between mutational effects on neutralization and ACE2 binding at these sites: mutations that reduce neutralization also reduce ACE2 binding, and mutations that increase neutralization also increase ACE2 binding (**Fig. 5a,b**). No such consistent correlation exists between neutralization and ACE2 binding for RBD escape sites in close proximity of ACE2 binding interface (**Fig. 5c**).

We hypothesize that non-RBD and ACE2-distal RBD mutations that increase both neutralization and ACE2 binding do so by shifting the RBD to a more “up” position, whereas those that decrease neutralization and ACE2 binding do so by shifting the RBD to a more “down” position^{42–44}. Prior work has shown that mutations that put the RBD in a down position reduce neutralization by antibodies that target RBD residues only accessible in the up position, whereas antibodies that can bind both the up and down RBD are unaffected by such mutations^{16,45}. Consistent with this prior work, we confirmed that the mutations at ACE2-distal sites identified in our full-spike deep mutational scanning as likely affecting RBD conformation only affect neutralization by monoclonal antibodies that bind only to the up conformation of the RBD (**Extended Data 10**).

Our results show that mutations that affect neutralization and ACE2 binding by modulating RBD conformation are common in certain regions of spike—a result that makes structural sense, since most of these mutations are located near the interfaces between the RBD and other spike domains (**Fig. 5d, Extended Data Fig. 11**). Furthermore, many of these strong escape sites, including N234, F371, P373, F375, A376, S408, A570, T572, have been previously shown by structural methods to affect RBD conformation^{24,42–44,46–49} or the conformation of key RBD epitopes^{21,50}.

Experimentally measured spike phenotypes partially predict evolutionary success of human SARS-CoV-2 clades

SARS-CoV-2 evolution in humans is characterized by the repeated emergence of new viral clades, which often possess additional amino-acid mutations in spike relative to their predecessors. To test if our deep mutational scanning measurements could help explain which clades are evolutionarily successful, we estimated the relative growth rates in humans of sufficiently-sampled SARS-CoV-2 clades using multinomial logistic regression (**Extended Data Fig. 12a**)⁵¹. As expected, more recent

clades generally had higher growth rates, consistent with evolution selecting for viral clades that are more fit (**Fig. 6a**), presumably in part due to additional mutations in spike⁵².

We sought to determine if the growth of clades could be predicted from how their mutations affect the spike phenotypes measured by deep mutational scanning. Note that almost any mutation-based measurement (such as just counting mutations) trivially correlates with clade growth because newer clades typically have both better growth rates and more spike mutations (**Fig 6a** and **Extended Data Fig. 12b**). For instance, clade growth rates strongly correlate with the number of spike mutations relative to the early Wuhan-Hu-1 sequence (**Extended Data Fig. 12c**). But this correlation is not informative since the question of evolutionary interest is not whether SARS-CoV-2's spike will acquire more mutations over time (we already know this will happen), but rather *which* of the various mutant viruses present at any given time will spread. Furthermore, phylogenetic correlations can exaggerate associations between mutations and clade growth⁵³. Therefore, we focused on predicting *changes* in clade growth for each pair of parent-descendant clades separated by at least one spike mutation (**Fig. 6b**). This approach eliminates the confounding effects of phylogenetic relatedness and the accumulation of mutations over time (**Extended Data Fig. 12c-g**), and better answers **the question of how specific mutations affect clade growth**.

Changes in growth between parent-descendant clade pairs were positively correlated with all three experimentally measured spike phenotypes both among just XBB-descended clades (**Fig. 6c** and **Extended Data Fig. 13**) and among all BA.2, BA.5, and XBB-descended clades (**Extended Data Fig. 14**). The correlations were statistically significant for sera escape and cell entry as assessed by randomization of the measurements among mutations. However, these univariate correlations do not fully capture the information in the experiments, since the effects of mutations on the spike phenotypes are themselves correlated (e.g., mutations that cause sera escape sometimes decrease ACE2 binding). We therefore performed ordinary-least squares multiple linear regression of changes in clade growth versus all three phenotypes. The predictions of this regression correlated with changes in clade growth better than any individual phenotype, and were highly statistically significant as assessed by randomization of the measurements among mutations (**Fig. 6d** and **Extended Data Fig. 14**). Sera escape uniquely explained the largest fraction of the variance in changes in clade growth, but ACE2 binding and cell entry effects also explained some variance. In contrast, neither RBD yeast-display deep mutational scanning of antibody escape^{9,54} and ACE2 affinity²² nor the EVEscape deep learning model⁵⁵ were consistently better than randomized data at predicting changes in clade growth at a significance level of $P = 0.05$ (**Extended Data Figs. 13** and **14**). Overall, these results show that full-spike deep mutational scanning can partially predict the evolutionary success of human SARS-CoV-2 clades, and that its predictive power exceeds that of several other methods.

We also sought to test the ability of full-spike deep mutational scanning to explain the high fitness of BA.2.86 and its descendant clades (e.g., JN.1), which were identified after the completion of the experiments described in this study⁵⁶. Because there are not yet sufficient distinct BA.2.86-descended clades to make meaningful comparisons with clade growth, instead

we performed a different test inspired by Thadani et al⁵⁵: we generated sequences with random sets of naturally observed spike amino-acid mutations that had the same number of differences relative to BA.2 as did BA.2.86, or relative to BA.2.86 as all designated BA.2.86-descended clades. Our XBB.1.5-based full-spike deep mutational scanning could distinguish the true BA.2.86 and BA.2.86-descended clades from sequences with the same number of mutations with high statistical significance, and did so better than RBD yeast-display deep mutational scanning or EVEscape (**Extended Data Fig. 15**).

Discussion

Over 16-million human SARS-CoV-2 genomes have been sequenced to date, enabling rapid identification of variants with new mutations at the sequence level. However, interpreting the consequences of these mutations on viral spread in a partially immune population remains a major challenge. Here we show how pseudovirus-based deep mutational scanning can characterize the effects of mutations throughout spike on three distinct phenotypes critical to viral fitness: cell entry, ACE2 binding, and serum antibody escape.

The full-spike pseudovirus data we generate enables several key insights that were not apparent from prior yeast-display RBD deep mutational scanning approaches^{2,20,54}. Most obviously, the data encompass all spike domains, not just the RBD. Strikingly, these data show that non-RBD mutations can affect ACE2 binding, probably by altering the conformation of the RBD in the context of the spike trimer (e.g., in up versus down position). Such mutations are highly relevant for SARS-CoV-2 evolution—for instance, enhancement of ACE2 binding by non-RBD mutations appears to explain why EG.5.1 spread so rapidly after it acquired Q52H, why A222V subvariants of Delta spread widely, why A570D was selected in Alpha, and why T572I is currently arising so frequently in BA.2.86-descended variants.

Pseudovirus deep mutational scanning also enables us to directly measure how mutations affect neutralization by polyclonal sera. In contrast, prior RBD-display deep mutational scanning could only measure how mutations affect antibody binding², and so to estimate mutational effects on serum neutralization escape it was necessary to characterize hundreds of individual antibodies assumed to represent the polyclonal neutralizing repertoire of humans^{3,9}. The ability to directly map how mutations affect serum neutralization leads to two new insights. First, it reveals the heterogeneity in how mutations affect neutralization by sera from different individuals. For instance, we characterize sera from XBB* infected individuals that are both strongly affected and almost completely unaffected by mutations at key sites like 456 or 473. The sera examined in this study came from individuals with varied immunization and infection histories, which likely contributes to observed escape heterogeneity, although individual-to-individual variation in humoral response may also play a role. This person-to-person heterogeneity in the antigenic effects of spike mutations will increase as individuals accumulate increasingly distinct exposure histories, and could come to play an important role in shaping SARS-CoV-2 evolution and disease susceptibility as it does for influenza virus⁵⁷⁻⁵⁹.

The second major insight from direct mapping of serum escape is that mutations outside the RBD can have marked effects on neutralization. For instance, NTD mutations such as Y42F and N234T decrease neutralization by some sera by nearly 6-fold. The existence of such strong non-RBD escape mutations may seem surprising given that most neutralizing activity in human sera come from antibodies that bind the RBD^{2,10,38,60}. However, our data suggest that the strongest non-RBD serum escape mutations act primarily by shifting the RBD to the down conformation, thereby indirectly escaping class 1 and 4 antibodies that bind to RBD surfaces only accessible in the up conformation^{16,45}. Of course, such mutations come at a cost to ACE2 binding, since the RBD cannot bind receptor in the down conformation^{61,62}. Nonetheless, the ubiquity of such mutations suggests that this mechanism of escape merits monitoring and is in line with prior observations made with endemic human coronaviruses⁶³⁻⁶⁵. For instance, the RBD of the CoV-229E spike has never been observed in the up conformation^{66,67} despite the fact that this spike somehow manages to bind its receptor during infection. Whether SARS-CoV-2's spike could eventually evolve to also much more strongly favor a down RBD conformation is unknown.

The most important indication of the relevance of our work is that our measurements of spike phenotypes partially explain the evolutionary success of different SARS-CoV-2 clades in humans. A longstanding goal of evolutionary biology is to understand the molecular phenotypes that contribute to fitness⁶⁸, and then measure them with sufficient accuracy to predict which mutants will actually spread in the real world. We have taken a real step towards this goal, since the spike phenotypes measured by our deep mutational scanning explain a substantial amount of the changes in growth rates of recent SARS-CoV-2 clades. Of course, pseudovirus spike deep mutational scanning will never perfectly predict SARS-CoV-2 evolution: evolution itself is partially stochastic⁶⁹, pseudovirus experiments do not capture all phenotypes of spike relevant to transmission or multicycle replication, and our experiments completely ignore mutations to non-spike genes that contribute to fitness^{15,70}. Furthermore, it remains technically challenging for deep mutational scanning to account for epistatic interactions among mutations⁷¹, and we need modeling approaches that better account for how person-to-person heterogeneity in immune-escape mutations shape viral evolution⁵⁷. However, the fact that our deep mutational scanning has substantial power to explain clade growth shows that we have reached the point where experiments can enable useful predictions about SARS-CoV-2 evolution. An important area of future work will be integrating these highly informative experimental measurements into even more sophisticated models of viral evolution^{55,72,73}.

Methods

Data accessibility and computer code

The data described in this paper are available in both interactive and numerical form at various levels of detail. For easy interactive visualization of the data, we suggest the following interactive charts of how mutations affect all measured phenotypes after applying a reasonable set of filters to remove lower-confidence measurements:

- XBB.1.5 spike: https://dms-vep.github.io/SARS-CoV-2_XBB.1.5_spike_DMS/htmls/summary_overlaid.html
- BA.2 spike: https://dms-vep.github.io/SARS-CoV-2_Omicron_BA.2_spike_ACE2_binding/htmls/summary_overlaid.html

- XBB.1.5 RBD: https://dms-vep.github.io/SARS-CoV-2_XBB.1.5_RBD_DMS/htmls/summary_overlaid.html

For numerical data on mutational effects on all measured phenotypes after applying the same reasonable set of filters, see:

- XBB.1.5 spike: https://github.com/dms-vep/SARS-CoV-2_XBB.1.5_spike_DMS/blob/main/results/summaries/summary.csv
- XBB.1.5 spike, per-serum escape: https://github.com/dms-vep/SARS-CoV-2_XBB.1.5_spike_DMS/blob/main/results/summaries/per_antibody_escape.csv
- BA.2 spike: https://github.com/dms-vep/SARS-CoV-2_Omicron_BA.2_spike_ACE2_binding/blob/main/results/summaries/summary.csv
- XBB.1.5 RBD: https://github.com/dms-vep/SARS-CoV-2_XBB.1.5_RBD_DMS/blob/main/results/summaries/summary.csv

In addition to the above interactive charts and numerical data, the entire computational pipelines are available along with rich interactive HTML displays of results. These numerical data and HTML displays include additional options to filter the data for higher and lower confidence values, such as by examining the measurements in each of the two replicate libraries or filtering measurements by how many variants a mutation is seen in. Specifically, full interactive HTML documentation for each deep mutational scanning experiment are rendered on GitHub Pages at:

- XBB.1.5 full spike: https://dms-vep.github.io/SARS-CoV-2_XBB.1.5_spike_DMS/
- BA.2 full spike: https://dms-vep.github.io/SARS-CoV-2_Omicron_BA.2_spike_ACE2_binding/
- XBB.1.5 RBD: https://dms-vep.github.io/SARS-CoV-2_XBB.1.5_RBD_DMS/

GitHub repositories with the actual computer code as well as numerical data are at:

- XBB.1.5 spike: https://github.com/dms-vep/SARS-CoV-2_XBB.1.5_spike_DMS
- BA.2 spike: https://github.com/dms-vep/SARS-CoV-2_Omicron_BA.2_spike_ACE2_binding
- XBB.1.5 RBD: https://github.com/dms-vep/SARS-CoV-2_XBB.1.5_RBD_DMS

Note that most of the analysis in these GitHub repos is performed using dms-vep-pipeline-3 (<https://github.com/dms-vep/dms-vep-pipeline-3>), version 3.5.3.

Raw sequencing data files have been uploaded under BioProjects: PRJNA1034580 for XBB.1.5 full spike library, PRJNA1035795 for XBB.1.5 RBD-only library, PRJNA1035933 for BA.2 full spike library.

Python notebooks and raw event data used for mass photometry analysis are available at <https://github.com/JackTaylorBrown/massphotometry>.

Design of deep mutational scanning libraries

Deep mutational scanning libraries were designed with codon-optimized XBB.1.5 and BA.2 spikes. The sequence of the XBB.1.5 spike is at https://github.com/jbloomlab/SARS-CoV-2-XBB.1.5_Spike_DMS_validations/blob/main/plasmid_maps/3779_pH2rU3_ForInd_XBB15_Sinobiological_CMV_ZsGT2APurR.gb and the BA.2 spike is at https://github.com/jbloomlab/SARS-CoV-2-XBB.1.5_Spike_DMS_validations/blob/main/plasmid_maps/3332_pH2rU3_ForInd_Omicron_sinobiological_BA2_B11529_Spiked21_T7_CMV_ZsGT2APurR.gb. Note that due to an error on our part early in library design, the XBB.1.5 spike used for libraries lacks F490S mutation present in XBB* variants.

The XBB.1.5 full spike libraries were designed to include all accessible and tolerated mutations by including mutations that appeared in more than 50 sequences on GISAID⁷⁴, occurred independently at least 15 times on pre-built SARS-CoV-2 phylogenies from UShER⁷⁵ or occurred independently at least 2 times in any of the following clades: BA.2.75, BQ.1.1, XBB, XBB.1.5. Deletions that met the above criteria were only included if they occurred in the NTD and we specifically added deletions at sites 342-349, 444-449, and 483-486. We also performed saturating mutagenesis on the sites that met the following criteria: occurred independently at least 2500 times on pre-built SARS-CoV-2 phylogenies from UShER or occurred independently at least 100 times in the clades mentioned above. We also saturated mutations at sites that had strong antigenic effects or otherwise were of special interest^{3,9} full list of these sites can be found at https://github.com/dms-vep/SARS-CoV-2_XBB.1.5_spike_DMS/blob/main/library_design/config.yaml under

sites_to_saturate. The full list of mutations included in the XBB.1.5 full spike libraries can be found at https://github.com/dms-vep/SARS-CoV-2_XBB.1.5_spike_DMS/blob/main/library_design/results/mutation_design_classification.csv. As shown in Figure 1b this design strategy biases libraries to contain mostly functional mutations. The reason for choosing such a strategy is: (i) it makes variants with multiple mutations more likely to remain functional and (ii) it limits the number of mutations that need to be included in the final library.

For the XBB.1.5 RBD-only libraries, every position in the RBD (positions 331-531) was mutagenized to all possible amino acids.

For the BA.2 full spike libraries the design of mutations to be included in the library was performed the same way as described previously for BA.1 libraries⁴. The final list of mutations in BA.2 libraries can be found at https://github.com/dms-vep/SARS-CoV-2_Omicron_BA.2_spike_DMS/blob/main/library_design/results/aggregated_mutations.csv. Note that the BA.2 libraries used in this study are the same ones briefly described in Haddox et al. 2023¹¹.

Analysis pipelines for designing mutagenesis primers are provided at https://github.com/dms-vep/SARS-CoV-2_XBB.1.5_spike_DMS/tree/main/library_design for XBB.1.5 full spike libraries, at https://github.com/dms-vep/SARS-CoV-2_XBB.1.5_RBD_DMS/tree/main/library_design for XBB.1.5 RBD-only libraries, and at https://github.com/dms-vep/SARS-CoV-2_Omicron_BA.2_spike_DMS/tree/main/library_design for BA.2 libraries.

Production of plasmid libraries used to generate deep mutational scanning libraries

Libraries of lentivirus backbone plasmids containing mutagenised XBB.1.5 or BA.2 spikes were made as described previously⁴. In brief, primers containing desired mutations described above were ordered from IDT as Oligo Pools. Full list of these primers for XBB.1.5 full spike library can be found at https://github.com/dms-vep/SARS-CoV-2_XBB.1.5_spike_DMS/blob/main/library_design/results/oPools.csv, for for XBB.1.5 RBD only library at https://github.com/dms-vep/SARS-CoV-2_XBB.1.5_RBD_DMS/blob/main/library_design/results/oPools.csv, and for BA.2 library at https://github.com/dms-vep/SARS-CoV-2_Omicron_BA.2_spike_ACE2_binding/tree/main/library_design/results (see csv files ending in *oPool.csv*). These primers were used to mutagenize spike sequences using PCR that involves multiple rounds of PCR mutagenesis reactions⁷⁶. Number of PCR rounds and cycles determines the number of mutations per spike introduced and we targeted ~2-3 mutations per spike, although the precise number of mutations per spike is determined only after lentiviral genomes have been integrated into cells and sequenced with long-read sequencing (see *Long-read PacBio sequencing for variant-barcode linkage* section below). For both XBB.1.5 full spike and RBD-only libraries we pooled spike mutagenesis primers at 2:1 molar ratio between mutations that occur independently multiple times on spike phylogenetic tree and those that occurred multiple times on spike sequences deposited on GISAID database (for RBD only libraries the latter included all possible RBD mutations). For both XBB.1.5 full spike and RBD-only libraries a single round of 10 PCR cycles was used to mutagenize the spike sequence. For BA.2 full spike libraries the same primer pooling strategy and the same number of mutagenesis cycles were used as described for BA.1 libraries⁴. Template spike sequences used for mutagenesis were amplified from https://github.com/jbloomlab/SARS-CoV-2-XBB.1.5_Spike_DMS_validations/blob/main/plasmid_maps/3779_pH2rU3_F_orlnd_XBB15_Sinobiological_CMV_ZsGT2APurR.gb plasmid for XBB.1.5 libraries and from https://github.com/jbloomlab/SARS-CoV-2-XBB.1.5_Spike_DMS_validations/blob/main/plasmid_maps/3332_pH2rU3_F_orlnd_Omicron_sinobiological_BA2_B11529_Spiked21_T7_CMV_ZsGT2APurR.gb plasmid for BA.2 libraries. Spikes for both variants were amplified using *VEP_amp_for* (5'CAGCCGAGCCACATCGCTC) and 3'rev_lib_LinJoin_KHDC (5'CGGAAGAGCGTCGTGTAGGAAAG) primers. After mutagenesis reaction spike sequences were barcoded in a PCR reaction using primers that contained a unique 16 nucleotide barcode that adds barcodes downstream of spike STOP codon. All libraries had two biological replicates (Lib-1 and Lib-2), which represent two independently produced libraries where mutations in spike are associated with unique barcodes in unique combinations with other mutations. Mutagenised and barcoded spike sequence templates were then added into MluI and XbaI digested lentivirus backbone (Addgene #204579) using HiFi reaction (NEB E2621L). Ampure XP bead purified HiFi reactions were then electroporated into 10-beta electrocompetent E. coli cells (NEB, C3020K) and plated overnight. At least 10 electroporation reactions were performed for each plasmid library in order to produce > 2 million CFUs per library. High diversity of barcoded genomes is required in the later steps of library production in order to minimize barcode duplication, which may happen

during lentivirus recombination. For each library bacterial colonies were scraped from overnight plates, pooled and QIAGEN HiSpeed Plasmid Maxi Kit (Cat. No. 12662) was used to prepare plasmid pools used for virus library production.

Production of cell stored deep mutational scanning libraries

Steps for producing cell-stored spike deep mutational scanning libraries have been described in detail previously⁴. In brief, two 6-well plates of 293T cells were transfected with plasmid pools described above, lentivirus helper plasmids (BEI: NR-52517, NR-52519, NR-52518) and VSV-G expression plasmid (Addgene #204156). This produced a VSV-G pseudotyped lentivirus pool carrying mutagenised spike sequences in their genomes. VSV-G pseudotyped viruses were then used to infect 293T-rtTA cells at low multiplicity of infection so that no more than one virus would infect each cell. Reverse tetracycline-controlled transactivator (rtTA) is required to induce expression from inducible TRE3G promoter in the lentivirus backbone in the presence of doxycycline (see Addgene #204579 plasmid structure). Note, that 293T-rtTA cells used here is a specific cell clone we isolated when producing rtTA overexpressing cells, which is especially good at producing high titers virus stocks that are required for successful library production. We described production of these 293T-rtTA cells previously⁴. VSV-G infection step was also used to bottleneck the libraries to the desired number of variants; we aimed for between 50,000 and 100,000 variants per library. Final number of variants in each library is shown in **Extended Data Fig. 1a**. After VSV-G infection, cells with successful lentivirus integration were selected for using puromycin. Puromycin selection was performed until visual inspection showed a pure population of cells express zsGreen (which is part of lentivirus backbone, see plasmid Addgene #204579). At this point all cell stored libraries were frozen until further use.

Long-read PacBio sequencing for variant-barcode linkage

Analysis of linkage between mutations in lentivirus backbone encoded spikes and the barcodes they are associated with was performed using long read PacBio sequencing as described previously⁴. First, we rescued VSV-G pseudotyped viruses from cell-stored libraries by transfecting those cells with lentivirus helper and VSV-G expression plasmids. VSV-G pseudotyped viruses produced from these libraries were then used to infect 293T cells and nonintegrated viral genomes were recovered as described previously⁴. To avoid strand switching and mixing of variant-barcode pairs viral genomes were then minimally PCR amplified using primers with tags that allow to detect strand switching via sequencing. Long read sequencing was performed with PacBio Sequel IIe machine. Consensus variant-barcode sequence was determined requiring at least two CCS sequences per barcode. Variant-barcode lookup tables for each library can be found at:

- For XBB.1.5 full spike library
https://github.com/dms-vep/SARS-CoV-2_XBB.1.5_spike_DMS/blob/main/results/variants/codon_variants.csv
- For XBB.1.5 RBD only
https://github.com/dms-vep/SARS-CoV-2_XBB.1.5_RBD_DMS/blob/main/results/variants/codon_variants.csv
- For BA.2 library
https://github.com/dms-vep/SARS-CoV-2_Omicron_BA.2_spike_ACE2_binding/blob/main/results/variants/codon_variants.csv

Long read sequencing data was also used to determine the average spike mutation frequency in each library. For XBB.1.5 full spike library Lib-1 and Lib-2 mutation frequency 1.91 mutations per spike. For XBB.1.5 RBD only library Lib-1 had an average of 1.82 mutations per spike and Lib-2 had an average of 1.9 mutations per spike. BA.2 libraries had an average of 2.32 and 2.33 mutations per spike for Lib-1 and Lib-2 libraries, respectively.

Cell entry effect measurement using deep mutational scanning libraries

Cell entry effects for each variant were measured as described previously⁴. In brief, ~1.5 million transcription units of spike pseudotyped library viruses and ~5 million of VSV-G pseudotyped transcription units made from the same cell-stored libraries were used to infect target cells. For spike pseudotyped libraries 293T-cells either overexpressing high amounts of ACE2 (described in Crawford et al 2020⁵) or cell expressing medium amount of ACE2 (described in Farrell et al. 2021³⁸) were used. Whenever cells were plated for infection with spike-pseudotyped viruses (including for ACE2 and sera selections described below) cells were additionally supplemented with 2.5 µg/ml of amphotericin B (Sigma, A2942) at the time of plating, which we have previously shown⁴ to increase virus titers. For VSV-G pseudotyped libraries 293T

cells were used (we used cells not expressing any ACE2 in order to avoid any selection of spike, which can still be present on the surface of these VSV-G pseudotyped viruses). 12-15 hours post infection unintegrated viral genomes were recovered using QIAprep Spin Miniprep kit and prepared for Illumina sequencing as described previously⁴.

For each variant functional score was calculated by getting a log enrichment ratio:

$\log_2 \left(\left[\frac{n_{post}^v}{n_{post}^{wt}} \right] / \left[\frac{n_{pre}^v}{n_{pre}^{wt}} \right] \right)$, where n_{post}^v is the count of variant v in the post-selection (spike-pseudotyped) infection, n_{pre}^v is the count of variant v in the pre-selection (VSV-G-pseudotyped) infection, and n_{post}^{wt} and n_{pre}^{wt} are the counts of variants without mutations, i.e. wildtype spike, in each condition. Positive functional scores indicate variant is able to enter cells better than wildtype and negative functional scores indicate variant is worse at entering the cells than wildtype.

The multi-dms software package¹¹ was used to fit a global epistasis model⁷⁷ with a sigmoid global epistasis function to the variant functional scores and to calculate mutation-level effects on cell entry. See https://dms-vep.github.io/SARS-CoV-2_XBB.1.5_spike_DMS/notebooks/func_effects_global_epistasis_Lib1-230614_high_ACE2.html for an example of this fitting for one library; the HTML documentation of the pipeline linked in the *Data availability* section has links to comparable fitting notebooks for each library.

The cell entry effects we describe in the paper are based on the cell entry experiments done on 293T-cells overexpressing high amounts of ACE2 as opposed to medium ACE2 expressing cells. Expression of more ACE2 in 293T-cells leads to higher virus titers on these cells and therefore the fitting of global epistasis model on data from these cells is slightly better.

Use of non-neutralizable standard for ACE2 binding and serum selection experiments

For both ACE2 binding and serum selection experiments a non-neutralizable standard was used in order to enable conversion of sequence counts to absolute neutralization⁴. We have previously described the use of VSV-G pseudotyped virus as the non-neutralizable standard in antibody selection experiments⁴, and that VSV-G standard was also used for selections with soluble ACE2 protein to measure receptor binding since VSV-G is not neutralized by ACE2. For serum selections, we found that high concentrations of serum appreciably neutralize VSV-G itself making it not suitable as a non-neutralizable standard. We screened multiple alternative viral entry proteins and found that the RDPro glycoprotein, a modified version of an endogenous feline virus RD114 containing HIV R-peptide⁷⁸ that we further modified to contain MLV-A cytoplasmic tail to improve pseudovirus titers⁷⁹, was not neutralized even at high serum concentrations (data not shown). The full sequence of RDPro viral entry protein used in this study can be found at https://github.com/jbloomlab/SARS-CoV-2-XBB.1.5_Spike_DMS_validations/blob/main/plasmid_maps/3737_HDM_RDPro_Twist_MLV-A_HIV-pep_correction.gb. RDPro envelope pseudotyped viruses were produced from cells with integrated barcoded lentivirus genomes as described previously for VSV-G pseudotyped standard⁴. Because RDPro-pseudotyped lentivirus titers were $\sim 10^4$ TU/ml, we further concentrated virus stocks using Lenti-XTM Concentrator (Takara, 631232) to between 3.5×10^5 - 1.5×10^6 TU/ml. Given that producing high titer RDPro stocks is more time consuming than making VSV-G stocks we chose to not switch to using RDPro for ACE2 binding experiments, since VSV-G is not neutralized by soluble ACE2. Note that both RDPro and VSV-G non-neutralizable standards contain the same barcodes and therefore as long as the non-neutralizable standard is not neutralized, the results should remain the same regardless of the standard used.

Recombinant Protein Production

SARS-CoV-2 spike ectodomain and human ACE2 ectodomains were expressed and purified as described previously^{19,21}. Mutant Spike ectodomain constructs were designed in the BA.2 and XBB.1.5 backgrounds with HexaPro²⁸ mutations, N terminal "MFVFLVLLPLVSS" signal peptide, C terminal GSSG linker, foldon, linker, avi-8x polyhistidine tag, and were cloned into a pCDNA3.1(+) vector. A222M, N405A, A570F, A570D, A701M, D950N, R493Q, R498V mutations were evaluated in the BA.2 background; Q115K, T167I, N234T, N405A, Q762L, F1121L, R498V, Q804L, Q493L, G614D, F371N, A222M mutations were evaluated in XBB.1.5 background. Expi293F cells were diluted to a density of 3 million cells per mL and transfected using ExpiFectamine 293 Transfection Kit (Thermo Fisher Scientific). Cells were incubated shaking at 130 rpm at 37°C and 8% CO₂. Three to four days post transfection proteins were purified from clarified

supernatants. Human ACE2 ectodomains were purified using 1mL HisTrap Fast Flow nickel affinity columns (Cytiva), and washed with 20 mM imidazole, 25 mM sodium phosphate pH 8.0, and 300 mM NaCl prior to elution with an imidazole gradient using a buffer containing 500 mM imidazole pH 8.0, 25 mM sodium phosphate, 300 mM NaCl pH 8.0. SARS-CoV-2 spike ectodomains were purified using 1mL of Ni Excel resin (Cytiva) and washed with 40 mM imidazole pH 8.0, 25 mM sodium phosphate pH 8.0, and 300 mM NaCl prior to elution with 300 mM imidazole pH 8.0, 25 mM sodium phosphate pH 8.0, and 300 mM NaCl. SARS-CoV-2 spike ectodomains were buffer exchanged into 20 mM sodium phosphate pH 8.0, and 100 mM NaCl (PBS) using centrifugal filters (corning) with a MWCO of 100 kDa. Purified BA.2 and XBB.1.5 S variants were analyzed by negative stain electron microscopy to confirm retention of proper folding and monodispersity (**Extended Data Fig. 5**). Human ACE2 ectodomain were concentrated using centrifugal filters (Corning) with a MWCO of 30kDa and were further purified by size exclusion chromatography and run through a Superdex 200 Increase 10/300 GL column (Cytiva) pre-equilibrated in PBS. All proteins were analyzed by SDS-PAGE for purity, then flash frozen and stored at -80°C. For deep mutational scanning ACE2 binding experiments biotinylated dimeric ACE2 was purchased from ACROBiosystems (AC2-H82E7-1mg).

ACE2 binding measurement using deep mutational scanning libraries

Previous research has shown that soluble ACE2 can neutralize SARS-CoV-2 variants with potency proportional to virus binding to the receptor^{3,17}. We used this observation to measure the effects of mutations in our deep mutational scanning libraries on ACE2 binding.

As described previously⁴, before starting ACE2 binding experiments we spiked-in a VSV-G non-neutralizable standard at 1-2% of the total virus titers used. ~1 million virus transcription units per sample were incubated with soluble monomeric or dimeric ACE2 at 37°C for 1 h before being added to 293T-ACE2 cells. 293T-ACE2 cells expressing a medium amount of ACE2 ('medium-ACE2' cells described in Farrell et al. 2021³⁸) were used for all ACE2 binding experiments. For these experiments we targeted a range of ACE2 concentrations to use that would span from less than IC50 to full virus neutralization in order to capture both mutations that increase ACE2 binding (those that are neutralized by soluble ACE2 very potently) and those that decrease it (which would be more difficult to neutralize with soluble ACE2 neutralized). For monomeric ACE2 the starting concentration was 2.88 µg/ml and it was increased 3-fold for the other dilutions. For dimeric ACE2 starting concentration was 0.21 µg/ml and it was similarly increased 3-fold for the other dilutions. 12-15 hours post infection non-integrated lentiviral genomes were extracted from cells and barcode sequencing libraries were prepared as described previously⁴. ACE2 binding experiments were performed with two biological replicates for each library.

Analysis of mutation-level effects and fitting of neutralization curves to the data was performed using *polyclonal* software⁸⁰ version 6.9. Examples of polyclonal model fitting for monomeric ACE2 data can be found at:

- For XBB.1.5 full spike libraries
https://dms-vep.github.io/SARS-CoV-2_XBB.1.5_spike_DMS/notebooks/fit_escape_ACE2_binding_Lib1-230614-monomeric_ACE2.html
- For XBB.1.5 RBD only libraries
https://dms-vep.github.io/SARS-CoV-2_XBB.1.5_RBD_DMS/notebooks/fit_escape_ACE2_binding_Lib1-230615-monomeric-ACE2.html
- For BA.2 libraries
https://dms-vep.github.io/SARS-CoV-2_Omicron_BA.2_spike_ACE2_binding/notebooks/fit_escape_ACE2_binding_Lib1-230114-monomeric_ACE2.html

The HTML documentation of the pipeline linked in the *Data availability* section has links to comparable fitting notebooks for each replicate library, as well as dimeric ACE2 selection data available for XBB.1.5 RBD only libraries.

Mass photometry

Mass photometry was performed on a Refeyn TwoMP system (Refeyn Ltd). Microscope cover slides were rinsed with isopropanol and Milli-Q water then dried under nitrogen flow. Sample chambers were assembled using silicon gaskets, and the instrument lens coated with immersol before placing slides on the MP sample stage. Samples were added to the sample chamber and the instrument was focused immediately prior to each data acquisition. Spike ectodomain samples

were diluted to 25 nM for all data acquisitions. Spike was mixed with 0, 25, 50, 75 or 100 nM of dimeric human ACE2 ectodomain and incubated at room temperature for 5 min prior to data acquisition. MP image data was analyzed in Refeyn DiscoverMP, using in-house protein standards for mass calibration, and processed single-particle mass detection events were exported for determination of RBD-ACE2 occupancy. The mass events were truncated to a range of 0-1600 kDa for apo spike runs and to 350-1300 kDa for spike-ACE2 runs, thereby excluding multiple spikes cross-linked by one or multiple ACE2 dimers, as well as free ACE2 (which is not used to calculate RBD occupancy). Retained mass events for each run were used to estimate two-, three-, or four-component Gaussian mixture models (GMMs) with Scikit-Learn⁸¹, and each component was assigned as representing unbound spike, or 1 ACE2-, 2 ACE2-, or 3 ACE2-bound spike trimers if its molecular mass (Gaussian mean) fell between 400-600 kDa, 600-800 kDa, 800-1000 kDa, or 1000-1200 kDa, respectively. The relative abundance of each of the four species, and thus RBD-ACE2 occupancy, were determined from the respective weights (proportions of overall probability mass) of the Gaussian components, as follows:

$$RBD\ occupancy = \frac{S^{1xACE2} + 2S^{2xACE2} + 3S^{3xACE2}}{S^{total} \times 3}, S^{1xRBD}\ occupancy = \frac{S^{1xACE2} + S^{3xACE2}}{S^{total}}, S^{2xRBD}\ occupancy = \frac{S^{2xACE2}}{S^{total}},$$

$$S^{3xRBD}\ occupancy = \frac{S^{3xACE2}}{S^{total}}$$

where S^{1xACE2} is the weight of the respective Gaussian components for single ACE2 bound spike, S^{2xACE2} is Gaussian component spike bound by 2 ACE2 molecules, S^{3xACE2} is Gaussian component spike bound by 3 ACE2 molecules and S^{total} is the sum of Gaussian components for spike bound by no, one, two or three ACE2 molecules (**Extended Data Fig. 6a**). For visualization of the modeling results (**Extended Data Figs. 3-4**) and for selection of the number of Gaussian components appropriate for each sample, each GMM was used to predict a (continuous) mass frequency distribution, which was area-scaled and overlaid on the corresponding full-range mass event histogram.

Serum escape mapping using deep mutational scanning libraries

Before use, sera were heat inactivated at 56°C for 1 hour to eliminate complement activity. XBB* infection sera neutralization was determined using standard pseudovirus neutralization assay described in Crawford et al. (2020)⁵. The sequence of the spike expression plasmid used for these experiments is provided at https://github.com/ibloomlab/SARS-CoV-2-XBB.1.5_Spike_DMS_validations/blob/main/plasmid_maps/HDM_XBB15.gb. Using these measurements we determined the amount of serum needed to neutralize the virus at IC98-IC99. As described previously⁴, before starting selections we spiked-in a non-neutralizable standard at 1-2% of the total virus titers used. RDPro pseudotyped non-neutralizable standard was used for all serum selections to avoid non-specific standard neutralization (see section *Use of non-neutralizable for ACE2 and serum selection experiments* above). For sera selection experiments ~1 million transcription units for each library sample were used. Libraries were incubated at three increasing serum concentrations starting with IC98-IC99 (depending on serum volume available) and increasing serum concentration 4-fold at each dilution. These serum concentrations were selected so that only a small percentage of variants would be able to pass sera selection, therefore selecting for strongest escape variants. Additional sera concentrations are used to cover a greater dynamic range as sometimes neutralization values determined against wild-type spike using luciferase-based system do not quite match neutralization values for library virus pool. Virus-serum mixtures were incubated for 1 h at 37°C and subsequently 293T-ACE2 cells were infected with them. We used medium ACE2 expressing cells for all serum selection experiments ('medium-ACE2' cells in Farrell et al. 2021³⁸) although as shown in **Extended Data Fig. 7** we did not detect major differences in serum escape compared to cells with high ACE2 expression. 12-15 hours post infections non-integrated lentiviral genomes were extracted from cells and barcode sequencing libraries were prepared as described previously⁴.

Polyclonal software⁸⁰ (version 6.9) was used to analyze mutation-level escape and fit neutralization curves to the data. An example for data fitting for one sera sample can be found at https://dms-vep.github.io/SARS-CoV-2_XBB.1.5_spike_DMS/notebooks/fit_escape_antibody_escape_Lib1-230815-sera-343C_mediumACE2.html for XBB.1.5 full spike library and at https://dms-vep.github.io/SARS-CoV-2_XBB.1.5_RBD_DMS/notebooks/fit_escape_antibody_escape_Lib1-230815-sera-343C_mediumACE2.html for XBB.1.5 RBD library. The HTML documentation of the pipeline linked in the *Data availability* section has links to comparable fitting notebooks for each biological replicate, as well as all other sera fits.

Validation of escape using standard pseudovirus neutralization assay

To validate serum escape mutations, we cloned desired point mutants into an expression plasmid coding for XBB.1.5 spike. The sequence of this XBB.1.5 expression plasmid is at https://github.com/jbloomlab/SARS-CoV-2-XBB.1.5_Spike_DMS_validations/blob/main/plasmid_maps/3813_HDM_XBB15_with_F490S.gb (note this spike sequence contains F490S mutation). Pseudoviruses were generated and titrated as described in Crawford et al. (2020)⁵ except that pHAGE6_Luciferase_IRES_ZsGreen backbone was used which requires only Gag/Pol (BEI: NR-52517) helper plasmid to produce pseudoviruses. Pseudovirus stocks were diluted to stock concentration of >200,000 relative light units per ul and neutralization assays were performed on medium-ACE2 cells. Starting serum dilution for neutralization assays was 1:30 and it was serially diluted 1:3 to generate neutralization curves. Neutralization curves were plotted by fitting a Hill curve to fraction infectivity data for each variant. This was done using *neutcurve* package (<https://jbloomlab.github.io/neutcurve/>, version 0.5.7). Analysis notebook for neutralization curves is at https://github.com/jbloomlab/SARS-CoV-2-XBB.1.5_Spike_DMS_validations/tree/main.

Cells

All cells in this study were maintained in D10 media (Dulbecco's Modified Eagle Medium with 10% heat-inactivated fetal bovine serum, 2 mM l-glutamine, 100 U/mL penicillin, and 100 µg/mL streptomycin). 293T-ACE2 cells expressing medium amount of ACE2 ('medium-ACE2' cells described in Farrell et al. 2021³⁸) were additionally supplemented to 2 µg/ml of doxycycline. Cells used to store spike libraries were maintained in media supplemented with doxycycline-free FBS as described previously⁴.

Ethics statement

XBB* infection sera were collected after informed consent from participants in the prospective longitudinal Hospitalized or Ambulatory Adults with Respiratory Viral Infections (HAARVI) study from Washington State, USA, which was approved by University of Washington Institutional Review Board (protocol #STUDY00000959).

Comparison of deep mutational scanning phenotypes to changes in clade growth

To estimate clade growth rates, we used a multinomial logistic regression model of global lineage frequency data. GISAID sequences were obtained from the bulk .fasta download (dated 2023-10-02) and processed with Nextclade (v2.14.0) using the BA.2 reference (sars-cov-2-21L). Using Nextclade quality metrics, only sequences with >90% coverage and an overall QC status of "good" were retained. Since outlier dates could distort model estimates, we required sequences to have a fully specified deposition and collection date (ie. year, month, and day), and to have a collection date within 150 days of deposition (primarily to avoid collection dates where the year was incorrectly annotated). Finally, for each lineage, we excluded sequences with dates that were extreme outliers for that specific lineage, falling outside of 3.5 times the interquartile range of the median. For the model fit, we retained countries with >500 sequences, and lineages with >200 sequences. Counts for each lineage were aggregated per country, per day.

We model lineage counts using multinomial logistic regression. For the i^{th} lineage, g_i denotes the global (ie. shared across all regions) per-lineage relative growth rate parameter (these are the parameters we wish to estimate), and c_j is a "nuisance" intercept parameter for the i^{th} lineage in the j^{th} region. We assume that the ratio of the frequency of any two lineages varies approximately exponentially over time t (here in years), and model the probability of the n^{th} sample $S_j^t[n]$ being from lineage i as:

$$P(S_j^t[n] = i) = \frac{e^{t g_i + c_{ij}}}{\sum_i e^{t g_i + c_{ij}}}$$

We use gradient descent to minimize the negative log probability $-\sum_n \log[P(S_j^t[n] = i)]$, with the relative growth rate parameters "centered" to have their mean equal to zero, removing a superfluous degree of freedom. Growth rates are interpreted such that the ratio of the frequencies of two lineages changes with time as

$$\frac{P(S_j^t[n]=i)}{P(S_j^t[n]=q)} = K \cdot e^{t(g_i - g_q)}$$

where $g_i - g_q$ is the difference in the growth rates between the two lineages and, for a given region, $K = \exp(c_{ij} - c_{iq})$ is a constant determined by their intercepts.

The model was implemented in the Julia language, using Flux.jl and CUDA.jl to allow for GPU computation, and optimized using Flux's "AdamW" optimizer with very weak (10^{-10}) "weight decay" numerical regularization of the parameters. The model implementation can be viewed at: <https://github.com/MurrellGroup/MultinomialLogisticGrowth>.

Visual inspection of the model fits to count data, aggregated daily for each lineage and region, showed an acceptable fit at the region level, despite using globally shared growth rate parameters, indicating relatively consistent growth rates across regions. A representative example (from Switzerland) is shown at **Extended Data 16**.

See <https://github.com/MurrellGroup/MultinomialLogisticGrowth/tree/main/plots> for similar visualizations across all regions used in the global model fit. Note that when a region has no samples for a given period, this will not inform the growth rate estimates, but we still plot the model's expected lineage frequencies.

The growth rate estimates themselves are at https://github.com/MurrellGroup/MultinomialLogisticGrowth/blob/main/model_fits/rates.csv. For the analyses in this paper, we considered only growth estimates from XBB descended clades with at least 400 sequences, since clades with more sequences have more accurate growth estimates. The definitions of the clades (e.g., which mutations they contain) as well as their phylogeny (parents and descendants for each clade) were taken from https://github.com/corneliusroemer/pango-sequences/blob/main/data/pango-consensus-sequences_summary.json.

As described in the results and **Fig. 6** and **Extended Data Figs. 12** and **13**, directly predicting growth rates of clades from the deep mutational scanning is a confounded approach due to both phylogeny and the simple fact that newer clades tend to have both more spike mutations and higher growth rates, leading to a trivial correlation of clade growth rate with number of spike mutations. The real question of interest is not whether more fit clades with additional mutations will be selected over time (we know they will), but rather which of the mutant clades present at any given time will be more successful. Therefore, as indicated in **Fig. 6b**, we computed the *change* in growth rate for each parent-descendant clade pair with estimates for both clade members and at least one spike mutation. We then also computed the change in each spike phenotype as measured by deep mutational scanning for the clade pairs based on the mutations separating the pair members, simply adding together the mutation effects for pairs separated by multiple spike mutations. Non-spike mutations were ignored. The Pearson correlations with each phenotype are shown in **Fig. 6c**, and the statistical significance of the correlations were assessed by randomizing the deep mutational scanning measurements among mutations 100 times and assessing how many randomizations had a correlation greater than or equal to the observed value. To test the predictive value of combining all spike phenotypes, we performed a similar analysis but using ordinary least squares multiple linear regression on all three phenotypes. Those results are shown in **Fig. 6d**, with the significance again assessed by comparing the actual Pearson correlation to that generated by fitting the model to data randomized among sites. To compute the unique variance explained by each phenotype, we removed the phenotypes one-by-one and computed the unique variance explained as the squared Pearson correlation for the full model minus the squared Pearson correlation for the model with that phenotype removed. See https://dms-vep.github.io/SARS-CoV-2_XBB.1.5_spike_DMS/notebooks/current_dms_compare_natural.html for the notebook performing this analysis, and https://github.com/dms-vep/SARS-CoV-2_XBB.1.5_spike_DMS/blob/main/results/compare_natural/current_dms_clade_pair_growth.csv for the numerical data on the clade pairs and their changes in spike phenotypes.

We compared the predictive value of the full-spike deep mutational scanning to the predictive value of models based on several other values. The first such comparator model simply involves counting the change in number of spike mutations relative to Wuhan-Hu-1 in each clade pair; as shown in **Extended Data Fig. 12d**, this model has no predictive value.

The second comparator model uses the effect of RBD mutations on ACE2 affinity and RBD expression as measured in yeast-display deep mutational scanning of the XBB.1.5 RBD²² as well as per-site escape values (same value assigned to each mutation at each site) as computed using the default settings of the antibody escape calculator⁹ at <https://github.com/jbloomlab/SARS2-RBD-escape-calc/tree/5ebb88e5b8c9adc1b601b3cb1cc5308532d97a38> which is based on monoclonal antibody deep mutational scanning data⁵⁴. For this model, all non-RBD mutations were assigned a value of zero for all phenotypes. As shown in **Extended Data Fig. 13b,d**, the predictions of this model are not significant at a level of $P = 0.05$ compared to models with the measurements randomized among sites.

The third comparator model uses the effects of mutations to XBB.1.5 as estimated using the EVEscape method (<https://evescape.org/data>)⁵⁵. As shown in **Extended Data Fig. 13c,f**, the predictions of this model are not significant at a level of $P = 0.05$.

The numerical data used for all the comparator models is at https://github.com/dms-vep/SARS-CoV-2_XBB.1.5_spike_DMS/tree/main/data/compare_natural_datasets.

Acknowledgments

We thank Ryan Hisner for helpful comments during library design. We thank Cornelius Roemer and Richard Neher for helpful comments on interpreting the effects of mutations in the context of SARS-CoV-2 evolution. We thank Thomas Peacock for useful discussions on ACE2 binding. We gratefully acknowledge all data contributors, including the authors and their originating laboratories responsible for obtaining the specimens, and their submitting laboratories that generated the genetic sequence and metadata and shared via the GISAID Initiative the data on which part of this research is based. JDB is an investigator at the Howard Hughes Medical Institute. D.V. is an Investigator of the Howard Hughes Medical Institute and the Hans Neurath Endowed Chair in Biochemistry at the University of Washington. This work was supported by the NIH/NIAID R01AI141707 grant and contract 75N93021C00015 to J.D.B as well as NIH/NIAID P01AI167966 and DP1AI158186 grants and 75N93022C00036 contract to D.V.), a Pew Biomedical Scholars Award (D.V.), an Investigators in the Pathogenesis of Infectious Disease Awards from the Burroughs Wellcome Fund (D.V.), the University of Washington Arnold and Mabel Beckman cryoEM center and the National Institute of Health grant S10OD032290 (to D.V.). This research was also supported by the Genomics & Bioinformatics Shared Resource, RRID:SCR_022606, of the Fred Hutch/University of Washington Cancer Consortium (P30 CA015704), by the Flow Cytometry Shared Resource, RRID:SCR_022613, of the Fred Hutch/University of Washington/Seattle Children's Cancer Consortium (P30 CA015704), and by Fred Hutch Scientific Computing, NIH grants S10-OD-020069 and S10-OD-028685. B.M. was funded by SciLifeLab's Pandemic Laboratory Preparedness programme (VC-2022-0028) and the Erling Persson Foundation (2021 0125). TB is an investigator at the Howard Hughes Medical Institute.

Author contributions

Conceptualization, B.D., J.D.B.; methodology, B.D., J.B., B.M., D.V., J.D.B.; experiments, B.D., J.B., T.E.M, J.L., A.G.F, D.S, C.S.; computational analysis, B.D., J.B., B.M., D.A. D.V., T.B., M.D.F., J.B.D; writing – original draft, B.D. and J.D.B.; writing – review & editing; B.D., D.V., J.D.B.;resources, J.L., H.Y.C.; supervision J.D.B.; funding acquisition, J.D.B.

Competing interests

J.D.B., and B.D. are inventors on Fred Hutch licensed patents related to the pseudovirus deep mutational scanning system used in this paper. J.D.B. consults for Apriori Bio, Invivyd, Aerium Therapeutics, and the Vaccine Company on topics related to viral evolution. HYC reports consulting with Ellume, Pfizer, and the Bill and Melinda Gates Foundation. She has served on advisory boards for Vir, Merck and Abbvie. She has conducted CME teaching with Medscape, Vindico, and Clinical Care Options. She has received research funding from Gates Ventures, and support and reagents from Ellume and Cepheid, all outside of the submitted work. D.V. is named as inventor on patents for coronavirus vaccines filed by the University of Washington.

Supplementary Information is available for this paper.

Correspondence and requests for materials should be addressed to Jesse D Bloom
jbloom@fredhutch.org

Reprints and permissions information is available at www.nature.com/reprints.

Figures

Fig. 1: Deep mutational scanning to measure phenotypes of the XBB.1.5 and BA.2 spikes

a, We measure the effects of mutations in spike on cell entry, receptor binding and serum escape. We then use these measurements to predict the evolutionary success of human SARS-CoV-2 clades. **b**, Distribution of effects of mutations in XBB.1.5 and BA.2 spikes on entry into 293T-ACE2 cells for all mutations in the deep mutational scanning libraries, stratified by the type of mutation and the domain in spike. Negative values indicate worse cell entry than the unmutated parental spike. Note that the library design favored introduction of substitutions and deletions that are well tolerated by spike, explaining why many mutations of these types have neutral to only modestly deleterious impacts on cell entry. **c**, Cell entry effects of mutations F456L, P1143L and deletion of V483 relative to the distribution of effects of all substitution and deletion mutations in the libraries. Interactive heatmaps with effects of individual mutations across the whole spike on cell entry are at https://dms-vep.github.io/SARS-CoV-2_XBB.1.5_spike_DMS/htmls/293T_high_ACE2_entry_func_effects.html and https://dms-vep.org/SARS-CoV-2_Omicron_BA.2_spike_ACE2_binding/htmls/

293T_high_ACE2_entry_func_effects.html . The boxes in panels b and c span the interquartile range, with the horizontal white line indicating the median. For panel c, the effect of deleting V483 was not measured in the BA.2 spike.

Fig. 2: Effects of mutations on full-spike ACE2 binding measured using pseudovirus deep mutational scanning

a, Neutralization of pseudoviruses with the indicated spikes by soluble monomeric ACE2. Viruses with spikes that have stronger binding to ACE2 are neutralized more efficiently by soluble ACE2 (lower NT50), whereas viruses with spikes with worse binding are neutralized more weakly. ACE2 affinity values measured by surface plasmon resonance for BA.2 and Wu-1+D614G are shown in brackets¹⁹. **b,** Correlation between neutralization NT50 by soluble ACE2 versus the RBD affinity for ACE2 as measured by titrations using yeast-displayed RBD²⁰. **c,** Effects of NTD and RBD mutations on full-spike ACE2 binding as measured using pseudovirus deep mutational scanning. Mutations that enhance ACE2 binding are shaded blue, mutations that decrease affinity are shaded orange, mutations that are too deleterious for cell entry to be measured in the binding assay are dark gray, and light gray indicates mutations not present in our libraries. Interactive heatmaps showing mutational effects on ACE2 binding for the full XBB.1.5 and BA.2 spikes are at https://dms-vep.github.io/SARS-CoV-2_XBB.1.5_spike_DMS/htmls/monomeric_ACE2_mut_effect.html and https://dms-vep.org/SARS-CoV-2_Omicron_BA.2_spike_ACE2_binding/htmls/monomeric_ACE2_mut_effect.html. Note that a few sites are missing in the static heatmap in this figure due to lack of coverage or deletions in the XBB.1.5 spike; see the interactive heatmaps for per-site numbering. **d,** Correlations between the effects of RBD mutations on ACE2 binding measured using the pseudovirus-based approach (this study) and yeast-based RBD display^{20,22}. **e,** Distribution of effects of individual mutations on full-spike ACE2 binding for all functionally tolerated mutations in our libraries, stratified by RBD versus non-RBD mutations. Note that effects of magnitude greater than two are clamped to the limits of the plots' x-axes.

c

mutation	# of clades	clades	XBB.1.5 ACE2 binding
T572I	9	XBB.1.5.44; GK.3.2; FL.36; GA.4.1.2; FY.1.2; FY.5.1; FY.5.4; FY.5.5.1; FY.8	0.43
T732I	6	HK.3.9; JG.1; XBB.1.16.7; XBB.1.16.8; XBB.1.38; GE.1.3	0.37
F157L	5	EU.1.1.3; HC.2; HK.3.1; HK.19; EG.5.1.6	0.29
S704L	9	XBB.1.5.58; GK.1; JG.3; HV.1.6.1; EG.5.2.1; EG.5.2.3; GA.4.1; KE.2; FY.4.1.1	0.27
E583D	4	FZ.1; JD.1.1.2; FY.1.3; HL.1	0.17
Q52H	6	FT.3; FL.20; EG.5.1; XBB.1.16.12; JM.1; XBB.2.10	0.17

Fig. 3: Non-RBD mutations impact ACE2 binding

a, ACE2 binding measurements using mass photometry. Histogram on the left shows distribution of spike molecular mass when no ($S^{0 \times ACE2}$) one ($S^{1 \times ACE2}$), two ($S^{2 \times ACE2}$) or three ($S^{3 \times ACE2}$) ACE2 molecules are bound. We measure how this mass distribution changes as spike is incubated with increasing concentrations of soluble dimeric ACE2. RBD occupancy is the fraction of RBDs bound

to ACE2, calculated using Gaussian components for $S^{0 \times ACE}$, $S^{1 \times ACE}$, $S^{2 \times ACE}$ and $S^{3 \times ACE}$ at each ACE2 concentration. **b**, RBD occupancy measured using mass photometry for different BA.2 and XBB.1.5 spike variants. Top left panel shows that a BA.2 spike mutation known to increase ACE2 binding (R493Q/blue) has greater RBD occupancy relative to unmutated BA.2 (black) spike, by contrast a mutation known to decrease ACE2 binding (R498V/green) has lower RBD occupancy in both BA.2 (top left panel) and XBB.1.5 (bottom left panel) backgrounds. Panels on the right show RBD occupancy for BA.2 (top right) and XBB.1.5 (bottom right) spike variants with mutations in S_1 or S_2 subunits measured to increase ACE2 binding in the deep mutational scanning. Values shown in parentheses after the mutation in the legend are the effect on ACE2 binding measured by deep mutational scanning. **c**, Non-RBD mutations measured to increase ACE2 binding in deep mutational scanning experiments that have arisen independently as defining mutations in at least four XBB-descended clades.

Fig. 4: Serum antibody escape mutations for individuals with prior XBB* infections

a, Escape at each site in the XBB.1.5 spike averaged across 10 sera collected from individuals with prior XBB* infections. The points indicate the total positive escape caused by all mutations at each site. See https://dms-vep.github.io/SARS-CoV-2_XBB.1.5_spike_DMS/htmls/summary_overlaid.html for an interactive version of this plot with additional mutation-level data. **b**,

Zoomed view of the escape at each site in RBD with each line representing one of the 10 sera. Key sites are labeled with red circles indicating escape for each of the 10 sera. Red data points indicate escape for each individual at select RBD positions. **c**, Logo plots showing the 16 sites of greatest total escape after averaging across the sera. Letter heights indicate escape caused by mutation to that

amino acid, and letters are colored light yellow to dark brown depending on the impact of that mutation on ACE2 binding (cf. color key). The top plot shows all amino-acid mutations measured, and the bottom plot shows only amino acids accessible by a single nucleotide mutation to the XBB.1.5 spike. **d**, Left: correlation between DMS escape scores and pseudovirus neutralization assay IC90 values for three sera. Right: logo plot showing escape for all sites with mutations validated in the neutralization assays, with the specific validated mutations in red.

Fig. 5: Sera escape and ACE2 binding are inversely correlated for non-RBD and ACE2-distal RBD sites

a, Left: correlation between ACE2 binding and sera escape for amino-acid mutations at non-RBD sites with the highest mutation-level sera escape (each point is a distinct amino acid mutation). Average escape for each mutation across all sera is shown. Right: logo plot for the same sites, with letter heights proportional to escape caused by that mutation (negative heights mean more neutralization), and letter colors indicating effect on ACE2 binding (green means better binding). **b**, A similar plot for RBD sites that are distal (at least 15 Å) from ACE2. **c**, A similar plot for RBD sites proximal to ACE2. Only sites with at least seven different mutations measured are included in the logo plots. **d**, Top-down view of XBB spike (PDB ID: 8IOT) with the non-RBD and ACE2-distal sites

shown in panels a and b highlighted as spheres. The RBD is pink, the NTD is blue, and sites in SD1 are green.

Fig. 6: Spike phenotypes measured by deep mutational scanning partially predict the evolutionary success of SARS-CoV-2 clades

a, Phylogenetic tree of XBB-descended Pango clades, colored by their relative growth rates. The tree shows only clades with at least 400 sequences and at least one new spike mutation, and their ancestors. Ancestor clades with insufficient sequences for growth rate estimates are in white. **b**, The same phylogeny but with branches colored by the change in growth rate between parent-descendant clade pairs. **c**, Correlation between the changes in growth rate for parent-descendant clade pairs versus the change in each spike phenotype measured in the XBB.1.5 full-spike deep mutational scanning (multiple mutations are assumed to have additive

effects). The text above each plot shows the Pearson correlation (r) and a P-value computed by comparing the actual correlation to that for 100 randomizations of the experimental data among mutations. **d**, Ordinary least squares multiple linear regression of changes in growth rate versus all three measured spike phenotypes. The small text indicates the unique variance explained by each variable as well as the coefficients in the regression. See https://dms-vep.github.io/SARS-CoV-2_XBB.1.5_spike_DMS/htmls/current_dms_clade_pair_growth.html and https://dms-vep.github.io/SARS-CoV-2_XBB.1.5_spike_DMS/htmls/current_dms_ols_clade_pair_growth.html for interactive versions of panels **c** and **d** where points can be moused over for details on clades and their mutations. See **Extended Data Fig. 14** for a similar analysis that also includes BA.2 and BA.5 descended clades.

Supplementary figures

Extended Data Fig. 1: XBB.1.5 and BA.2 spike deep mutational scanning libraries

a, Number of targeted and final number of mutations and barcoded variants in the XBB.1.5 and BA.2 full spike and XBB.1.5 RBD pseudovirus-based deep mutational scanning libraries. **b**, Total number of unique spike amino-acid mutations present in BA.2, BA.5, BA.2.86, and XBB descended Pango clades and the number of those mutations that are present in at least three

barcoded variants in each replicate of the XBB.1.5 full spike libraries, which was the minimum number of occurrences we needed to make high-confidence estimates of the mutational effects on cell entry. The first number is the total number of mutations meeting the criteria and the number in parentheses is the number of these mutations covered in the libraries: for example, there are 108 spike amino-acid mutations that occur in more than one XBB-descended clade, and 107 of those mutations are well covered in our XBB.1.5 full spike libraries. **c**, Method for creating genotype-phenotype linked spike deep mutational scanning libraries, as previously described in Dadonaite et al. (2023)⁴. Lentivirus backbone plasmids encoding barcoded mutagenised spike genes together with helper and VSV-G expression plasmids are transfected into 293T cells to make VSV-G pseudotyped virus. These viruses are used to infect 293T-rtTA cells at MOI < 0.01 so that no more than one spike variant is integrated into each cell. Transduced cells are selected for lentiviral integration, and spike pseudotyped virus libraries are produced from these cells by transfecting helper plasmids in the presence of doxycycline to induce spike expression. In the absence of doxycycline and with added VSV-G expression plasmid, VSV-G pseudotyped virus libraries are also produced from the same cell lines; these VSV-G pseudotyped viruses are used to help estimate effects of spike mutations on cell entry as described in the next panel. **d**, Method used to measure effects of mutations in spike on cell entry. The ability of each spike variant to mediate cell entry is assessed by quantifying its relative frequency in 293T-ACE2 cells infected with spike-pseudotyped versus VSV-G pseudotyped libraries. **e**, Correlations between the effects of mutations on cell entry measured using each of the two independent full spike libraries of XBB.1.5 or BA.2. Throughout the rest of this paper, we report the mean value between the two libraries. **f**, Correlation between mutational effects on cell entry measured for the XBB.1.5 versus BA.2 full spike libraries. **g**, Cell entry effects for mutations in the pseudovirus libraries that saturate all mutations in the XBB.1.5 RBD. The black line indicates the median entry effect, and the boxes indicate the interquartile range. Note how the mutational effects in this saturated library tend to be more deleterious than the effects of RBD mutations in the XBB.1.5 full-spike libraries (Fig. 1b), since the full-spike libraries were designed with the goal of including mostly tolerated mutations rather than all mutations. **h**, Cell-entry effects as measured in the deep mutational scanning of mutations in either the flexible loops or core β -sheets of the NTD. The left plot shows the effects of amino-acid mutations; the right plot shows the effects of single-residue deletions. The black line indicates the median entry effect, and the boxes indicate the interquartile range. **i**, Correlation between mutational effects measured with the XBB.1.5 or BA.2 full spike libraries and fitness effects of those mutations estimated from actual human SARS-CoV-2 sequences¹⁵.

Extended Data Fig. 2: Correlations among measured mutational effects on ACE2 binding.

a, Correlation between effects of mutations on ACE2 binding measured with XBB.1.5 full spike and XBB.1.5 RBD pseudovirus libraries. **b**, Correlation between effects of mutations on ACE2 binding measured using XBB.1.5 RBD pseudovirus library with monomeric and dimeric ACE2. Heatmaps with the XBB.1.5 RBD pseudovirus measurements made using monomeric and dimeric ACE2 are at https://dms-vep.github.io/SARS-CoV-2_XBB.1.5_RBD_DMS/htmls/monomeric_ACE2_mut

effect.html and https://dms-vep.github.io/SARS-CoV-2_XBB.1.5_RBD_DMS/htmls/dimeric_ACE2_mut_effect.html, respectively **c**, Correlation between effects of mutations on ACE2 binding and spike-mediated cell entry for different libraries. **d**, ACE2 binding heat map showing key non-RBD sites that have mutated in the past major SARS-CoV-2 variants. Specific variant mutations are highlighted in red outline. Table on the right indicates variants in which these mutations occurred. Interactive plot showing ACE2 binding for all mutations measured in spike is at https://dms-vep.github.io/SARS-CoV-2_XBB.1.5_spike_DMS/htmls/monomeric_ACE2_mut_effect.html.

Extended Data Fig. 3: Mass photometry measurements for individual BA.2 spike variants

Spike molecular mass distributions measured using mass photometry for each biological replicate (blue and orange) corresponding to independent purification batches. Each row shows a BA.2 spike mutant and each column shows measurements at different ACE2 concentrations. In the absence of ACE2, some samples had a small peak to the left which may be a misfolded spike monomer⁸² which was present only in some protein preparations and is excluded from Gaussian curve fitting in the presence of ACE2.

Extended Data Fig. 4: Mass photometry measurements for individual XBB.1.5 spike variants

Spike molecular mass distributions measured using mass photometry for each biological replicate (blue and orange) corresponding to independent purification batches. Each row shows an XBB.1.5 spike mutant and each column shows measurements at different ACE2 concentrations. In the absence of ACE2, some samples had a small peak to the left which may be a misfolded spike

monomer⁸² which was present only in some protein preparations and is excluded from Gaussian curve fitting in the presence of ACE2.

Extended Data Fig. 5: BA.2 and its mutant spike preparations

a, Reducing SDS-PAGE gel for purified BA.2 wildtype and mutant spike ectodomains. All constructs are pre-fusion stabilized with HexaPro mutations²⁸. 3µg of purified protein loaded. Single major band for all samples confirms sample purity. **b**, Negative stain electron microscopy images for the purified BA.2 spike mutants to confirm proper folding and monodispersity of the samples. **c**, Reducing SDS-PAGE gel for purified XBB.1.5 wildtype and mutant spike ectodomains. All constructs are pre-fusion stabilized with HexaPro mutations²⁸. **d**, Negative stain electron microscopy images for the purified XBB.1.5 spike mutants.

Extended Data Fig. 6: Mass photometry measurements for S_1 and S_2 occupancy

a, Illustration of Gaussian components for no ($S^{0 \times ACE2}$), one ($S^{1 \times ACE2}$), two ($S^{2 \times ACE2}$), or three ($S^{3 \times ACE2}$) ACE2-bound spikes. $S^{1 \times RBD}$ occupancy is the fraction of spikes bound by one ACE2 molecule and $S^{2 \times RBD}$ occupancy is the fraction of spikes bound by two ACE2 molecules. **b**, Top row - $S^{1 \times RBD}$ occupancy measured using mass photometry for different BA.2 spike mutants. Bottom row - $S^{2 \times RBD}$ occupancy measured using mass photometry for different BA.2 spike mutants. **c**, Top row - $S^{1 \times RBD}$ occupancy measured using mass photometry for different XBB.1.5 spike mutants. Bottom row - $S^{2 \times RBD}$ occupancy measured using mass photometry for different XBB.1.5 spike mutants.

Extended Data Fig. 7: Correlation among serum escape mapping experiments

a, Correlation between mutation escape scores for experiments using the full-spike XBB.1.5 libraries performed on 293T cells expressing high or medium amounts of ACE2 for four sera. Note that the medium cells were used for all other figures shown in this paper. **b**, Correlation between mutation escape scores for mutations in the XBB.1.5 full spike and RBD-only libraries. See https://dms-vep.github.io/SARS-CoV-2_XBB.1.5_spike_DMS/htmls/compare_high_medium_ace2_escape.html and https://dms-vep.github.io/SARS-CoV-2_XBB.1.5_spike_DMS/htmls/compare_spike_rbd_escape.html for interactive versions of these scatter plots that also show line plots of per-site escape values for each serum.

Extended Data Fig. 8: Escape at key sites for each serum

Logoplots showing XBB.1.5 spike escape at 16 highest escape sites for each of the 10 sera measured. Letter heights indicate the escape caused by mutation to that amino acid. Letters are colored light yellow to dark brown depending on mutation effect on ACE2 binding. Left: all mutations measured. Right: mutations accessible with a single-nucleotide substitution.

Extended Data Fig. 9: Mutations in XBB.1.5 spike that increase serum neutralization

Escape at each site in the XBB.1.5 spike averaged across the 10 sera from individuals with prior XBB* infections, showing negative as well as positive values (**Fig. 4** only shows positive values). Sites with negative escape in this plot are ones where many mutations make spike more sensitive to neutralization. Interactive plots with site and mutation-level escape are at https://dms-vep.github.io/SARS-CoV-2_XBB.1.5_spike_DMS/htmls/summary_overlaid.html (set 'floor escape at zero' at the bottom of the chart to false to show negative escape).

Extended Data Fig. 10: Only antibodies that bind RBD in the up conformation are escaped by mutations outside the structural epitope

This figure shows previously generated and published deep mutational scanning escape maps for three monoclonal antibodies, two of which bind to RBD only in the up conformation (REGN10933 and SC27) and one of which binds to the RBD in both the up and down conformation

(LY-CoV1404). All antibodies are escaped by mutations in their direct structural epitope, but only the antibodies that bind only the up conformation are escaped by ACE2-distal mutations outside their epitope that affect RBD up/down conformation. **a**, REGN10933 escape profile mapped using a Delta full spike deep mutational scanning library⁴. REGN10933 only binds RBD in the up position^{83,84}. Line plot shows mean escape at each position in Delta spike with sites that modulate RBD movement highlighted in red. Heatmap shows mutation escape scores for sites highlighted in red on the line plot. Surface representation of RBD is coloured by site mean escape score with sites showing escape in the RBD outside the main antibody labeled (PDB ID: 6XDG). **b**, SC27 antibody escape profile mapped using the XBB.1.5 saturated RBD deep mutational scanning library⁸⁵. SC27 only binds RBD in the up conformation. (PDB ID: 7MMO). **c**, LY-CoV1404 escape profile mapped using the BA.1 full spike deep mutational scanning library⁴. LY-CoV1404 binds RBD in both up and down conformations⁸⁶. (PDB ID: 7MMO).

Extended Data Fig. 11: Sites with highest inverse correlation between ACE2 binding and serum escape

a, Correlation between ACE2 binding and serum escape for sites in XBB.1.5 spike. Only sites with at least 7 mutations measured and Pearson $r < 0.82$ are shown. **b**, Most sites with strongly negative correlations between mutational effects on ACE2 binding and escape are at positions that could plausibly impact the RBD conformation in the context of the full spike, since they tend to be at the interface of the RBD and other spike domains. Left: all sites from **a** shown on spike structure

as spheres. RBD is colored in light pink, NTD light blue, SD1 green and the S₂ subunit in yellow. Spheres are shown on only one chain for each domain for clarity (PDB ID: 8IOU). Right: RBD sites from **a** shown on RBD in up position engaged with ACE2. RBD is colored in light pink and ACE2 is gray.

Extended Data Fig. 12: Correlations in absolute clade growth with absolute clade phenotypes

a, Correlation between clade growth estimates made using the Murrell lab multinomial logistic regression model (see methods) or a hierarchical multinomial logistic regression implemented by the Bedford lab⁷³ (see <https://github.com/nextstrain/forecasts-ncov/>). Both sets of estimates are for clades designated after Jan-1-2023 and use the data available as of Oct-2-2023. The estimates are highly correlated, and everywhere else in this paper we report analyses using the Murrell lab estimates. **b**, Number of spike amino-acid mutations relative to the early Wuhan-Hu-1 virus in all SARS-CoV-2 Pango clades versus the clade designation dates. XBB-descended clades are in orange. As can be seen from this plot, newer clades tend to have more spike mutations. **c**, Because newer clades tend to have both more mutations and better growth, clade growth rate is trivially correlated with a clade’s relative distance (number of spike mutations) from Wuhan-Hu-1.

However, this correlation is not informative as it is already known that new clades tend to have more mutations. **d**, If we instead correlate the change in growth rate between parent-descendant clade pairs separated by at least one spike mutation (Fig. 6b) with the change in spike mutational distance to Wuhan-Hu-1 there is no correlation, since this approach removes the co-variation with total mutation count. Therefore, simple mutation counting is not informative for predicting changes in clade growth. **e**, Correlations for the phenotypes measured by the full spike deep mutational scanning in the current paper; **f**, the phenotypes measured in yeast display RBD deep mutational scanning; **g**, predicted by the EVEscape method. These plots differ from **Fig. 6c** and **Extended Data Fig. 13** in that they show the correlations in absolute clade growth with the absolute clade phenotypes, rather than comparing the changes in both for each parent-descendant clade pair. Absolute clade phenotypes are computed as the sum of mutation effects. The P-values above the plots are the fraction of times the correlation is greater than that for the actual data after randomizing the phenotypic effects among mutations. Note that the correlations are not reflective of the P-values (there can be high correlations but non-significant P-values) for the reasons noted in the main text and in **c**—phylogenetic correlations, and the fact that new clades have both more mutations and higher growth so that any “phenotype” that amounts to counting mutations gives a correlation in these plots. For this reason, comparing changes in clade growth to changes in spike phenotypes as done in **Fig. 6c** and **Extended Data Fig. 13** is the correct approach to test whether a method can actually predict which new clades will be successful.

Extended Data Fig. 13: Correlations of changes in growth with various other properties of spike for XBB descended clades

This figure shows the *change* in growth rate between parent-descendant clade pairs versus the *change* in various spike phenotypes, rather than showing the absolute clade growth and absolute

spike phenotypes as in **Extended Data Fig. 12**. Comparing the changes removes phylogenetic correlations as discussed in the main text. **a**, Correlation between the changes in growth rate for parent-descendant clade pairs versus the change in each spike phenotype measured in the XBB.1.5 full-spike deep mutational scanning described in the current paper (multiple mutations are assumed to have additive effects). These panels are the same as those shown in **Fig. 6c**, and are re-printed here to enable easier comparison to other panels in this figure. **b**, Correlations of changes in clade growth with changes in site-level antibody escape, ACE2 affinity, and RBD expression measured for RBD mutations in yeast-display deep mutational scanning. **c**, Correlation of changes in the EVEscape score with changes in clade growth. **d**, Ordinary least-squares regression of changes in the yeast-display RBD deep mutational scanning phenotypes versus changes in XBB-descendant clade growth. The small text indicates the unique variance explained by each variable as well as the coefficients in the regression. **e**, Ordinary least squares multiple linear regression of changes in XBB-descendant clade growth rate versus all three measured spike phenotypes using the XBB.1.5 full spike deep mutational scanning. This panel is the same as **Fig. 6d**, and is re-printed here to enable easier comparison to other panels in this figure. All panels are labeled with the Pearson correlation (r) and a P-value determined by computing how many randomizations of the mutational data yield correlations as large as the actual one.

a**b****c****d****f**
Extended Data Fig. 14: Correlations of changes in growth with various other properties of spike for BA.2, BA.5, and XBB descended clades

This figure is the same as **Extended Data Fig. 13** except that it includes clades descended from any of BA.2, BA.5, and XBB whereas **Extended Data Fig. 13** includes just clades descended from XBB.

Extended Data Fig. 15: Ability of various spike properties to distinguish the actual BA.2.86 and BA.2.86-descended clades from randomly mutated sequences

This figure assesses the ability of various spike properties to correctly identify that BA.2.86 and its descendant clades (which have spread widely and so by definition have high fitness) from other sequences with the same number of mutations drawn randomly from mutations observed in human SARS-CoV-2 sequences. This test is inspired by that used in the ninth extended data figure of Thadani et al⁶⁵. **a**, The blue circles show the phenotype of the actual BA.2.86 spike relative to its parent BA.2 as computed from the sum of the mutation effects measured in the XBB.1.5 full-spike deep mutational scanning, XBB.1.5 RBD yeast-display, or predicted by EVEscape. The gray shows min-max boxplots the phenotypes of 1,000 spike sequences generated by adding to the BA.2 spike the same number of amino-acid mutations in BA.2.86 relative to BA.2, drawing the mutations randomly from the set of all amino-acid mutations observed at least 50 times in GISAID. The P-value represents the fraction of these randomly mutated sequences with phenotypes at least as favorable as that of the actual BA.2.86 spike. Therefore, when the blue circle is far to the right of the gray distribution (small P value) that means the spike phenotype is highly effective at distinguishing the actual high-fitness BA.2.86 spike from randomly mutated sequences. For the panels labeled “spike pseudovirus DMS (combined phenotypes)” and “RBD yeast-display DMS

(combined phenotypes)”, the phenotype is a linear combination of the three phenotypes measured in each type of deep mutational scanning weighted with the coefficients determined by the multiple-linear regression on XBB-descended clades (see **Fig. 6c** and **Extended Data Fig. 13**). Overall, this figure shows that full-spike pseudovirus sera escape, full-spike pseudovirus cell entry, and RBD yeast-display ACE2 affinity are the three phenotypes with the best ability to distinguish the actual BA.2.86 spike from randomly mutated sequences. **b**, An analysis conceptually similar to that in panel a but comparing all designated Pango clades descended from BA.2.86 to their parental BA.2.86 versus a set of randomly mutated sequences generated by adding the same number of random mutations (observed at least 50 times in GISAID). The blue min-max boxplots show the distribution of the phenotype among the actual BA.2.86-descended clades that have evolved, whereas the gray shows the distribution of the phenotype among the randomly mutated sequences. When the blue distribution is shifted far to the right relative to the gray distribution, that indicates that the phenotype can effectively distinguish the actual clades that have evolved from randomly mutated ones. See https://dms-vep.org/SARS-CoV-2_XBB.1.5_spike_DMS/notebooks/compare_BA.2.86.html for the computer code implementing the analysis shown here.

Extended Data Fig. 16: Example of model fit to lineage counts

A representative example of the modeled lineage counts using multinomial logistic regression (lines) versus actual lineage counts (points) for Switzerland. Each color represents a different clade. These fits were used to estimate the clade growth rates.

Supplementary Table 1: Information on sera used in this study

Sera used in this study. Table shows the number and dates for infections and vaccinations each individual had. All individuals either had a confirmed XBB* infection (marked by * in the table above) or had the last recorded infection during the period when XBB or its descendant lineages were the most common circulating variants in Washington state. In February 2023 70% of sequenced cases were confirmed XBB or its descendant lineages and between March and May this number grew from 88% to 97% according to the samples sequenced at University of Washington Virology labs⁸⁷.

References

1. Liu, Y. *et al.* The N501Y spike substitution enhances SARS-CoV-2 infection and transmission. *Nature* **602**, 294–299 (2022).
2. Greaney, A. J. *et al.* Antibodies elicited by mRNA-1273 vaccination bind more broadly to the receptor binding domain than do those from SARS-CoV-2 infection. *Sci. Transl. Med.* **13**, eabi9915 (2021).
3. Cao, Y. *et al.* Imprinted SARS-CoV-2 humoral immunity induces convergent Omicron RBD evolution. *Nature* **614**, 521–529 (2023).
4. Dadonaite, B. *et al.* A pseudovirus system enables deep mutational scanning of the full SARS-CoV-2 spike. *Cell* **186**, 1263–1278.e20 (2023).
5. Crawford, K. H. D. *et al.* Protocol and Reagents for Pseudotyping Lentiviral Particles with SARS-CoV-2 Spike Protein for Neutralization Assays. *Viruses* **12**, 513 (2020).
6. Cantoni, D. *et al.* Correlation between pseudotyped virus and authentic virus neutralisation assays, a systematic review and meta-analysis of the literature. *Front. Immunol.* **14**, (2023).
7. Roemer Cornelius & Neher Richard. SARS-CoV-2 variant reports (2023-09-28). *GitHub* https://github.com/neherlab/SARS-CoV-2_variant-reports/tree/main/reports (2023).
8. WHO. Statement on the antigen composition of COVID-19 vaccines. <https://www.who.int/news/item/18-05-2023-statement-on-the-antigen-composition-of-covid-19-vaccines>.
9. Greaney, A. J., Starr, T. N. & Bloom, J. D. An antibody-escape estimator for mutations to the SARS-CoV-2 receptor-binding domain. *Virus Evol.* **8**, veac021 (2022).
10. Piccoli, L. *et al.* Mapping Neutralizing and Immunodominant Sites on the SARS-CoV-2 Spike Receptor-Binding Domain by Structure-Guided High-Resolution Serology. *Cell* **183**, 1024–1042.e21 (2020).
11. Haddox, H. K. *et al.* Jointly modeling deep mutational scans identifies shifted mutational effects among SARS-CoV-2 spike homologs. 2023.07.31.551037 Preprint at <https://doi.org/10.1101/2023.07.31.551037> (2023).
12. Starr, T. N. *et al.* Shifting mutational constraints in the SARS-CoV-2 receptor-binding domain during viral evolution. *Science* **377**, 420–424 (2022).
13. Cantoni, D. *et al.* Evolutionary remodelling of N-terminal domain loops fine-tunes SARS-CoV-2 spike. *EMBO Rep.* **23**, e54322 (2022).
14. McCallum, M. *et al.* N-terminal domain antigenic mapping reveals a site of vulnerability for SARS-CoV-2. *Cell* **184**, 2332–2347.e16 (2021).
15. Bloom, J. D. & Neher, R. A. Fitness effects of mutations to SARS-CoV-2 proteins. *Virus Evol.* **9**, vead055 (2023).
16. Wang, Q. *et al.* Antigenicity and receptor affinity of SARS-CoV-2 BA.2.86 spike. *Nature* 1–3 (2023) doi:10.1038/s41586-023-06750-w.
17. Wang, Q. *et al.* Antigenic characterization of the SARS-CoV-2 Omicron subvariant BA.2.75. *Cell Host Microbe* **30**, 1512–1517.e4 (2022).
18. Wang, Q. *et al.* Antibody evasion by SARS-CoV-2 Omicron subvariants BA.2.12.1, BA.4 and BA.5. *Nature* **608**, 603–608 (2022).
19. Bowen, J. E. *et al.* Omicron spike function and neutralizing activity elicited by a comprehensive panel of vaccines. *Science* **377**, 890–894.

20. Starr, T. N. *et al.* Deep mutational scans for ACE2 binding, RBD expression, and antibody escape in the SARS-CoV-2 Omicron BA.1 and BA.2 receptor-binding domains. *PLoS Pathog.* **18**, e1010951 (2022).
21. Addetia, A. *et al.* Neutralization, effector function and immune imprinting of Omicron variants. *Nature* **621**, 592–601 (2023).
22. Taylor, A. L. & Starr, T. N. Deep mutational scans of XBB.1.5 and BQ.1.1 reveal ongoing epistatic drift during SARS-CoV-2 evolution. 2023.09.11.557279 Preprint at <https://doi.org/10.1101/2023.09.11.557279> (2023).
23. Yurkovetskiy, L. *et al.* Structural and Functional Analysis of the D614G SARS-CoV-2 Spike Protein Variant. *Cell* **183**, 739–751.e8 (2020).
24. Henderson, R. *et al.* Controlling the SARS-CoV-2 spike glycoprotein conformation. *Nat. Struct. Mol. Biol.* **27**, 925–933 (2020).
25. Wang, Q. *et al.* Functional differences among the spike glycoproteins of multiple emerging severe acute respiratory syndrome coronavirus 2 variants of concern. *iScience* **24**, 103393 (2021).
26. Walls, A. C. *et al.* Unexpected Receptor Functional Mimicry Elucidates Activation of Coronavirus Fusion. *Cell* **176**, 1026–1039.e15 (2019).
27. Soltermann, F. *et al.* Quantifying Protein–Protein Interactions by Molecular Counting with Mass Photometry. *Angew. Chem. Int. Ed.* **59**, 10774–10779 (2020).
28. Hsieh, C.-L. *et al.* Structure-based design of prefusion-stabilized SARS-CoV-2 spikes. *Science* **369**, 1501–1505 (2020).
29. Escalera, A. *et al.* Mutations in SARS-CoV-2 variants of concern link to increased spike cleavage and virus transmission. *Cell Host Microbe* **30**, 373–387.e7 (2022).
30. Furusawa, Y. *et al.* In SARS-CoV-2 delta variants, Spike-P681R and D950N promote membrane fusion, Spike-P681R enhances spike cleavage, but neither substitution affects pathogenicity in hamsters. *eBioMedicine* **91**, 104561 (2023).
31. Ke, Z. *et al.* Virion morphology and on-virus spike protein structures of diverse SARS-CoV-2 variants. 2023.12.21.572824 Preprint at <https://doi.org/10.1101/2023.12.21.572824> (2023).
32. Low, J. S. *et al.* ACE2-binding exposes the SARS-CoV-2 fusion peptide to broadly neutralizing coronavirus antibodies. *Science* **377**, 735–742 (2022).
33. Weissman, D. *et al.* D614G Spike Mutation Increases SARS CoV-2 Susceptibility to Neutralization. *Cell Host Microbe* **29**, 23–31.e4 (2021).
34. Zhang, J. *et al.* Structural impact on SARS-CoV-2 spike protein by D614G substitution. *Science* **372**, 525–530 (2021).
35. Roemer Cornelius & Neher Richard. SARS-CoV-2 variant reports (2023-07-26). *GitHub* https://github.com/neherlab/SARS-CoV-2_variant-reports/blob/main/reports/variant_report_2023-07-26.md (2023).
36. Kaku, Y. *et al.* Antiviral efficacy of the SARS-CoV-2 XBB breakthrough infection sera against Omicron subvariants including EG.5. 2023.08.08.552415 Preprint at <https://doi.org/10.1101/2023.08.08.552415> (2023).
37. Roemer, C. & Neher, R. A. SARS-CoV-2_variant-reports (2024-03-04). *GitHub* https://github.com/neherlab/SARS-CoV-2_variant-reports/blob/main/reports/variant_report_latest_draft.md.
38. Farrell, A. G. *et al.* Receptor-Binding Domain (RBD) Antibodies Contribute More to SARS-CoV-2 Neutralization When Target Cells Express High Levels of ACE2. *Viruses* **14**, 2061 (2022).

39. Lempp, F. A. *et al.* Lectins enhance SARS-CoV-2 infection and influence neutralizing antibodies. *Nature* **598**, 342–347 (2021).
40. Liu, L. *et al.* Potent neutralizing antibodies against multiple epitopes on SARS-CoV-2 spike. *Nature* **584**, 450–456 (2020).
41. Starr, T. N. *et al.* SARS-CoV-2 RBD antibodies that maximize breadth and resistance to escape. *Nature* **597**, 97–102 (2021).
42. Zhao, Z. *et al.* Omicron SARS-CoV-2 mutations stabilize spike up-RBD conformation and lead to a non-RBM-binding monoclonal antibody escape. *Nat. Commun.* **13**, 4958 (2022).
43. Zheng, B. *et al.* S373P Mutation Stabilizes the Receptor-Binding Domain of the Spike Protein in Omicron and Promotes Binding. *JACS Au* **3**, 1902–1910 (2023).
44. Dokainish, H. M. *et al.* The inherent flexibility of receptor binding domains in SARS-CoV-2 spike protein. *eLife* **11**, e75720 (2022).
45. Liu, L. *et al.* Striking antibody evasion manifested by the Omicron variant of SARS-CoV-2. *Nature* **602**, 676–681 (2022).
46. Yang, T.-J. *et al.* Effect of SARS-CoV-2 B.1.1.7 mutations on spike protein structure and function. *Nat. Struct. Mol. Biol.* **28**, 731–739 (2021).
47. Casalino, L. *et al.* Beyond Shielding: The Roles of Glycans in the SARS-CoV-2 Spike Protein. *ACS Cent. Sci.* **6**, 1722–1734 (2020).
48. Henderson, R. *et al.* Glycans on the SARS-CoV-2 Spike Control the Receptor Binding Domain Conformation. 2020.06.26.173765 Preprint at <https://doi.org/10.1101/2020.06.26.173765> (2020).
49. Miller, N. L., Clark, T., Raman, R. & Sasisekharan, R. Insights on the mutational landscape of the SARS-CoV-2 Omicron variant receptor-binding domain. *Cell Rep. Med.* **3**, 100527 (2022).
50. Park, Y.-J. *et al.* Imprinted antibody responses against SARS-CoV-2 Omicron sublineages. *Science* **378**, 619–627 (2022).
51. Volz, E. *et al.* Evaluating the Effects of SARS-CoV-2 Spike Mutation D614G on Transmissibility and Pathogenicity. *Cell* **184**, 64–75.e11 (2021).
52. Roemer, C. *et al.* SARS-CoV-2 evolution in the Omicron era. *Nat. Microbiol.* 1–8 (2023) doi:10.1038/s41564-023-01504-w.
53. Felsenstein, J. Phylogenies and the Comparative Method. *Am. Nat.* **125**, 1–15 (1985).
54. Yisimayi, A. *et al.* Repeated Omicron exposures override ancestral SARS-CoV-2 immune imprinting. 2023.05.01.538516 Preprint at <https://doi.org/10.1101/2023.05.01.538516> (2023).
55. Thadani, N. N. *et al.* Learning from pre-pandemic data to forecast viral escape. *Nature* **622**, 818–825 (2023).
56. Khan, K. *et al.* Evolution and neutralization escape of the SARS-CoV-2 BA.2.86 subvariant. *Nat. Commun.* **14**, 8078 (2023).
57. Kim, K. *et al.* Measures of population immunity can predict the dominant clade of influenza A (H3N2) and reveal age-associated differences in susceptibility and specificity. 2023.10.26.23297569 Preprint at <https://doi.org/10.1101/2023.10.26.23297569> (2023).
58. Lee, J. M. *et al.* Mapping person-to-person variation in viral mutations that escape polyclonal serum targeting influenza hemagglutinin. *eLife* **8**, e49324 (2019).
59. Li, Y. *et al.* Immune history shapes specificity of pandemic H1N1 influenza antibody responses. *J. Exp. Med.* **210**, 1493–1500 (2013).
60. Bowen, J. E. *et al.* SARS-CoV-2 spike conformation determines plasma neutralizing activity elicited by a wide panel of human vaccines. *Sci. Immunol.* **7**, eadf1421 (2022).

61. Wrapp, D. *et al.* Cryo-EM structure of the 2019-nCoV spike in the prefusion conformation. *Science* **367**, 1260–1263 (2020).
62. Walls, A. C. *et al.* Structure, Function, and Antigenicity of the SARS-CoV-2 Spike Glycoprotein. *Cell* **181**, 281–292.e6 (2020).
63. Walls, A. C. *et al.* Glycan shield and epitope masking of a coronavirus spike protein observed by cryo-electron microscopy. *Nat. Struct. Mol. Biol.* **23**, 899–905 (2016).
64. Pronker, M. F. *et al.* Sialoglycan binding triggers spike opening in a human coronavirus. *Nature* 1–6 (2023) doi:10.1038/s41586-023-06599-z.
65. Tortorici, M. A. *et al.* Structural basis for human coronavirus attachment to sialic acid receptors. *Nat. Struct. Mol. Biol.* **26**, 481–489 (2019).
66. Li, Z. *et al.* The human coronavirus HCoV-229E S-protein structure and receptor binding. *eLife* **8**, e51230 (2019).
67. Song, X. *et al.* Cryo-EM analysis of the HCoV-229E spike glycoprotein reveals dynamic prefusion conformational changes. *Nat. Commun.* **12**, 141 (2021).
68. Dean, A. M. & Thornton, J. W. Mechanistic approaches to the study of evolution: the functional synthesis. *Nat. Rev. Genet.* **8**, 675–688 (2007).
69. Blount, Z. D., Lenski, R. E. & Losos, J. B. Contingency and determinism in evolution: Replaying life's tape. *Science* **362**, eaam5979 (2018).
70. Thorne, L. G. *et al.* Evolution of enhanced innate immune evasion by SARS-CoV-2. *Nature* **602**, 487–495 (2022).
71. Jian, F. *et al.* Convergent evolution of SARS-CoV-2 XBB lineages on receptor-binding domain 455-456 synergistically enhances antibody evasion and ACE2 binding. 2023.08.30.555211 Preprint at <https://doi.org/10.1101/2023.08.30.555211> (2023).
72. Meijers, M., Ruchnewitz, D., Eberhardt, J., Łuksza, M. & Lässig, M. Population immunity predicts evolutionary trajectories of SARS-CoV-2. *Cell* **186**, 5151–5164.e13 (2023).
73. Abousamra, E., Figgins, M. & Bedford, T. Fitness models provide accurate short-term forecasts of SARS-CoV-2 variant frequency. 2023.11.30.23299240 Preprint at <https://doi.org/10.1101/2023.11.30.23299240> (2024).
74. Khare, S. *et al.* GISAID's role in pandemic response. *China CDC Wkly.* **3**, 1049 (2021).
75. Turakhia, Y. *et al.* Ultrafast Sample placement on Existing tRees (USHER) enables real-time phylogenetics for the SARS-CoV-2 pandemic. *Nat. Genet.* **53**, 809–816 (2021).
76. Bloom, J. D. An Experimentally Determined Evolutionary Model Dramatically Improves Phylogenetic Fit. *Mol. Biol. Evol.* **31**, 1956–1978 (2014).
77. Otwinowski, J., McCandlish, D. M. & Plotkin, J. B. Inferring the shape of global epistasis. *Proc. Natl. Acad. Sci.* **115**, E7550–E7558 (2018).
78. Bell, A. J., Fegen, D., Ward, M. & Bank, A. RD114 envelope proteins provide an effective and versatile approach to pseudotype lentiviral vectors. *Exp. Biol. Med.* **235**, 1269–1276 (2010).
79. Tomás, H. A. *et al.* Improved GaLV-TR Glycoproteins to Pseudotype Lentiviral Vectors: Impact of Viral Protease Activity in the Production of LV Pseudotypes. *Mol. Ther. - Methods Clin. Dev.* **15**, 1–8 (2019).
80. Yu, T. C. *et al.* A biophysical model of viral escape from polyclonal antibodies. *Virus Evol.* **8**, veac110 (2022).
81. Pedregosa, F. *et al.* Scikit-learn: Machine Learning in Python. *J. Mach. Learn. Res.* **12**, 2825–2830 (2011).
82. Burnap, S. A. & Struwe, W. B. Mass photometry reveals SARS-CoV-2 spike stabilisation to impede

- ACE2 binding through altered conformational dynamics. *Chem. Commun.* **58**, 12939–12942 (2022).
83. Barnes, C. O. *et al.* SARS-CoV-2 neutralizing antibody structures inform therapeutic strategies. *Nature* **588**, 682–687 (2020).
 84. Hansen, J. *et al.* Studies in humanized mice and convalescent humans yield a SARS-CoV-2 antibody cocktail. *Science* **369**, 1010–1014 (2020).
 85. Voss, W. N. *et al.* Hybrid immunity to SARS-CoV-2 arises from serological recall of IgG antibodies distinctly imprinted by infection or vaccination. 2024.01.22.576742 Preprint at <https://doi.org/10.1101/2024.01.22.576742> (2024).
 86. Zhou, T. *et al.* Structural basis for potent antibody neutralization of SARS-CoV-2 variants including B.1.1.529. *Science* **376**, eabn8897 (2022).
 87. UW Virology COVID-19 Dashboard. <https://depts.washington.edu/labmed/covid19/#sequencing-information>.

Reviewer Reports on the Second Revision:

Referees' comments:

Referee #1 (Remarks to the Author):

The authors have answered all my questions. The discussion of 504 mutations is inspiring.